# Bellman-consistent Pessimism for Offline Reinforcement Learning

**Tengyang Xie**
UIUC
tx10@illinois.edu

**Ching-An Cheng**
Microsoft Research
chinganc@microsoft.com

**Nan Jiang**
UIUC
nanjiang@illinois.edu

**Paul Mineiro**
Microsoft Research
pmineiro@microsoft.com

**Alekh Agarwal**
Google Research
alekhagarwal@google.com

## Abstract

The use of pessimism, when reasoning about datasets lacking exhaustive exploration, has recently gained prominence in offline reinforcement learning. Despite the robustness it adds to the algorithm, overly pessimistic reasoning can be equally damaging in precluding the discovery of good policies, which is an issue for the popular bonus-based pessimism. In this paper, we introduce the notion of Bellman-consistent pessimism for general function approximation: instead of calculating a point-wise lower bound for the value function, we implement pessimism at the initial state over the set of functions consistent with the Bellman equations. Our theoretical guarantees only require Bellman closedness as standard in the exploratory setting, in which case bonus-based pessimism fails to provide guarantees. Even in the special case of linear function approximation where stronger expressivity assumptions hold, our result improves upon a recent bonus-based approach by $\mathcal{O}(d)$ in its sample complexity when the action space is finite and small. Remarkably, our algorithms automatically adapt to the best bias-variance tradeoff in the hindsight, whereas most prior approaches require tuning extra hyperparameters a priori.

## 1 Introduction

Using past experiences to learn improved behavior for future interactions is a critical capability for a Reinforcement Learning (RL) agent. However, robustly extrapolating knowledge from a historical dataset for sequential decision making is highly challenging, particularly in settings where function approximation is employed to generalize across related observations. In this paper, we provide a systematic treatment of such scenarios with *general* function approximation, and devise algorithms that can provably leverage an arbitrary historical dataset to discover the policy that obtains the largest guaranteed rewards, amongst all possible scenarios consistent with the dataset.

The problem of learning a good policy from historical datasets, typically called batch or offline RL, has a long history [see e.g., Precup et al., 2000; Antos et al., 2008; Levine et al., 2020, and references therein]. Many prior works [e.g., Precup et al., 2000; Antos et al., 2008; Chen and Jiang, 2019] make the so-called *coverage assumptions* on the dataset, requiring the dataset to contain any possible state, action pair or trajectory with a lower bounded probability. These assumptions are evidently prohibitive in practice, particularly for problems with large state and/or action spaces. Furthermore, the methods developed under these assumptions routinely display unstable behaviors such as lack of convergence or error amplification, when coverage assumptions are violated [Wang et al., 2020, 2021].

35th Conference on Neural Information Processing Systems (NeurIPS 2021).

Driven by these instabilities, a growing body of recent literature has pursued a so-called *best effort* style of guarantee instead. The key idea is to replace the stringent assumptions on the dataset with a dataset-dependent performance bound, which gracefully degrades from guaranteeing a near-optimal policy under standard coverage assumptions to offering no improvement over the data collection policy in the most degenerate case. Algorithmically, these works all leverage the principle of *pessimistic extrapolation* from offline data and aim to maximize the rewards the trained agent would obtain in the worst possible MDP that is consistent with the observed dataset. These methods have been shown to be typically more robust to the violation of coverage assumptions in practice, and their theoretical guarantees often provide non-trivial conclusions in settings where the previous results did not apply.

Even though many such best-effort methods have now been developed, very few works provide a comprehensive theory for using generic function approximation, unlike the setting where the dataset satisfies the coverage assumptions [Antos et al., 2008; Munos, 2003; Szepesvári and Munos, 2005; Munos and Szepesvári, 2008; Farahmand et al., 2010; Chen and Jiang, 2019; Xie and Jiang, 2020]. For example, [Kidambi et al., 2020] provides a partial theory under the assumption of an uncertainty quantification oracle, which however is highly nontrivial to obtain for general function approximation. [Fujimoto et al., 2019; Kumar et al., 2020] develop sound theoretical arguments in the tabular setting, which were only heuristically extended to the function approximation setting. The works that explicitly consider function approximation in their design either use an ad-hoc truncation of Bellman backups [Liu et al., 2020] or strongly rely on particular parameterizations such as linear function approximation [Jin et al., 2021]. In particular, [Liu et al., 2020] additionally requires the ability to approximate stationary distribution of the behavior policy, which is a challenging density estimation problem for complex state spaces and cannot be provably performed in the standard linear MDP setting (see Section 3.1).

Our paper takes an important step in this direction. We provide a systematic way to encode pessimism compatible with an arbitrary function approximation class and MDP and give strong theoretical guarantees *without* requiring any coverage assumptions on the dataset. Our first contribution is an information theoretic algorithm that returns a policy with a small regret to any comparator policy, for which coverage assumptions (approximately) hold with respect to the data collection policy. This regret bound is identical to what can be typically obtained when the coverage assumptions hold *for all policies* [Antos et al., 2008; Chen and Jiang, 2019]. But our algorithm requires neither the coverage assumptions, nor additional assumptions such as reliable density estimation for the data generating distribution used by existing best-effort approaches [Liu et al., 2020]. We furthermore instantiate these results in the special case of linear parameterization; under the linear MDP assumption, our sample complexity bound leads to a factor of $\mathcal{O}(d)$ improvement for a $d$-dimensional linear MDP, compared with the best known result translated to our discounted setting [Jin et al., 2021], when the action set is small in size. In addition to the information theoretic algorithm, we also develop a computationally practical version of our algorithm using a Lagrangian relaxation combined with recent advances in soft policy iteration [Even-Dar et al., 2009; Geist et al., 2019; Agarwal et al., 2019]. We show that this algorithm can be executed efficiently by querying a (regularized) loss minimization oracle over the value function class, although it has slightly worse theoretical guarantees than the information theoretic version. Both our algorithms display an adaptive property in selecting the best possible form of a bias-variance decomposition, where most prior approaches had to commit to a particular point through their choice of hyperparameters (see the discussion following Theorem 3.1).

## 2  Preliminaries

**Markov Decision Processes**  We consider dynamical systems modeled as Markov Decision Processes (MDPs). An MDP is specified by $(\mathcal{S}, \mathcal{A}, P, R, \gamma, s_0)$, where $\mathcal{S}$ is the state space, $\mathcal{A}$ is the action space, $P : \mathcal{S} \times \mathcal{A} \to \Delta(\mathcal{S})$ is the transition function with $\Delta(\cdot)$ being the probability simplex, $R : \mathcal{S} \times \mathcal{A} \to [0, R_{\max}]$ is the reward function, $\gamma \in [0, 1)$ is the discount factor, and $s_0$ is a deterministic initial state, which is without loss of generality. We assume the state and the action spaces are finite but can be arbitrarily large. A (stochastic) policy $\pi : \mathcal{S} \to \Delta(\mathcal{A})$ specifies a decision-making strategy, and induces a random trajectory $s_0, a_0, r_0, s_1, a_1, r_1, \ldots$, where $a_t \sim \pi(\cdot|s_t), r_t = R(s_t, a_t), s_{t+1} \sim P(\cdot|s_t, a_t), \forall t \geq 0$. We denote the expected discounted return of a policy $\pi$ as $J(\pi) := \mathbb{E}[\sum_{t=0}^{\infty} \gamma^t r_t | \pi]$, and the learning goal is to find the maximizer of this value: $\pi^\star := \arg\max_\pi J(\pi)$. A related concept is the *policy-specific Q-function*, $Q^\pi : \mathcal{S} \times \mathcal{A} \to \mathbb{R}$. $Q^\pi(s, a)$ is the discounted return when the trajectory starts with $(s, a)$ and all remaining actions

are taken according to $\pi$. $Q^\pi$ is the unique fixed point of the (policy-specific) Bellman operator $\mathcal{T}^\pi : \mathbb{R}^{\mathcal{S} \times \mathcal{A}} \to \mathbb{R}^{\mathcal{S} \times \mathcal{A}}$, defined as:

$$\forall f, \quad (\mathcal{T}^\pi f)(s, a) = R(s, a) + \gamma \mathbb{E}_{s' \sim P(\cdot|s,a)}[f(s', \pi)],$$

where $f(s', \pi)$ is a shorthand for $\mathbb{E}_{a' \sim \pi(\cdot|s')}[f(s', a')]$.

Another important concept is the notion of discounted state-action occupancy, $d_\pi \in \Delta(\mathcal{S} \times \mathcal{A})$, defined as $d_\pi(s, a) := (1 - \gamma)\mathbb{E}[\sum_{t=0}^\infty \gamma^t \mathbb{1}[s_t = s, a_t = a]|\pi]$, which characterizes the states and actions visited by a policy $\pi$.

**Offline RL**   In the offline setting, the learner only has access to a pre-collected dataset and cannot directly interact with the environment. We assume the standard i.i.d. data generation protocol in our theoretical derivations, that the offline dataset $\mathcal{D}$ consists of $n$ i.i.d. $(s, a, r, s')$ tuples generated as $(s, a) \sim \mu, r = R(s, a), s' \sim P(\cdot|s, a)$ for some *data distribution* $\mu$. We will also use $\mathbb{E}_\mu[\cdot]$ for taking expectation with respect to $\mu$. We will frequently use the data-weighted 2-norm (squared) $\|f\|_{2,\mu}^2 := \mathbb{E}_\mu[f^2]$, and the definition extends when we replace $\mu$ with any other state-action distribution $\nu$. The empirical approximation of $\|f\|_{2,\mu}^2$ is $\|f\|_{2,\mathcal{D}}^2 := \frac{1}{n} \sum_{(s,a,r,s') \in \mathcal{D}} f(s, a)^2$.

**Function Approximation**   Function approximation is crucial to generalizing over large and complex state and action spaces. In this work, we search for a good policy in a policy class $\Pi \subset (\mathcal{S} \to \Delta(\mathcal{A}))$ with the help of a value-function class $\mathcal{F} \subset (\mathcal{S} \times \mathcal{A} \to [0, V_{\max}])$ to model $Q^\pi$, where $V_{\max} = R_{\max}/(1 - \gamma)$. Such a combination is commonly found in approximate policy iteration and actor-critic algorithms [e.g., Bertsekas and Tsitsiklis, 1996; Konda and Tsitsiklis, 2000]. For most part of the paper we do not make any structural assumptions on $\Pi$ and $\mathcal{F}$, making our approach and guarantees applicable to generic function approximators. For simplicity we will assume that these function classes are finite but exponentially large, and use log-cardinality to measure their statistical complexities in the generic results (Section 3 and Section 4). These guarantees easily extend to continuous function classes where log-cardinalities are replaced by the appropriate notions of covering numbers, which we demonstrate when we instantiate our results in the linear function approximation setting and work with continuous linear classes (Section 3.1).

We now recall two standard expressivity assumptions on $\mathcal{F}$ [e.g., Antos et al., 2008]. To our knowledge, no existing works on offline RL with insufficient data coverage have provided guarantees under these standard assumptions for general function approximation, and they often require stronger or tweaked assumptions (see Section 1).

**Assumption 1** (Realizability). *For any $\pi \in \Pi$, we have*

$$\inf_{f \in \mathcal{F}} \sup_{admissible\ \nu} \|f - \mathcal{T}^\pi f\|_{2,\nu}^2 \leq \varepsilon_\mathcal{F},$$

*where an admissible distribution $\nu$ means that $\nu \in \{d_{\pi'} : \pi' \in \Pi\}$.*

Assumption 1 requires that for every $\pi \in \Pi$, there exists $f \in \mathcal{F}$ that well-approximates $Q^\pi$. This assumption is often called *realizability*.[1] Technically this is asserted by requiring $f$ to have small Bellman error w.r.t. $\mathcal{T}^\pi$ under all possible *admissible* distributions. As a sufficient condition, we have $\varepsilon_\mathcal{F} = 0$ if $Q^\pi \in \mathcal{F}, \forall \pi \in \Pi$.

**Assumption 2** (Completeness). *For any $\pi \in \Pi$, we have*

$$\sup_{f \in \mathcal{F}} \inf_{f' \in \mathcal{F}} \|f' - \mathcal{T}^\pi f\|_{2,\mu}^2 \leq \varepsilon_{\mathcal{F},\mathcal{F}}.$$

Assumption 2 asserts that $\mathcal{F}$ is approximately closed under $\mathcal{T}^\pi$.[2] Such an assumption is widely used in RL theory and can be only avoided in some rare cases [Xie and Jiang, 2021], and the hardness of learning with realizability alone has been established in various settings (e.g., [Weisz et al., 2021; Zanette, 2021]). We also emphasize that we only measure the violation of completeness under $\mu$ and do not need to reason about all admissible distributions.

---

[1]In the exploratory setting, realizability is usually stated in the form of $\inf_{f \in \mathcal{F}} \|f - \mathcal{T}^\pi f\|_{2,\mu}^2 \leq \varepsilon_\mathcal{F}$ for any $\pi \in \Pi$. However, the exploratory setting also usually has data coverage assumptions in the form of $\sup_\nu \|\nu/\mu\|_\infty \leq C$. Combining them together implies Assumption 1.

[2]Sometimes completeness implies realizability, so the latter does not need to be assumed separately [Chen and Jiang, 2019]. However, this often relies on $\mu$ being exploratory, which is not the case here.

**Distribution shift** A unique challenge in RL is that the learned policy may induce a state (and action) distribution that is different from the data distribution $\mu$, and the issue is particularly salient when we do not impose any coverage assumption on $\mu$. Therefore, it is important to carefully characterize the distribution shift, which we measure using the following definition, which generalizes prior definitions specific to linear function approximation [Agarwal et al., 2019; Duan et al., 2020]:

**Definition 1.** *We define $\mathscr{C}(\nu; \mu, \mathcal{F}, \pi)$ as follows to measure the distribution shift from an arbitrary distribution $\nu$ to the data distribution $\mu$, w.r.t. $\mathcal{F}$ and $\pi$,*

$$\mathscr{C}(\nu; \mu, \mathcal{F}, \pi) := \max_{f \in \mathcal{F}} \frac{\|f - \mathcal{T}^\pi f\|_{2,\nu}^2}{\|f - \mathcal{T}^\pi f\|_{2,\mu}^2}.$$

Intuitively, $\mathscr{C}(\nu; \mu, \mathcal{F}, \pi)$ measures how well Bellman errors under $\pi$ transfer between the distributions $\nu$ and $\mu$. For instance, a small value of $\mathscr{C}(d_\pi; \mu, \mathcal{F}, \pi)$ enables accurate policy evaluation for $\pi$ using data collected under $\mu$. More generally, we observe that

$$\mathscr{C}(\nu; \mu, \mathcal{F}, \pi) \le \|\nu/\mu\|_\infty := \sup_{s,a} \frac{\nu(s,a)}{\mu(s,a)}, \quad \text{for any } \pi, \mathcal{F}.$$

and the RHS is a classical notion of bounded distribution ratio for error transfer (e.g., [Munos and Szepesvári, 2008; Chen and Jiang, 2019; Xie and Jiang, 2020]). Moreover, our measure can be tighter than $\|\nu/\mu\|_\infty$: Even two distributions $\nu$ and $\mu$ that are sufficiently disparate might admit a reasonable transfer, so long as this difference is not detected by $\pi$ and $\mathcal{F}$. To this end, our definition better captures the crucial role of function approximation in generalizing across different states. As an example, in the special case of linear MDPs, full coverage under our definition (i.e., boundedness of $\mathscr{C}$ *for all admissible $\nu$*) can be implied from the standard coverage assumption for linear MDPs that considers the spectrum of the feature covariance matrix under $\mu$; see Section 3.1 for more details.

## 3 Information-Theoretic Results with Bellman-consistent Pessimism

In this section, we provide our first theoretical result which is information-theoretic, in that it uses a computationally inefficient algorithm. The approach uses the offline dataset to first compute a lower bound on the value of each policy $\pi \in \Pi$, and then returns the policy with the highest pessimistic value estimate. While this high-level template is at the heart of many recent approaches [e.g., Fujimoto et al., 2019; Kumar et al., 2019; Liu et al., 2020; Kidambi et al., 2020; Yu et al., 2020; Kumar et al., 2020], our main novelty is in the design and analysis of *Bellman-consistent pessimism* for general function approximation.

For a policy $\pi$, we first form a *version space* of all the functions $f \in \mathcal{F}$ which have a small Bellman error under the evaluation operator $\mathcal{T}^\pi$. We then return the predicted value of $\pi$ in the initial state $s_0$ by the functions in this version space. The use of pessimism at the initial state, while maintaining Bellman consistency (by virtue of having a small Bellman error) limits over pessimism, which is harder to preclude in the pointwise pessimistic penalties used in some other works [Jin et al., 2021]. More formally, given a dataset $\mathcal{D}$, let us define

$$\mathcal{L}(f', f, \pi; \mathcal{D}) := \frac{1}{n} \sum_{(s,a,r,s') \in \mathcal{D}} \left( f'(s,a) - r - \gamma f(s', \pi) \right)^2,$$

and an empirical estimate of the Bellman error $\mathcal{E}(f, \pi; \mathcal{D})$ is

$$\mathcal{E}(f, \pi; \mathcal{D}) := \mathcal{L}(f, f, \pi; \mathcal{D}) - \min_{f' \in \mathcal{F}} \mathcal{L}(f', f, \pi; \mathcal{D}). \tag{3.1}$$

**Our algorithm.** With this notation, our information-theoretic approach finds a policy by optimizing:

$$\widehat{\pi} = \operatorname*{argmax}_{\pi \in \Pi} \min_{f \in \mathcal{F}_{\pi,\varepsilon}} f(s_0, \pi), \quad \text{where } \mathcal{F}_{\pi,\varepsilon} = \{ f \in \mathcal{F} : \mathcal{E}(f, \pi; \mathcal{D}) \le \varepsilon \}, \tag{3.2}$$

In the formulation above, $\mathcal{F}_{\pi,\varepsilon}$ is the version space of policy $\pi$. To better understand the intuition behind the estimator in Equation 3.2, let us define

$$f_{\pi,\min} := \operatorname*{argmin}_{f \in \mathcal{F}_{\pi,\varepsilon}} f(s_0, \pi), \ f_{\pi,\max} := \operatorname*{argmax}_{f \in \mathcal{F}_{\pi,\varepsilon}} f(s_0, \pi), \ \text{and } \Delta f_\pi(s,a) := f_{\pi,\max}(s,a) - f_{\pi,\min}(s,a).$$

Intuitively, if the parameter $\varepsilon$ is defined to ensure that $Q^\pi$ (or its best approximation in $\mathcal{F}$) is in $\mathcal{F}_{\pi,\varepsilon}$, we easily see that $\Delta f_\pi(s_0, \pi)$ is an upper bound on the error in our estimate of $J(\pi)$ for any $\pi \in \Pi$. In fact, an easy argument in our analysis shows that if $Q^\pi \in \mathcal{F}_{\pi,\varepsilon}$ for all $\pi \in \Pi$, then $\Delta f(s_0, \pi)$ is an upper bound on the regret $J(\pi) - J(\widehat{\pi})$ of our estimator relative to any $\pi$ we wish to compete with.

**Theoretical analysis.** To leverage this observation, we first define the a critical threshold $\varepsilon_r$ which ensures that (the best approximation of) $Q^\pi$ is indeed contained in our version spaces for all $\pi$:

$$\varepsilon_r := \frac{139 V_{\max}^2 \log \frac{|\mathcal{F}||\Pi|}{\delta}}{n} + 39\varepsilon_{\mathcal{F}}. \tag{3.3}$$

With this definition, we now give a more refined bound on the regret of our algorithm (3.2) by further splitting the error estimate $\Delta f_\pi(s_0, \pi)$ which is random owing to its dependence on the version space, and analyze it through a novel decomposition into on-support and off-support components. While we bound the on-support error using standard techniques, the off-support error is akin to a bias term which captures the interplay between the data collection distribution and function approximation in the quality of the final solution. Also note that our choice of $\varepsilon_r$ requires the knowledge of $\varepsilon_{\mathcal{F}}$, which is a common characteristic of version-space-based algorithms [e.g., Jiang et al., 2017]. The challenge of unknown $\varepsilon_F$ can be possibly addressed using model-selection techniques in practice and we leave further investigation to future work.

**Theorem 3.1.** *Let $\varepsilon = \varepsilon_r$ where is $\varepsilon_r$ defined in Eq.(3.3) and $\widehat{\pi}$ be obtained by Eq.(3.2). Then, for any policy $\pi \in \Pi$ and any constant $C_2 \geq 1$, with probability at least $1 - \delta$,*

$$J(\pi) - J(\widehat{\pi}) \leq \underbrace{\mathcal{O}\left(\frac{V_{\max}\sqrt{C_2}}{1-\gamma}\sqrt{\frac{\log \frac{|\mathcal{F}||\Pi|}{\delta}}{n}} + \frac{\sqrt{C_2(\varepsilon_{\mathcal{F},\mathcal{F}} + \varepsilon_{\mathcal{F}})}}{1-\gamma}\right)}_{\mathrm{err_{on}}(\pi):\ \textit{on-support error}}$$

$$+ \underbrace{\frac{1}{1-\gamma} \cdot \min_{\nu:\mathscr{C}(\nu;\mu,\mathcal{F},\pi) \leq C_2} \sum_{(s,a)\in\mathcal{S}\times\mathcal{A}} (d_\pi \setminus \nu)(s,a)\left[\Delta f_\pi(s,a) - \gamma(\mathcal{P}^\pi \Delta f_\pi)(s,a)\right],}_{\mathrm{err_{off}}(\pi):\ \textit{off-support error}}$$

*where $\mathscr{C}(\nu;\mu,\mathcal{F},\pi)$ is defined in Definition 1, $(d_\pi \setminus \nu)(s,a) := \max(d_\pi(s,a) - \nu(s,a), 0)$ and $(\mathcal{P}^\pi f)(s,a) = \mathbb{E}_{s'\sim P(\cdot|s,a)}[f(s',\pi)]$ for any $f$.*

**Bias-variance decomposition.** Note that decomposition of our error bound into on-support and off-support parts effectively achieves a bias-variance tradeoff. A small value of the concentrability threshold $C_2$ requires the choice of the distribution $\nu$ closer to $\mu$, which results in better estimation error guarantee (which is $\mathcal{O}(\sqrt{C_2/n})$) when we transfer from $\mu$ to $\nu$, but potentially pays a high bias due to the mismatch between $d_\pi$ and $\nu$. A larger threshold permits more flexibility in choosing $\nu$ similar to $d_\pi$ for a smaller bias, but results in a larger variance and estimation error. Rather than commit to a particular tradeoff, our estimator automatically adapts to the best possible splitting (Figure 1 illustrates this concept) by allowing us to choose the best threshold $C_2$. The on-support part matches the $n$ rate (fast rate error bound) of API or AVI analysis

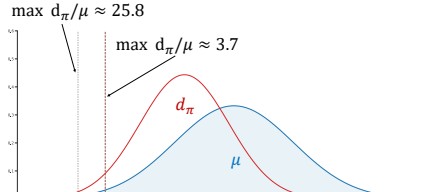

max $d_\pi/\mu \approx 25.8$

max $d_\pi/\mu \approx 3.7$

$d_\pi$

$\mu$

Figure 1: An example illustrating different on-support and off-support splittings (denoted by two different vertical lines). Different splitting has different $C_2$ values, and further yields different bias-variance trade-offs.

(e.g., [Pires and Szepesvári, 2012; Lazaric et al., 2012; Chen and Jiang, 2019]). The dependency on horizon is only linear and matches the best previous result with concentrability assumption [Xie and Jiang, 2020]. For the off-support part, it depends on the off-support mass $d_\pi \setminus \nu$ and the "quality" of the off-support estimation: if all value functions in the version space are close to each other in the off-support region for policy $\pi$, the gap between $J(\pi)$ and $J(\widehat{\pi})$ can still be small even with a large off-support mass. The following corollary formally states this property.

**Corollary 1** ("Double Robustness"). *Under conditions of Theorem 3.1, for any $\pi$ and $C_2 \geq 0$, $\mathrm{err_{off}}(\pi) = 0$ when either (1) $\mathscr{C}(d_\pi; \mu, \mathcal{F}, \pi) \leq C_2$, or, (2) $\Delta f_\pi - \gamma\mathcal{P}^\pi \Delta f_\pi \equiv 0$.*

**Adaptive guarantees by algorithm design.** As mentioned above, Theorem 3.1 implicitly selects the best bias-variance decomposition through the best choice of $C_2$ in hindsight, with this decomposition being purely a proof technique, not an knob in the algorithm. In contrast, many prior approaches [Liu et al., 2020; Fujimoto et al., 2019; Kumar et al., 2019] employ explicit thresholds to control density ratios in their algorithms, which makes the tradeoff a hyperparameter in their algorithms. Since choosing hyperparameters is particularly challenging in offline RL, where even policy evaluation can be unreliable, this novel axis of adaptivity is an extremely desirable property of our approach.

**Comparison to guarantees in the exploratory setting.** When a dataset with full coverage is given, classical analyses provide near-optimality guarantees that compete with the optimal policy $\pi^\star$ with a polynomial sample complexity, when $\pi^\star \in \Pi$ and both realizability and completeness hold for $\mathcal{F}$; see [Antos et al., 2008] for a representative analysis. As mentioned earlier, such analysis often requires boundedness of $\|\nu/\mu\|_\infty$ *for all admissible distributions* $\nu \in \{d_\pi : \pi \in \Pi\}$. On the other hand, it is easily seen that we can compete with $\pi^\star$ under much weaker conditions.

**Corollary 2** (Competing with optimal policy). *Under conditions of Theorem 3.1, if* $\mathscr{C}(d_{\pi^\star}; \mu, \mathcal{F}, \pi^\star) \le C_2$, *we have*

$$J(\pi^\star) - J(\widehat{\pi}) \le \mathcal{O}\left(\frac{V_{\max}\sqrt{C_2}}{1-\gamma}\sqrt{\frac{\log\frac{|\mathcal{F}||\Pi|}{\delta}}{n}} + \frac{\sqrt{C_2(\varepsilon_{\mathcal{F},\mathcal{F}} + \varepsilon_{\mathcal{F}})}}{1-\gamma}\right).$$

Notably, these milder coverage assumptions in Corollaries 1 and 2 provide offline RL counterparts to the benefits of policy-gradient-style methods with *online* access to the environment [Kakade and Langford, 2002; Scherrer, 2014; Agarwal et al., 2019].

**Comparison with Liu et al. [2020].** The closest prior result to our work is that of [Liu et al., 2020], who develop a pessimistic estimator that truncates Bellman backups from state-action pairs infrequently visited by $\mu$, and analyzes the resulting pessimistic policy and value iteration algorithms under general function approximation. For truncating Bellman backups, however, their work requires estimating the state-action distribution of data, which can be challenging in high-dimensions and they incur additional errors from density estimation which we avoid. Further, their algorithms only compete with policies $\pi$ where $\|d_\pi/\mu\|_\infty$ is bounded instead of the more general result that we provide, and makes their results vacuous in a linear MDP setting under typical feature coverage assumptions.

**Safe Policy Improvement.** Some prior works [e.g. Laroche et al., 2019; Liu et al., 2020] discuss the scenario where the dataset $\mathcal{D}$ is collected with a behavior policy $\pi_b$ with $\mu = d_{\pi_b}$, and demonstrate that their algorithms always return a policy competitive with $\pi_b$. In our setup, this is straightforward as $d_{\pi_b}$ is always covered, as shown next.

**Corollary 3** (Bounded degradation from behavior policy). *Under conditions of Theorem 3.1, if* $\mu = d_{\pi_b}$ *for some policy* $\pi_b \in \Pi$, *we have*

$$J(\pi_b) - J(\widehat{\pi}) \le \mathcal{O}\left(\frac{V_{\max}}{1-\gamma}\sqrt{\frac{\log\frac{|\mathcal{F}||\Pi|}{\delta}}{n}} + \frac{\sqrt{\varepsilon_{\mathcal{F},\mathcal{F}} + \varepsilon_{\mathcal{F}}}}{1-\gamma}\right).$$

*Proof sketch of Theorem 3.1.* We now briefly describe the core ideas in the proof. More detailed arguments are deferred to the full proof in Appendix B.1.

The key to prove Theorem 3.1 is to translate the $J(\pi) - J(\widehat{\pi})$ to the Bellman error of value functions in the version space $\mathcal{F}_{\pi,\varepsilon}$. Our main proving strategies are as follows:

1. As the selection of $\varepsilon = \varepsilon_r$ ensures the accurate estimation $Q^\pi$ is contained in the version space $\mathcal{F}_{\widehat{\pi},\varepsilon}$ for any $\pi$, we can obtain $J(\pi) - J(\widehat{\pi}) \le \max_{f \in \mathcal{F}_{\pi,\varepsilon}} f(s_0, \pi) - \min_{f \in \mathcal{F}_{\widehat{\pi},\varepsilon}} f(s_0, \widehat{\pi}) +$ approximation error.
2. By the optimality of $\widehat{\pi}$, we have $\min_{f \in \mathcal{F}_{\widehat{\pi},\varepsilon}} f(s_0, \widehat{\pi}) \ge \min_{f \in \mathcal{F}_{\pi,\varepsilon}} f(s_0, \pi)$. This indicates that $J(\pi) - J(\widehat{\pi}) \le \max_{f \in \mathcal{F}_{\pi,\varepsilon}} f(s_0, \pi) - \min_{f \in \mathcal{F}_{\pi,\varepsilon}} f(s_0, \pi) +$ approximation error.

3. By using a standard telescoping argument (e.g., [Xie and Jiang, 2020, Lemma 1]), $\max_{f \in \mathcal{F}_{\pi,\varepsilon}} f(s_0, \pi) - \min_{f \in \mathcal{F}_{\pi,\varepsilon}} f(s_0, \pi)$ can be upper bounded by the Bellman error of $\operatorname{argmax}_{f \in \mathcal{F}_{\pi,\varepsilon}} f(s_0, \pi)$ and $\operatorname{argmin}_{f \in \mathcal{F}_{\pi,\varepsilon}} f(s_0, \pi)$ over distribution $d_\pi$.

After combining all the three steps above together and considering the distribution shift effect, we complete the proof. □

## 3.1 Results for Linear Function Approximation

Here we perform a case study in linear function approximation. We will show that our results—when instantiated under linear function approximation (with realizability and completeness assumptions)—automatically provides state-of-the-art guarantees, improving over existing results specialized to this setting by a factor of $\mathcal{O}(d)$ [Jin et al., 2021] when the action space is finite and small.

We recall the linear function approximation setup (we set $R_{\max} = 1$ and $V_{\max} = \frac{1}{1-\gamma}$ for consistency with literature).

**Definition 2** (Linear Function Approximation). *Let $\phi : \mathcal{S} \times \mathcal{A} \to \mathbb{R}^d$ be a feature mapping. Without loss of generality, we assume $\|\phi(s,a)\|_2 \le 1, \forall(s,a) \in \mathcal{S} \times \mathcal{A}$. We define the value-function class $\mathcal{F}_\Phi$ as $\mathcal{F}_\Phi := \{\phi(\cdot,\cdot)^\mathsf{T}\theta : \theta \in \mathbb{R}^d, \phi(\cdot,\cdot)^\mathsf{T}\theta \in [0, V_{\max}]\}$, and the policy class $\Pi_\Phi$ consists of the greedy policies of each value function in $\mathcal{F}_\Phi$.*

**Assumption 3** (Realizability and Completeness). *$\varepsilon_{\mathcal{F},\mathcal{F}} = \varepsilon_{\mathcal{F}} = 0$.*

Note that, when the feature mapping $\phi(\cdot,\cdot)$ is the one induced by the linear MDP [Jin et al., 2020], it automatically ensure that $\pi^\star \in \Pi_\Phi$ and $\mathcal{F}_\Phi$ satisfies Assumptions 3. In contrast, we highlight that the standard linear function approximation or linear MDP setup does *not* entail all the assumptions needed by Liu et al. [2020] as mentioned earlier.

Below is our main result in the linear function approximation setting.

**Theorem 3.2.** *Suppose the value-function class $\mathcal{F}$ is a linear function class that satisfies realizability and completeness (Definition 2 and Assumption 3) and $\widehat{\pi}$ is the output of Eq.(3.2) using value-function class $\mathcal{F}_\Phi$ and policy class $\Pi_\Phi$. If we choose $\varepsilon = cV_{\max}^2 d \log \frac{V_{\max}|\mathcal{A}|d}{\delta}/n$, then, for any policy $\pi : \mathcal{S} \to \Delta(\mathcal{A})$, we have*

$$J(\pi) - J(\widehat{\pi}) \le \mathcal{O}\left(\frac{V_{\max}}{1-\gamma}\sqrt{\frac{d\log\frac{V_{\max}|\mathcal{A}|d}{\delta}}{n}}\mathbb{E}_{d_\pi}\left[\sqrt{\phi(s,a)^\mathsf{T}\Sigma_\mathcal{D}^{-1}\phi(s,a)}\right]\right),$$

*where $c$ is an absolute constant, and $\Sigma_\mathcal{D} := \mathbb{E}_\mathcal{D}\left[\phi(s,a)\phi(s,a)^\mathsf{T}\right]$.*

The detailed proof of Theorem 3.2 is provided in Appendix B.2. Our guarantee is structurally very similar to that of Jin et al. [2021, Theorem 4.4], except that we only need linear function approximation with realizability and completeness assumptions, and they consider the finite-horizon linear MDP setting. If we translate their result to the discounted setting by setting $H = \mathcal{O}(1/(1-\gamma))$, we enjoy a net improvement of order $\mathcal{O}(d)$ in sample complexity when the action space is finite and small. To make it concrete, that is $\mathcal{O}(\sqrt{d^2\log(dn/\delta)/n})$ vs. $\mathcal{O}(\sqrt{d\log(d|\mathcal{A}|/\delta)/n})$ error bounds. The bound also shows that having a full-rank $\Sigma_\mathcal{D}$ (which is ensured by a full-rank covariance under $\mu$) is sufficient for consistent offline RL in linear function approximation. Crucially, the full-rank covariance is an easily checkable condition on data, as opposed to unverifiable concentrability assumptions. As a caveat, our results do not imply a computationally efficient algorithm, as a naïve implementation involves evaluating each policy pessimistically to pick the best. We discuss a computationally efficient adaptation of our approach in the next section.

# 4 Practical Algorithm — Regularized Offline Policy Optimization

A major challenge using the proposed algorithm in Section 3 in practice is that searching the policy with the best pessimistic evaluation over the policy space $\Pi$ is not computationally tractable. In this section, we present a practical algorithm that is computationally efficient assuming access to a (regularized) loss minimization oracle over the value function class $\mathcal{F}$, and also comes with rigorous theoretical guarantees.

Our practical algorithm is summarized in Algorithm 1. It has three key differences from the information-theoretic version in Eq.(3.2):

---

**Algorithm 1** PSPI: Pessimistic Soft Policy Iteration

---

**Input:** Batch data $\mathcal{D}$, regularization coefficient $\lambda$.

1: Initialize policy $\pi_1$ as the uniform policy.
2: **for** $t = 1, 2, \ldots, T$ **do**
3:   Obtain the pessimistic estimation for $\pi_t$ as $f_t$,

$$f_t \leftarrow \underset{f \in \mathcal{F}}{\arg\min} \left( f(s_0, \pi_t) + \lambda \mathcal{E}(f, \pi_t; \mathcal{D}) \right), \tag{4.1}$$

  where $\mathcal{E}(f, \pi_t; \mathcal{D})$ is defined in Eq.(3.1).
4:   Calculate $\pi_{t+1}$ by,

$$\pi_{t+1}(a|s) \propto \pi_t(a|s) \exp\left( \eta f_t(s, a) \right), \ \forall s, a \in \mathcal{S} \times \mathcal{A}.$$

5: **end for**
6: Output $\bar{\pi} := \mathsf{Unif}(\pi_{[1:T]})$.                 $\triangleright$ *uniformly mix $\pi_1, \ldots, \pi_T$ at the trajectory level*

---

1. The pessimistic policy evaluation is now performed via regularization (Line 3) instead of constrained optimization.
2. Instead of searching over an explicit policy space $\Pi$, we search over a policy class implicitly induced from $\mathcal{F}$ (defined in Eq.(4.2)) and therefore no longer have a policy class independent of $\mathcal{F}$ separately, which is a common practice in API-style algorithms [Munos, 2003; Antos et al., 2008; Lazaric et al., 2012].
3. We optimize the policy using mirror descent updates, which yields computationally tractable optimization over the implicit policy class. This property has been leveraged in many prior works, although typically in online RL settings [Even-Dar et al., 2009; Agarwal et al., 2019; Geist et al., 2019; Cai et al., 2020; Shani et al., 2020].

Note that the use of a specific policy class above can be relaxed if a stronger structural assumption is made on the MDP (e.g., linear MDPs [Jin et al., 2020, 2021]).

## 4.1 Analysis of Algorithm 1

We now provide the analysis of Algorithm 1 in this section. For ease of presentation, we formally define the implicit policy class for this section:

$$\Pi_{\mathrm{SPI}} := \{\pi'(\cdot|s) \propto \exp(\eta \sum_{i=1}^{t} f^{(t)}(s, \cdot)) : 1 \le t \le T, f^{(1)}, \ldots, f^{(i)} \in \mathcal{F}\}, \tag{4.2}$$

which is the natural policy class for soft policy-iteration approaches. The following theorem describes the performance guarantee of $\bar{\pi}$.

**Theorem 4.1.** *Let $\lambda = \sqrt[3]{V_{\max}/(1-\gamma)^2 \varepsilon_r^2}$ with $\varepsilon_r$ in Eq.(3.3), $\eta = \sqrt{\frac{\log |\mathcal{A}|}{2V_{\max}^2 T}}$, and $\bar{\pi}$ be obtained from Algorithm 1. For any policy $\pi : \mathcal{S} \to \Delta(\mathcal{A})$ we wish to compete with, suppose Assumptions 1 and 2 hold with respect to the policy class $\Pi_{SPI} \cup \{\pi\}$. Then, for any constant $C_2 \ge 1$, we have with probability at least $1 - \delta$,*

$$J(\pi) - J(\bar{\pi})$$

$$\le \underbrace{\mathcal{O}\left( \sqrt{C_2} \left( \frac{\sqrt{\varepsilon_{\mathcal{F},\mathcal{F}} + \varepsilon_{\mathcal{F}}}}{1-\gamma} + \frac{V_{\max}}{1-\gamma} \sqrt[3]{\frac{T \log \frac{|\mathcal{F}|}{\delta}}{n}} + \sqrt[3]{\frac{V_{\max}\varepsilon_{\mathcal{F}}}{(1-\gamma)^2}} \right) \right)}_{\mathrm{err_{on}}(\pi):\ \textit{on-support error}} + \underbrace{\mathcal{O}\left( \frac{V_{\max}}{1-\gamma} \sqrt{\frac{\log |\mathcal{A}|}{T}} \right)}_{\textit{optimization error}}$$

$$+ \underbrace{\frac{1}{T} \sum_{t=1}^{T} \left( \min_{\nu:\mathscr{C}(\nu;\mu,\mathcal{F},\pi_t) \le C_2} \left| \sum_{(s,a)\in\mathcal{S}\times\mathcal{A}} \frac{(d_\pi \setminus \nu)(s,a) \left[ f_t(s,a) - (\mathcal{T}^{\pi_t} f_t)(s,a) \right]}{1-\gamma} \right| \right)}_{\mathrm{err_{off}}(\pi):\ \textit{off-support error}},$$

*where $\mathscr{C}(\nu; \mu, \mathcal{F}, \pi_t)$ is defined in Definition 1, $(d_\pi \setminus \nu)(s,a) := \max(d_\pi(s,a) - \nu(s,a), 0)$.*

We provide a proof sketch of Theorem 4.1 at the end of this section, and defer the full proof to Appendix C. We now make a few remarks about the results in Theorem 4.1.

**Measurement of distribution shift effect.** Compared with the information-theoretical result (provided in Theorem 3.1), the measurement of distribution shift in Theorem 4.1 depends on the optimization trajectory. That is, it measures the distance between two distribution $\nu$ and $\mu$ by $\mathscr{C}(\nu; \mu, \mathcal{F}, \pi_t)$ ($\pi_{[1:T]}$ is the sequence of policies produced by the algorithm) whereas Theorem 3.1 uses $\mathscr{C}(\nu; \mu, \mathcal{F}, \pi)$ ($\pi$ is the baseline policy we compete with). We remark that both of these two measurements are weaker then traditional density-ratio definitions (e.g., [Munos and Szepesvári, 2008; Chen and Jiang, 2019; Xie and Jiang, 2020]) as we demonstrated before, as the dependence of $\mathscr{C}$ on $\pi$ is relatively secondary.

**Dependence on $T$.** The number of optimization rounds $T$ affects the bound in two opposite ways: as $T$ increases, the optimization error term decreases, whereas the second term of the on-support error increases. The latter increase is due to the complexity of the implicit policy class $\Pi$ growing exponentially with $T$, which affects our concentration bounds. To optimize the bound, the optimal choice is $T = \mathcal{O}(n^{2/5})$, leading to an overall $\mathcal{O}(n^{-1/5})$ rate. While such a rate is relatively slow, we remark that the complexity bound of $\Pi$ is conservative, and in certain cases it is possible to obtain much sharper bounds: for example, in linear function approximation (Section 3.1), $\Pi_{\mathrm{SPI}}$ are a priori captured by the space of softmax policies, whose complexity has no dependence on $T$ (up to mild logarithmic dependence due to norms). That is, the $\mathrm{err}_{\mathrm{on}}(\pi)$ term in Theorem 4.1 reduces to $\widetilde{\mathcal{O}}(\frac{V_{\max}}{1-\gamma} \sqrt[3]{d/n})$ ($\varepsilon_{\mathcal{F},\mathcal{F}} = \varepsilon_{\mathcal{F}} = 0$ in linear function approximation), and yields an overall $\mathcal{O}(n^{-1/3})$ rate.

**Bias-variance decomposition.** Similar to Theorem 3.1, Theorem 4.1 also allows arbitrary decomposition of the error bound into on-support and off-support components by setting the concentrability threshold $C_2$, which serves as a bias-variance tradeoff as before. In fact, the splitting can be done separately for each $\pi_t$ in $1 \leq t \leq T$ and we omit such flexibility for readability. The optimization error does not depend on the splitting. Our performance guarantee naturally adapts to the best possible decomposition as before. As in Theorem 3.1, if the estimation on the off-support region is "high-quality", we can further simplify the performance guarantees, but the requirement of "high-quality" is different from that of Corollary 1. We make it formal in the following corollary.

**Corollary 4** ("Double Robustness"). *For any $\pi$ and $C_2 \geq 0$, $\mathrm{err}_{\mathrm{off}}(\pi) = 0$ when either (1) $\mathscr{C}(d_\pi; \mu, \mathcal{F}, \pi_t) \leq C_2$ for all $t \in [T]$, or, (2) $f_t - \mathcal{T}^{\pi_t} \Delta f_t \equiv 0$ for all $t \in [T]$.*

We note that the conditions above depend on the optimization trajectory through their dependence on $\pi_t$, but can be made algorithm-independent by instead asserting the stronger requirement that $\mathscr{C}(d_\pi, \mu, \mathcal{F}, \pi') \leq C_2$ for all $\pi' \in \Pi_{\mathrm{SPI}}$ in the first condition.

**Competing with the optimal policy.** As before, we can provide a guarantee for competing with the optimal policy, under coverage assumptions weaker than the typical batch RL literature, albeit slightly stronger than those of Corollary 2. We state the formal result below.

**Corollary 5** (Competing with optimal policy). *Under conditions of Theorem 4.1, if $\mathscr{C}(d_{\pi^\star}; \mu, \mathcal{F}, \pi) \leq C_2$ for all $\pi \in \Pi_{SPI}$, we have*

$$J(\pi^\star) - J(\widehat{\pi}) \leq \mathcal{O}\left( \frac{V_{\max}\sqrt{C_2}}{1-\gamma} \left( \frac{\log \frac{|\mathcal{F}|}{\delta} \log |\mathcal{A}|}{n} \right)^{1/5} + \frac{\sqrt{C_2(\varepsilon_{\mathcal{F},\mathcal{F}} + \varepsilon_{\mathcal{F}})}}{1-\gamma} \right).$$

Note that the conditions of Corollary 5 are satisfied as before whenever $\|d_{\pi^\star}/\mu\|_\infty \leq C_2$.

**Computationally-efficient implementation with linear function approximation** We remark that our algorithm is computationally efficient when the value-function class $\mathcal{F}$ is linear, that is, $\mathcal{F} := \{\phi(\cdot, \cdot)^\mathsf{T} \theta : \theta \in \mathbb{R}^d\}$. In this case, the objective of Eq.(4.1) has a closed-form expression which is quadratic in $\theta$. In addition, under additional matrix invertibility conditions, Eq.(4.1) has a closed-form solution which generalizes LSTDQ [Lagoudakis and Parr, 2003; Sutton et al., 2009; Dann et al., 2014]. A similar connection has been made by Antos et al. [2008], but our derivation is more general. See Appendix Appendix D for further details.

We conclude the section with a proof sketch showing the key insights used in establishing the proof.

**Proof sketch of Theorem 4.1.** Our proof constructs a corresponding MDP $\mathcal{M}_t$ for every $f_t, \pi_t$ pair. Each $\mathcal{M}_t$ has the same dynamics as the ground-truth MDP, but chooses a different reward function, such that $f_t$ is the $Q$-function of $\pi_t$ in $\mathcal{M}_t$, $Q^{\pi_t}_{\mathcal{M}_t}$ (we use the subscript of $\mathcal{M}_t$ to denote the corresponding value or operator in MDP $\mathcal{M}_t$). Our proof relies on some key properties of $\mathcal{M}_t$, such as $Q^\pi - \mathcal{T}^\pi_{\mathcal{M}_t} Q^\pi = f_t - \mathcal{T}^{\pi_t} f_t$. We decompose $J(\pi) - J(\bar{\pi})$ as follows.

$$
J(\pi) - J(\bar{\pi}) \le \underbrace{\frac{1}{T}\sum_{t=1}^{T}\left(J_{\mathcal{M}_t}(\pi) - J_{\mathcal{M}_t}(\pi_t)\right)}_{\text{optimization error}} + \underbrace{\frac{1}{T}\sum_{t=1}^{T}\left(J(\pi) - J_{\mathcal{M}_t}(\pi)\right)}_{\text{controlled by } \|Q^\pi - \mathcal{T}^\pi_{\mathcal{M}_t} Q^\pi\|_{2,d_\pi} = \|f_t - \mathcal{T}^{\pi_t} f_t\|_{2,d_\pi}}
$$
$$
+ \text{ approximation/statistical errors.}
$$

The proof is completed by bounding $\|f_t - \mathcal{T}^{\pi_t} f_t\|_{2,d_\pi}$ on both on-support and off-support regions. $\square$

## 5 Conclusions

This paper investigates sample-efficient offline reinforcement learning without data coverage assumptions (e.g., concentrability). To achieve that goal, our paper contributes several crucial improvements to the literature. We introduce the concept of Bellman-consistent pessimism. It enables the sample-efficient guarantees with only the Bellman-completeness assumption which is standard in the exploratory setting, whereas the point-wise/bonus-based pessimism popularly adopted in the literature usually requires stronger and/or extra assumptions. Algorithmically, we demonstrate how to implicitly infer a policy value lower bound through a version space and provide a tractable implementation. A particularly important aspect of our results is the ability to adapt to the best bias-variance tradeoff in the hindsight, which no prior algorithms achieve to the best of our knowledge. When applying our results in linear function approximation, we attain an $\mathcal{O}(d)$ improvement in sample complexity, compared with the best-known recent work of offline RL in linear MDPs, whenever the action space is finite and small.

As of limitations and future work, the sample complexity of our practical algorithm is worse than that of the information-theoretic approach, and it will be interesting to close this gap. Another future direction is to empirically evaluate PSPI on benchmarks and compare it to existing approaches.

## Acknowledgment

Part of this work was carried out while TX and AA worked at Microsoft Research. NJ acknowledges funding support from the ARL Cooperative Agreement W911NF-17-2-0196, NSF IIS-2112471, and Adobe Data Science Research Award.

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
