# Appendix

## A    Estimating Mean-Squared Bellman Error

We provide theoretical properties of $\mathcal{E}(f, \pi; \mathcal{D})$ (defined in Eq.(3.1)), when using it to bound Bellman error. Over this section, we use $\mu \times (\mathcal{P}, R)$ to denote the joint distribution of $(s, a, r, s')$.

All of our results in the appendix strongly depends one following two constants

1. $\varepsilon_r$ — For any $\pi \in \Pi$, if $f_\pi$ is the "best estimation" of $Q^\pi$ in $\mathcal{F}$ (formal definition of $f_\pi$: Eq.(A.1) for general function approximation, $Q^\pi$ for Linear function approximation), then $\mathcal{E}(f_\pi, \pi; \mathcal{D}) \le \varepsilon_r$.
2. $\varepsilon_b$ — For any function $f \in \mathcal{F}$ and any $\pi \in \Pi$, if $\mathcal{E}(f, \pi; \mathcal{D}) \le \varepsilon_r$, then $\|f - \mathcal{T}^\pi f\|_{2,\mu}^2 \le \varepsilon_b$.

The detailed discussion about $\varepsilon_r$ and $\varepsilon_b$ is provided in Appendix A.1 (general function approximation) and and Appendix A.2 (linear Linear function approximation).

### A.1    Results for General Function Approximation

This section summarizes the results regarding $\mathcal{E}(f, \pi; \mathcal{D})$, we defer the full proof of Theorem A.1 and Theorem A.2 to Appendix A.1.2.

**Theorem A.1.** *For any $\pi \in \Pi$, let $f_\pi$ be defined as follows,*

$$f_\pi := \underset{f \in \mathcal{F}}{\operatorname{argmin}} \ \underset{admissible \ \nu}{\sup} \ \|f - \mathcal{T}^\pi f\|_{2,\nu}^2 . \tag{A.1}$$

*Then, for $\mathcal{E}(f_\pi, \pi; \mathcal{D})$ (defined in Eq.(3.1)), we have*

$$\mathcal{E}(f_\pi, \pi; \mathcal{D}) \le \frac{139 V_{\max}^2 \log \frac{|\mathcal{F}||\Pi|}{\delta}}{n} + 39 \varepsilon_\mathcal{F} =: \varepsilon_r. \tag{A.2}$$

We now show that $\mathcal{E}(f, \pi; \mathcal{D})$ could effectively estimate $\|f - \mathcal{T}^\pi f\|_{2,\mu}^2$.

**Theorem A.2.** *For any $\pi \in \Pi$, $f \in \mathcal{F}$, and any $\varepsilon > 0$, if $\mathcal{E}(f, \pi; \mathcal{D}) \le \varepsilon$, then,*

$$\|f - \mathcal{T}^\pi f\|_{2,\mu} \le V_{\max} \sqrt{\frac{231 \log \frac{|\mathcal{F}||\Pi|}{\delta}}{n}} + \sqrt{\varepsilon_{\mathcal{F},\mathcal{F}}} + \sqrt{\varepsilon_{\mathcal{F},\mathcal{F}} + \varepsilon}. \tag{A.3}$$

We also define $\sqrt{\varepsilon_b}$ when setting $\varepsilon = \varepsilon_r$ in Eq.(A.3), i.e.,

$$\sqrt{\varepsilon_b} := V_{\max} \sqrt{\frac{231 \log \frac{|\mathcal{F}||\Pi|}{\delta}}{n}} + \sqrt{\varepsilon_{\mathcal{F},\mathcal{F}}} + \sqrt{\varepsilon_{\mathcal{F},\mathcal{F}} + \varepsilon_r}. \tag{A.4}$$

#### A.1.1    Complementary Lemmas for General Function Approximation

We first provide some complementary lemmas that used in our detailed proofs of Theorem A.1 and Theorem A.2.

**Lemma A.3.** *For any $f_1, f_2 \in \mathcal{F}$ and $\pi \in \Pi$, w.p. $1 - \delta$,*

$$\left| \|f_1(s,a) - (\mathcal{T}^\pi f_2)(s,a)\|_{2,\mu} - \|f_1(s,a) - (\mathcal{T}^\pi f_2)(s,a)\|_{2,\mathcal{D}} \right| \le \sqrt{\frac{5 V_{\max}^2 \log \frac{|\mathcal{F}||\Pi|}{\delta}}{n}}.$$

***Proof of Lemma A.3.*** The proof of this lemma follows the similar proving strategy of [Xie and Jiang, 2021, Lemma 10]. We first apply Bernstein's inequality with a union bound over $\mathcal{F} \times \mathcal{F}$, and obtain: w.p. $1 - \delta$, for any $f_1, f_2 \in \mathcal{F}$,

$$\left| \|f_1(s,a) - (\mathcal{T}^\pi f_2)(s,a)\|_{2,\mu}^2 - \|f_1(s,a) - (\mathcal{T}^\pi f_2)(s,a)\|_{2,\mathcal{D}}^2 \right|$$

$$= \left| \mathbb{E}_{\mu \times (\mathcal{P},R)} \left[ (f_1(s,a) - (\mathcal{T}^\pi f_2)(s,a))^2 \right] - \frac{1}{n} \sum_{(s,a,r,s') \in \mathcal{D}} (f_1(s,a) - (\mathcal{T}^\pi f_2)(s,a))^2 \right|$$

$$\leq \sqrt{\frac{4\mathbb{V}_\mu\left[(f_1(s,a) - (\mathcal{T}^\pi f_2)(s,a))^2\right]\log\frac{|\mathcal{F}||\Pi|}{\delta}}{n}} + \frac{2V_{\max}^2\log\frac{|\mathcal{F}||\Pi|}{\delta}}{3n}$$

$$\leq \|f_1(s,a) - (\mathcal{T}^\pi f_2)(s,a)\|_{2,\mu}\sqrt{\frac{4V_{\max}^2\log\frac{|\mathcal{F}||\Pi|}{\delta}}{n}} + \frac{2V_{\max}^2\log\frac{|\mathcal{F}||\Pi|}{\delta}}{3n}, \tag{A.5}$$

where the last inequality is obtained by the following argument,

$$\mathbb{V}_{\mu\times(\mathcal{P},R)}\left[(f_1(s,a) - (\mathcal{T}^\pi f_2)(s,a))^2\right]$$

$$\leq \mathbb{E}_{\mu\times(\mathcal{P},R)}\left[(f_1(s,a) - (\mathcal{T}^\pi f_2)(s,a))^4\right]$$

$$\leq V_{\max}^2\mathbb{E}_{\mu\times(\mathcal{P},R)}\left[(f_1(s,a) - (\mathcal{T}^\pi f_2)(s,a))^2\right],$$

where the last inequality follows from $|f_1(s,a) - (\mathcal{T}^\pi f_2)(s,a)| \leq V_{\max}, \forall(s,a,r,s')$.
By the fact of $|a-b|^2 \leq |a^2 - b^2|$, we know

$$\left|\|f_1(s,a) - (\mathcal{T}^\pi f_2)(s,a)\|_{2,\mu} - \|f_1(s,a) - (\mathcal{T}^\pi f_2)(s,a)\|_{2,\mathcal{D}}\right|$$

$$\leq \sqrt{\left|\|f_1(s,a) - (\mathcal{T}^\pi f_2)(s,a)\|_{2,\mu}^2 - \|f_1(s,a) - (\mathcal{T}^\pi f_2)(s,a)\|_{2,\mathcal{D}}^2\right|}$$

$$\leq \sqrt{\|f_1(s,a) - (\mathcal{T}^\pi f_2)(s,a)\|_{2,\mu}\sqrt{\frac{4V_{\max}^2\log\frac{|\mathcal{F}||\Pi|}{\delta}}{n}} + \frac{2V_{\max}^2\log\frac{|\mathcal{F}||\Pi|}{\delta}}{3n}} \qquad \text{(by Eq.(A.5))}$$

$$\leq \sqrt{\|f_1(s,a) - (\mathcal{T}^\pi f_2)(s,a)\|_{2,\mu}}\sqrt[4]{\frac{4V_{\max}^2\log\frac{|\mathcal{F}||\Pi|}{\delta}}{n}} + \sqrt{\frac{2V_{\max}^2\log\frac{|\mathcal{F}||\Pi|}{\delta}}{3n}}. \tag{A.6}$$

On the other hand,

$$\left|\|f_1(s,a) - (\mathcal{T}^\pi f_2)(s,a)\|_{2,\mu} - \|f_1(s,a) - (\mathcal{T}^\pi f_2)(s,a)\|_{2,\mathcal{D}}\right|$$

$$\leq \frac{\left|\|f_1(s,a) - (\mathcal{T}^\pi f_2)(s,a)\|_{2,\mu}^2 - \|f_1(s,a) - (\mathcal{T}^\pi f_2)(s,a)\|_{2,\mathcal{D}}^2\right|}{\|f_1(s,a) - (\mathcal{T}^\pi f_2)(s,a)\|_{2,\mu}}$$

$$\leq \frac{\|f_1(s,a) - (\mathcal{T}^\pi f_2)(s,a)\|_{2,\mu}\sqrt{\frac{4V_{\max}^2\log\frac{|\mathcal{F}||\Pi|}{\delta}}{n}} + \frac{2V_{\max}^2\log\frac{|\mathcal{F}||\Pi|}{\delta}}{3n}}{\|f_1(s,a) - (\mathcal{T}^\pi f_2)(s,a)\|_{2,\mu}} \qquad \text{(by Eq.(A.5))}$$

$$\leq \sqrt{\frac{4V_{\max}^2\log\frac{|\mathcal{F}||\Pi|}{\delta}}{n}} + \frac{2V_{\max}^2\log\frac{|\mathcal{F}||\Pi|}{\delta}}{3n\|f_1(s,a) - (\mathcal{T}^\pi f_2)(s,a)\|_{2,\mu}}. \tag{A.7}$$

Combining Eq.(A.6) and Eq.(A.7), we obtain

$$\left|\|f_1(s,a) - (\mathcal{T}^\pi f_2)(s,a)\|_{2,\mu} - \|f_1(s,a) - (\mathcal{T}^\pi f_2)(s,a)\|_{2,\mathcal{D}}\right|$$

$$\leq \min\left(\sqrt{\underbrace{\|f_1(s,a) - (\mathcal{T}^\pi f_2)(s,a)\|_{2,\mu}}_{=:g\in\mathbb{R}}}\sqrt[4]{\frac{4V_{\max}^2\log\frac{|\mathcal{F}||\Pi|}{\delta}}{n}} + \sqrt{\frac{2V_{\max}^2\log\frac{|\mathcal{F}||\Pi|}{\delta}}{3n}},\right.$$

$$\left.\sqrt{\frac{4V_{\max}^2\log\frac{|\mathcal{F}||\Pi|}{\delta}}{n}} + \frac{2V_{\max}^2\log\frac{|\mathcal{F}||\Pi|}{\delta}}{3n\|f_1(s,a) - (\mathcal{T}^\pi f_2)(s,a)\|_{2,\mu}}\right)$$

$$\leq \max_{g\in\mathbb{R}}\min\left(\sqrt{g}\sqrt[4]{\frac{4V_{\max}^2\log\frac{|\mathcal{F}||\Pi|}{\delta}}{n}} + \sqrt{\frac{2V_{\max}^2\log\frac{|\mathcal{F}||\Pi|}{\delta}}{3n}}, \sqrt{\frac{4V_{\max}^2\log\frac{|\mathcal{F}||\Pi|}{\delta}}{n}} + \frac{2V_{\max}^2\log\frac{|\mathcal{F}||\Pi|}{\delta}}{3ng}\right)$$

$$\leq \min_{g\in\mathbb{R}} \max\left(\sqrt{g}\sqrt[4]{\frac{4V_{\max}^2\log\frac{|\mathcal{F}||\Pi|}{\delta}}{n}} + \sqrt{\frac{2V_{\max}^2\log\frac{|\mathcal{F}||\Pi|}{\delta}}{3n}}, \sqrt{\frac{4V_{\max}^2\log\frac{|\mathcal{F}||\Pi|}{\delta}}{n}} + \frac{2V_{\max}^2\log\frac{|\mathcal{F}||\Pi|}{\delta}}{3ng}\right)$$

$$\overset{(a)}{\leq} \max\left(\sqrt{\frac{4V_{\max}^2\log\frac{|\mathcal{F}||\Pi|}{\delta}}{n}} + \sqrt{\frac{2V_{\max}^2\log\frac{|\mathcal{F}||\Pi|}{\delta}}{3n}}, \sqrt{\frac{4V_{\max}^2\log\frac{|\mathcal{F}||\Pi|}{\delta}}{n}} + \sqrt{\frac{V_{\max}^2\log\frac{|\mathcal{F}||\Pi|}{\delta}}{9n}}\right)$$

$$\leq \sqrt{\frac{5V_{\max}^2\log\frac{|\mathcal{F}||\Pi|}{\delta}}{n}},$$

where (a) is by selecting $g = \sqrt{4V_{\max}^2\log\frac{|\mathcal{F}||\Pi|}{\delta}/n}$. This completes the proof. $\qquad\square$

**Lemma A.4.** *For any $f, g_1, g_2 \in \mathcal{F}$ and $\pi \in \Pi$, we have*

$$\left|\|g_1 - \mathcal{T}^\pi f\|_{2,\mu}^2 - \|g_2 - \mathcal{T}^\pi f\|_{2,\mu}^2 \right.$$

$$\left. - \frac{1}{n}\sum_{(s,a,r,s')\in\mathcal{D}}(g_1(s,a) - r - \gamma f(s',\pi))^2 + \frac{1}{n}\sum_{(s,a,r,s')\in\mathcal{D}}(g_2(s,a) - r - \gamma f(s',\pi))^2\right|$$

$$\leq 4V_{\max}\|g_1 - g_2\|_{2,\mu}\sqrt{\frac{\log\frac{|\mathcal{F}||\Pi|}{\delta}}{n}} + \frac{2V_{\max}^2\log\frac{|\mathcal{F}||\Pi|}{\delta}}{3n}.$$

***Proof of Lemma A.4.*** By a standard calculation,

$$\frac{1}{n}\sum_{(s,a,r,s')\in\mathcal{D}}(g_1(s,a) - r - \gamma f(s',\pi))^2 - \frac{1}{n}\sum_{(s,a,r,s')\in\mathcal{D}}(g_2(s,a) - r - \gamma f(s',\pi))^2$$

$$= \frac{1}{n}\sum_{(s,a,r,s')\in\mathcal{D}}\left((g_1(s,a) - r - \gamma f(s',\pi))^2 - (g_2(s,a) - r - \gamma f(s',\pi))^2\right)$$

$$= \frac{1}{n}\sum_{(s,a,r,s')\in\mathcal{D}}\left((g_1(s,a) - g_2(s,a))(g_1(s,a) + g_2(s,a) - 2r - 2\gamma f(s',\pi))\right). \tag{A.8}$$

Similarly, we also have the following fact,

$$\mathbb{E}_{\mu\times(\mathcal{P},R)}\left[(g_1(s,a) - r - \gamma f(s',\pi))^2\right] - \mathbb{E}_{\mu\times(\mathcal{P},R)}\left[(g_2(s,a) - r - \gamma f(s',\pi))^2\right]$$

$$\overset{(a)}{=} \mathbb{E}_{\mu\times(\mathcal{P},R)}\left[(g_1(s,a) - g_2(s,a))(g_1(s,a) + g_2(s,a) - 2r - 2\gamma f(s',\pi))\right]$$

$$= \mathbb{E}_\mu\left[\mathbb{E}\left[(g_1(s,a) - g_2(s,a))(g_1(s,a) + g_2(s,a) - 2r - 2\gamma f(s',\pi))|s,a\right]\right]$$

$$= \mathbb{E}_\mu\left[(g_1(s,a) - g_2(s,a))(g_1(s,a) + g_2(s,a) - 2(\mathcal{T}^\pi f)(s,a))\right] \tag{A.9}$$

$$\overset{(b)}{=} \mathbb{E}_\mu\left[(g_1(s,a) - (\mathcal{T}^\pi f)(s,a))^2\right] - \mathbb{E}_\mu\left[(g_2(s,a) - (\mathcal{T}^\pi f)(s,a))^2\right], \tag{A.10}$$

where (a) and (b) follow from the similar argument to Eq.(A.8).

By using Eq.(A.8) and Eq.(A.10), we know

$$\mathbb{E}_{\mu\times(\mathcal{P},R)}\left[\frac{1}{n}\sum_{(s,a,r,s')\in\mathcal{D}}(g_1(s,a) - r - \gamma f(s',\pi))^2 - \frac{1}{n}\sum_{(s,a,r,s')\in\mathcal{D}}(g_2(s,a) - r - \gamma f(s',\pi))^2\right]$$

$$= \mathbb{E}_\mu\left[(g_1(s,a) - (\mathcal{T}^\pi f)(s,a))^2\right] - \mathbb{E}_\mu\left[(g_2(s,a) - (\mathcal{T}^\pi f)(s,a))^2\right].$$

Then,

$$\left|\mathbb{E}_\mu\left[(g_1(s,a) - (\mathcal{T}^\pi f)(s,a))^2\right] - \mathbb{E}_\mu\left[(g_2(s,a) - (\mathcal{T}^\pi f)(s,a))^2\right] - \right.$$

$$\frac{1}{n} \sum_{(s,a,r,s') \in \mathcal{D}} (g_1(s,a) - r - \gamma f(s', \pi))^2 + \frac{1}{n} \sum_{(s,a,r,s') \in \mathcal{D}} (g_2(s,a) - r - \gamma f(s', \pi))^2 \Bigg|$$

$$= \left| \mathbb{E}_\mu \left[ (g_1(s,a) - (\mathcal{T}^\pi f)(s,a))^2 \right] - \mathbb{E}_\mu \left[ (g_2(s,a) - (\mathcal{T}^\pi f)(s,a))^2 \right] - \right.$$

$$\left. \frac{1}{n} \sum_{(s,a,r,s') \in \mathcal{D}} ((g_1(s,a) - g_2(s,a)) (g_1(s,a) + g_2(s,a) - 2r - 2\gamma f(s', \pi))) \right|$$

$$\leq \sqrt{\frac{4 \mathbb{V}_{\mu \times (\mathcal{P}, R)} [(g_1(s,a) - g_2(s,a)) (g_1(s,a) + g_2(s,a) - 2r - 2\gamma f(s', \pi))] \log \frac{|\mathcal{F}||\Pi|}{\delta}}{n}} \quad \text{(A.11)}$$

$$+ \frac{2 V_{\max}^2 \log \frac{|\mathcal{F}||\Pi|}{\delta}}{3n},$$

where the first equation follows from Eq.(A.9) and the last inequality follows from the Bernsiten's inequality.

We now study the term $\mathbb{V}_{\mu \times (\mathcal{P}, R)} [(g_1(s,a) - g_2(s,a)) (g_1(s,a) + g_2(s,a) - 2r - 2\gamma f(s', \pi))]$,

$$\mathbb{V}_{\mu \times (\mathcal{P}, R)} [(g_1(s,a) - g_2(s,a)) (g_1(s,a) + g_2(s,a) - 2r - 2\gamma f(s', \pi))]$$

$$\leq \mathbb{E}_{\mu \times (\mathcal{P}, R)} \left[ (g_1(s,a) - g_2(s,a))^2 (g_1(s,a) + g_2(s,a) - 2r - 2\gamma f(s', \pi))^2 \right]$$

$$\leq 4 V_{\max}^2 \mathbb{E}_\mu \left[ (g_1(s,a) - g_2(s,a))^2 \right]$$

where the last inequality follows from the fact of $|g_1(s,a) + g_2(s,a) - 2r - 2\gamma f(s', \pi)| \leq 2V_{\max}$. This completes the proof. $\qquad \square$

**Lemma A.5.** *For any $\pi \in \Pi$, let $f_\pi$ and $g$ be defined as follows,*

$$f_\pi := \operatorname*{argmin}_{f \in \mathcal{F}} \sup_{admissible\ \nu} \|f - \mathcal{T}^\pi f\|_{2,\nu}^2$$

$$g := \operatorname*{argmin}_{g \in \mathcal{F}} \frac{1}{n} \sum_{(s,a,r,s') \in \mathcal{D}} (g(s,a) - r - \gamma f_\pi(s', \pi))^2.$$

*Then, we have*

$$\|f_\pi - g\|_{2,\mu} \leq 9 V_{\max} \sqrt{\frac{\log \frac{|\mathcal{F}||\Pi|}{\delta}}{n}} + 2\sqrt{\varepsilon_\mathcal{F}}.$$

***Proof of Lemma A.5.*** By applying Lemma A.4, we have

$$\left| \mathbb{E}_\mu \left[ (f_\pi(s,a) - (\mathcal{T}^\pi f_\pi)(s,a))^2 \right] - \mathbb{E}_\mu \left[ (g(s,a) - (\mathcal{T}^\pi f_\pi)(s,a))^2 \right] - \right.$$

$$\left. \frac{1}{n} \sum_{(s,a,r,s') \in \mathcal{D}} (f_\pi(s,a) - r - \gamma f_\pi(s', \pi))^2 + \frac{1}{n} \sum_{(s,a,r,s') \in \mathcal{D}} (g(s,a) - r - \gamma f_\pi(s', \pi))^2 \right|$$

$$\leq 4 V_{\max} \sqrt{\frac{\mathbb{E}_\mu \left[ (f_\pi(s,a) - g(s,a))^2 \right] \log \frac{|\mathcal{F}||\Pi|}{\delta}}{n}} + \frac{2 V_{\max}^2 \log \frac{|\mathcal{F}||\Pi|}{\delta}}{3n}.$$

In addition,

$$\mathbb{E}_\mu \left[ (f_\pi(s,a) - g(s,a))^2 \right]$$

$$= \|f_\pi - g\|_{2,\mu}^2$$

$$\leq 2\|f_\pi - \mathcal{T}^\pi f_\pi\|_{2,\mu}^2 + 2\|g - \mathcal{T}^\pi f_\pi\|_{2,\mu}^2$$

$$= 2\|g - \mathcal{T}^\pi f_\pi\|_{2,\mu}^2 - 2\|f_\pi - \mathcal{T}^\pi f_\pi\|_{2,\mu}^2 + 4\|f_\pi - \mathcal{T}^\pi f_\pi\|_{2,\mu}^2$$

$$\overset{(a)}{\le} 2\|g - \mathcal{T}^\pi f_\pi\|_{2,\mu}^2 - 2\|f_\pi - \mathcal{T}^\pi f_\pi\|_{2,\mu}^2 + 4\varepsilon_{\mathcal{F}}$$

$$\overset{(b)}{\le} 8V_{\max}\sqrt{\frac{\mathbb{E}_\mu\left[(f_\pi(s,a) - g(s,a))^2\right]\log\frac{|\mathcal{F}||\Pi|}{\delta}}{n}} + \frac{4V_{\max}^2\log\frac{|\mathcal{F}||\Pi|}{\delta}}{3n} + 4\varepsilon_{\mathcal{F}}, \quad \text{(A.12)}$$

where (a) follows from the fact of $\|f_\pi - \mathcal{T}^\pi f_\pi\|_{2,\mu}^2 \le \varepsilon_{\mathcal{F}}$ by Assumption 1, and (b) is obtained by the following argument

$$\|g - \mathcal{T}^\pi f_\pi\|_{2,\mu}^2 - \|f_\pi - \mathcal{T}^\pi f_\pi\|_{2,\mu}^2$$

$$\le \frac{1}{n}\sum_{(s,a,r,s')\in\mathcal{D}}(g(s,a) - r - \gamma f_\pi(s',\pi))^2 - \frac{1}{n}\sum_{(s,a,r,s')\in\mathcal{D}}(f_\pi(s,a) - r - \gamma f_\pi(s',\pi))^2$$

$$+ 4V_{\max}\sqrt{\frac{\mathbb{E}_\mu\left[(f_\pi(s,a) - g(s,a))^2\right]\log\frac{|\mathcal{F}||\Pi|}{\delta}}{n}} + \frac{2V_{\max}^2\log\frac{|\mathcal{F}||\Pi|}{\delta}}{3n} \qquad \text{(by Eq.(A.11))}$$

$$\le 4V_{\max}\sqrt{\frac{\mathbb{E}_\mu\left[(f_\pi(s,a) - g(s,a))^2\right]\log\frac{|\mathcal{F}||\Pi|}{\delta}}{n}} + \frac{2V_{\max}^2\log\frac{|\mathcal{F}||\Pi|}{\delta}}{3n} \qquad \text{(by the optimality of } g)$$

By solving Eq.(A.12), we obtain

$$\sqrt{\mathbb{E}_\mu\left[(f_\pi(s,a) - g(s,a))^2\right]}$$

$$\le 4V_{\max}\sqrt{\frac{\log\frac{|\mathcal{F}||\Pi|}{\delta}}{n}} + 2\sqrt{\frac{5V_{\max}^2\log\frac{|\mathcal{F}||\Pi|}{\delta}}{n}} + \varepsilon_{\mathcal{F}}$$

$$\le 4V_{\max}\sqrt{\frac{\log\frac{|\mathcal{F}||\Pi|}{\delta}}{n}} + 2V_{\max}\sqrt{\frac{5\log\frac{|\mathcal{F}||\Pi|}{\delta}}{n}} + 2\sqrt{\varepsilon_{\mathcal{F}}}$$

$$= 9V_{\max}\sqrt{\frac{\log\frac{|\mathcal{F}||\Pi|}{\delta}}{n}} + 2\sqrt{\varepsilon_{\mathcal{F}}}.$$

This completes the proof. $\qquad\square$

### A.1.2 Detailed Proofs of Theorem A.1 and Theorem A.2

***Proof of Theorem A.1.*** Let $g$ be defined as

$$g := \underset{g\in\mathcal{F}}{\arg\min}\frac{1}{n}\sum_{(s,a,r,s')\in\mathcal{D}}(g(s,a) - r - \gamma f_\pi(s',\pi))^2.$$

By applying Lemma A.4,

$$\left|\|f_\pi - \mathcal{T}^\pi f_\pi\|_{2,\mu}^2 - \|g - \mathcal{T}^\pi f_\pi\|_{2,\mu}^2 - \right.$$

$$\left.\frac{1}{n}\sum_{(s,a,r,s')\in\mathcal{D}}(f_\pi(s,a) - r - \gamma f_\pi(s',\pi))^2 + \frac{1}{n}\sum_{(s,a,r,s')\in\mathcal{D}}(g(s,a) - r - \gamma f_\pi(s',\pi))^2\right|$$

$$\le 4V_{\max}\|f_\pi - g\|_{2,\mu}\sqrt{\frac{\log\frac{|\mathcal{F}||\Pi|}{\delta}}{n}} + \frac{2V_{\max}^2\log\frac{|\mathcal{F}||\Pi|}{\delta}}{3n}.$$

By Lemma A.5, we know $\|f_\pi - g\|_{2,\mu} \le 9V_{\max}\sqrt{\frac{\log\frac{|\mathcal{F}||\Pi|}{\delta}}{n}} + 2\sqrt{\varepsilon_{\mathcal{F}}}$. Thus,

$$\left|\|f_\pi - \mathcal{T}^\pi f_\pi\|_{2,\mu}^2 - \|g - \mathcal{T}^\pi f_\pi\|_{2,\mu}^2 - \right.$$

$$\frac{1}{n}\sum_{(s,a,r,s')\in\mathcal{D}}\left(f_\pi(s,a)-r-\gamma f_\pi(s',\pi)\right)^2+\frac{1}{n}\sum_{(s,a,r,s')\in\mathcal{D}}\left(g(s,a)-r-\gamma f_\pi(s',\pi)\right)^2\Bigg|$$

$$\leq 4V_{\max}\left(9V_{\max}\sqrt{\frac{\log\frac{|\mathcal{F}||\Pi|}{\delta}}{n}}+2\sqrt{\varepsilon_\mathcal{F}}\right)\sqrt{\frac{\log\frac{|\mathcal{F}||\Pi|}{\delta}}{n}}+\frac{2V_{\max}^2\log\frac{|\mathcal{F}||\Pi|}{\delta}}{3n}$$

$$=8V_{\max}\sqrt{\frac{\log\frac{|\mathcal{F}||\Pi|}{\delta}}{n}\varepsilon_\mathcal{F}}+\frac{37V_{\max}^2\log\frac{|\mathcal{F}||\Pi|}{\delta}}{n}. \tag{A.13}$$

We now bound $\|f_\pi-\mathcal{T}^\pi f_\pi\|_{2,\mu}^2-\|g-\mathcal{T}^\pi f_\pi\|_{2,\mu}^2$,

$$\|f_\pi-\mathcal{T}^\pi f_\pi\|_{2,\mu}^2-\|g-\mathcal{T}^\pi f_\pi\|_{2,\mu}^2$$

$$\leq\left(\|f_\pi-\mathcal{T}^\pi f_\pi\|_{2,\mu}+\|g-\mathcal{T}^\pi f_\pi\|_{2,\mu}\right)\left|\|f_\pi-\mathcal{T}^\pi f_\pi\|_{2,\mu}-\|g-\mathcal{T}^\pi f_\pi\|_{2,\mu}\right|$$

$$\overset{(a)}{\leq}\left(2\|f_\pi-\mathcal{T}^\pi f_\pi\|_{2,\mu}+\|f_\pi-g\|_{2,\mu}\right)\|f_\pi-g\|_{2,\mu}$$

$$\overset{(b)}{\leq}\left(4\sqrt{\varepsilon_\mathcal{F}}+9V_{\max}\sqrt{\frac{\log\frac{|\mathcal{F}||\Pi|}{\delta}}{n}}\right)\left(9V_{\max}\sqrt{\frac{\log\frac{|\mathcal{F}||\Pi|}{\delta}}{n}}+2\sqrt{\varepsilon_\mathcal{F}}\right)$$

$$=54V_{\max}\sqrt{\frac{\log\frac{|\mathcal{F}||\Pi|}{\delta}}{n}\varepsilon_\mathcal{F}}+81V_{\max}^2\frac{\log\frac{|\mathcal{F}||\Pi|}{\delta}}{n}+8\varepsilon_\mathcal{F}, \tag{A.14}$$

where (a) follows from triangle inequality, and (b) is obtained by Lemma A.5 and the fact of $\|f_\pi-\mathcal{T}^\pi f_\pi\|_{2,\mu}^2\leq\varepsilon_\mathcal{F}$ (Assumption 1).

Combining Eq.(A.13) and Eq.(A.14), we obtain

$$\frac{1}{n}\sum_{(s,a,r,s')\in\mathcal{D}}\left(f_\pi(s,a)-r-\gamma f_\pi(s',\pi)\right)^2-\frac{1}{n}\sum_{(s,a,r,s')\in\mathcal{D}}\left(g(s,a)-r-\gamma f_\pi(s',\pi)\right)^2$$

$$\leq 8V_{\max}\sqrt{\frac{\log\frac{|\mathcal{F}||\Pi|}{\delta}}{n}\varepsilon_\mathcal{F}}+\frac{37V_{\max}^2\log\frac{|\mathcal{F}||\Pi|}{\delta}}{n}+54V_{\max}\sqrt{\frac{\log\frac{|\mathcal{F}||\Pi|}{\delta}}{n}\varepsilon_\mathcal{F}}+81V_{\max}^2\frac{\log\frac{|\mathcal{F}||\Pi|}{\delta}}{n}+8\varepsilon_\mathcal{F}$$

$$\leq 62V_{\max}\sqrt{\frac{\log\frac{|\mathcal{F}||\Pi|}{\delta}}{n}\varepsilon_\mathcal{F}}+\frac{118V_{\max}^2\log\frac{|\mathcal{F}||\Pi|}{\delta}}{n}+8\varepsilon_\mathcal{F}$$

$$\leq\frac{139V_{\max}^2\log\frac{|\mathcal{F}||\Pi|}{\delta}}{n}+39\varepsilon_\mathcal{F},$$

where the last inequality follows from Cauchy–Schwarz inequality. This completes the proof. $\quad\square$

*Proof of Theorem A.2.* Let $g$ be defined as follows,

$$g:=\underset{f'\in\mathcal{F}}{\arg\min}\frac{1}{n}\sum_{(s,a,r,s')\in\mathcal{D}}\left(f'(s,a)-r-\gamma f(s',\pi)\right)^2.$$

**Bounding** $\|g-\mathcal{T}^\pi f\|_{2,\mu}$. We show that $g$ could approximate $\mathcal{T}^\pi f$ well over distribution $\mu$. We define $f_1$ as follows

$$f_1:=\underset{f'\in\mathcal{F}}{\arg\min}\|f'-\mathcal{T}^\pi f\|_{2,\mu}^2.$$

Then, we consider the following function

$$\frac{1}{n}\sum_{(s,a,r,s')\in\mathcal{D}}\left(g(s,a)-r-\gamma f(s',\pi)\right)^2-\frac{1}{n}\sum_{(s,a,r,s')\in\mathcal{D}}\left(f_1(s,a)-r-\gamma f(s',\pi)\right)^2.$$

We have

$$\mathbb{E}_{\mu \times (\mathcal{P}, R)} \left[ \frac{1}{n} \sum_{(s,a,r,s') \in \mathcal{D}} (g(s,a) - r - \gamma f(s', \pi))^2 - \frac{1}{n} \sum_{(s,a,r,s') \in \mathcal{D}} (f_1(s,a) - r - \gamma f(s', \pi))^2 \right]$$

$$= \|g - \mathcal{T}^\pi f\|_{2,\mu}^2 - \|f_1 - \mathcal{T}^\pi f\|_{2,\mu}^2,$$

by similar arguments of Eq.(A.8) and Eq.(A.10).

Then

$$\left| \|g - \mathcal{T}^\pi f\|_{2,\mu}^2 - \|f_1 - \mathcal{T}^\pi f\|_{2,\mu}^2 - \frac{1}{n} \sum_{(s,a,r,s') \in \mathcal{D}} (g(s,a) - r - \gamma f(s', \pi))^2 \right.$$

$$\left. + \frac{1}{n} \sum_{(s,a,r,s') \in \mathcal{D}} (f_1(s,a) - r - \gamma f(s', \pi))^2 \right|$$

$$\le 4V_{\max} \|g - f_1\|_{2,\mu} \sqrt{\frac{\log \frac{|\mathcal{F}||\Pi|}{\delta}}{n}} + \frac{2V_{\max}^2 \log \frac{|\mathcal{F}||\Pi|}{\delta}}{3n},$$

where the inequality follows from Lemma A.4.

Thus,

$$\|g - \mathcal{T}^\pi f\|_{2,\mu}^2$$

$$\le \frac{1}{n} \sum_{(s,a,r,s') \in \mathcal{D}} (g(s,a) - r - \gamma f(s', \pi))^2 - \frac{1}{n} \sum_{(s,a,r,s') \in \mathcal{D}} (f_1(s,a) - r - \gamma f(s', \pi))^2$$

$$+ \|f_1 - \mathcal{T}^\pi f\|_{2,\mu}^2 + 4V_{\max} \|g - f_1\|_{2,\mu} \sqrt{\frac{\log \frac{|\mathcal{F}||\Pi|}{\delta}}{n}} + \frac{2V_{\max}^2 \log \frac{|\mathcal{F}||\Pi|}{\delta}}{3n}$$

$$\le \|f_1 - \mathcal{T}^\pi f\|_{2,\mu}^2 + 4V_{\max} \|g - f_1\|_{2,\mu} \sqrt{\frac{\log \frac{|\mathcal{F}||\Pi|}{\delta}}{n}} + \frac{2V_{\max}^2 \log \frac{|\mathcal{F}||\Pi|}{\delta}}{3n}$$

$$\le \varepsilon_{\mathcal{F}, \mathcal{F}} + 4V_{\max} \|g - \mathcal{T}^\pi f\|_{2,\mu} \sqrt{\frac{\log \frac{|\mathcal{F}||\Pi|}{\delta}}{n}} + 4V_{\max} \sqrt{\varepsilon_{\mathcal{F}, \mathcal{F}}} \sqrt{\frac{\log \frac{|\mathcal{F}||\Pi|}{\delta}}{n}} + \frac{2V_{\max}^2 \log \frac{|\mathcal{F}||\Pi|}{\delta}}{3n}$$

$$\le 4V_{\max} \|g - \mathcal{T}^\pi f\|_{2,\mu} \sqrt{\frac{\log \frac{|\mathcal{F}||\Pi|}{\delta}}{n}} + \frac{5V_{\max}^2 \log \frac{|\mathcal{F}||\Pi|}{\delta}}{n} + 5\varepsilon_{\mathcal{F}, \mathcal{F}}, \quad (A.15)$$

where the second inequality follows from the optimality of $g$, and the third inequality is by Assumption 2, and the last equation follows from the Cauchy–Schwarz inequality.

By solving Eq.(A.15), we obtain

$$\|g - \mathcal{T}^\pi f\|_{2,\mu}$$

$$\le 2V_{\max} \sqrt{\frac{\log \frac{|\mathcal{F}||\Pi|}{\delta}}{n}} + \sqrt{\frac{5V_{\max}^2 \log \frac{|\mathcal{F}||\Pi|}{\delta}}{n} + 5\varepsilon_{\mathcal{F}, \mathcal{F}}}$$

$$\le 2V_{\max} \sqrt{\frac{\log \frac{|\mathcal{F}||\Pi|}{\delta}}{n}} + V_{\max} \sqrt{\frac{5 \log \frac{|\mathcal{F}||\Pi|}{\delta}}{n}} + \sqrt{5\varepsilon_{\mathcal{F}, \mathcal{F}}}$$

$$= 5V_{\max} \sqrt{\frac{\log \frac{|\mathcal{F}||\Pi|}{\delta}}{n}} + \sqrt{5\varepsilon_{\mathcal{F}, \mathcal{F}}}. \quad (A.16)$$

**Bounding $\|f - \mathcal{T}^\pi f\|_{2,\mu}$.** Similar to Eq.(A.8) and Eq.(A.10), we have

$$\frac{1}{n} \sum_{(s,a,r,s') \in \mathcal{D}} (f(s,a) - r - \gamma f(s', \pi))^2 - \frac{1}{n} \sum_{(s,a,r,s') \in \mathcal{D}} (g(s,a) - r - \gamma f(s', \pi))^2$$

$$= \frac{1}{n} \sum_{(s,a,r,s') \in \mathcal{D}} \left( (f(s,a) - r - \gamma f(s', \pi))^2 - (g(s,a) - r - \gamma f(s', \pi))^2 \right)$$

$$= \frac{1}{n} \sum_{(s,a,r,s') \in \mathcal{D}} \left( (f(s,a) - g(s,a)) \left( f(s,a) + g(s,a) - 2r - 2\gamma f(s', \pi) \right) \right)$$

and

$$\mathbb{E}_{\mu \times (\mathcal{P}, R)} \left[ (f(s,a) - r - \gamma f(s', \pi))^2 \right] - \mathbb{E}_{\mu \times (\mathcal{P}, R)} \left[ (g(s,a) - r - \gamma f(s', \pi))^2 \right]$$

$$= \mathbb{E}_{\mu \times (\mathcal{P}, R)} \left[ (f(s,a) - g(s,a)) \left( f(s,a) + g(s,a) - 2r - 2\gamma f(s', \pi) \right) \right]$$

$$= \mathbb{E}_{\mu} \left[ \mathbb{E} \left[ (f(s,a) - g(s,a)) \left( f(s,a) + g(s,a) - 2r - 2\gamma f(s', \pi) \right) | s, a \right] \right]$$

$$= \mathbb{E}_{\mu} \left[ (f(s,a) - g(s,a)) \left( f(s,a) + g_2(s,a) - 2 \left( \mathcal{T}^{\pi} f \right) (s,a) \right) \right]$$

$$= \mathbb{E}_{\mu} \left[ (f(s,a) - (\mathcal{T}^{\pi} f) (s,a))^2 \right] - \mathbb{E}_{\mu} \left[ (g(s,a) - (\mathcal{T}^{\pi} f) (s,a))^2 \right].$$

It implies that

$$\| f - \mathcal{T}^{\pi} f \|_{2,\mu}^2 - \| g - \mathcal{T}^{\pi} f \|_{2,\mu}^2$$

$$= \mathbb{E}_{\mu} \left[ (f(s,a) - (\mathcal{T}^{\pi} f) (s,a))^2 \right] - \mathbb{E}_{\mu} \left[ (g(s,a) - (\mathcal{T}^{\pi} f) (s,a))^2 \right]$$

$$= \mathbb{E}_{\mu \times (\mathcal{P}, R)} \left[ \frac{1}{n} \sum_{(s,a,r,s') \in \mathcal{D}} (f(s,a) - r - \gamma f(s', \pi))^2 - \frac{1}{n} \sum_{(s,a,r,s') \in \mathcal{D}} (g(s,a) - r - \gamma f(s', \pi))^2 \right].$$

By applying Lemma A.4,

$$\left| \| f - \mathcal{T}^{\pi} f \|_{2,\mu}^2 - \| g - \mathcal{T}^{\pi} f \|_{2,\mu}^2 \right.$$

$$\left. - \frac{1}{n} \sum_{(s,a,r,s') \in \mathcal{D}} (f(s,a) - r - \gamma f(s', \pi))^2 + \frac{1}{n} \sum_{(s,a,r,s') \in \mathcal{D}} (g(s,a) - r - \gamma f(s', \pi))^2 \right|$$

$$\leq 4 V_{\max} \| f - g \|_{2,\mu} \sqrt{\frac{\log \frac{|\mathcal{F}||\Pi|}{\delta}}{n}} + \frac{2 V_{\max}^2 \log \frac{|\mathcal{F}||\Pi|}{\delta}}{3n}$$

$$\leq 4 V_{\max} \left( \| f - \mathcal{T}^{\pi} f \|_{2,\mu} + \| g - \mathcal{T}^{\pi} f \|_{2,\mu} \right) \sqrt{\frac{\log \frac{|\mathcal{F}||\Pi|}{\delta}}{n}} + \frac{2 V_{\max}^2 \log \frac{|\mathcal{F}||\Pi|}{\delta}}{3n}$$

$$\leq 4 V_{\max} \| f - \mathcal{T}^{\pi} f \|_{2,\mu} \sqrt{\frac{\log \frac{|\mathcal{F}||\Pi|}{\delta}}{n}} + 4 V_{\max} \sqrt{\frac{\log \frac{|\mathcal{F}||\Pi|}{\delta}}{n}} \varepsilon_{\mathcal{F}, \mathcal{F}} + \frac{13 V_{\max}^2 \log \frac{|\mathcal{F}||\Pi|}{\delta}}{n},$$

where the last inequality follows from Eq.(A.16).

Rearranging the inequality above, we obtain

$$\| f - \mathcal{T}^{\pi} f \|_{2,\mu}^2$$

$$\leq \| g - \mathcal{T}^{\pi} f \|_{2,\mu}^2 + \frac{1}{n} \sum_{(s,a,r,s') \in \mathcal{D}} (f(s,a) - r - \gamma f(s', \pi))^2 - \frac{1}{n} \sum_{(s,a,r,s') \in \mathcal{D}} (g(s,a) - r - \gamma f(s', \pi))^2$$

$$+ 4 V_{\max} \| f - \mathcal{T}^{\pi} f \|_{2,\mu} \sqrt{\frac{\log \frac{|\mathcal{F}||\Pi|}{\delta}}{n}} + 4 V_{\max} \sqrt{\frac{\log \frac{|\mathcal{F}||\Pi|}{\delta}}{n}} \varepsilon_{\mathcal{F}, \mathcal{F}} + \frac{13 V_{\max}^2 \log \frac{|\mathcal{F}||\Pi|}{\delta}}{n}$$

$$\overset{(a)}{\leq} \left( 5 V_{\max} \sqrt{\frac{\log \frac{|\mathcal{F}||\Pi|}{\delta}}{n}} + \sqrt{\varepsilon_{\mathcal{F}, \mathcal{F}}} \right)^2 + \varepsilon$$

$$+ 4V_{\max}\|f - \mathcal{T}^\pi f\|_{2,\mu}\sqrt{\frac{\log\frac{|\mathcal{F}||\Pi|}{\delta}}{n}} + 4V_{\max}\sqrt{\frac{\log\frac{|\mathcal{F}||\Pi|}{\delta}}{n}}\varepsilon_{\mathcal{F},\mathcal{F}} + \frac{13V_{\max}^2\log\frac{|\mathcal{F}||\Pi|}{\delta}}{n}$$

$$= 4V_{\max}\|f - \mathcal{T}^\pi f\|_{2,\mu}\sqrt{\frac{\log\frac{|\mathcal{F}||\Pi|}{\delta}}{n}} + 14V_{\max}\sqrt{\frac{\log\frac{|\mathcal{F}||\Pi|}{\delta}}{n}}\varepsilon_{\mathcal{F},\mathcal{F}} + \frac{38V_{\max}^2\log\frac{|\mathcal{F}||\Pi|}{\delta}}{n} + \varepsilon_{\mathcal{F},\mathcal{F}} + \varepsilon. \tag{A.17}$$

where (a) follows from Eq.(A.16) and the definition of $\varepsilon$ in the theorem statement.

Solving the quadratic form of Eq.(A.17), we have

$$\|f - \mathcal{T}^\pi f\|_{2,\mu}$$

$$\leq 2V_{\max}\sqrt{\frac{\log\frac{|\mathcal{F}||\Pi|}{\delta}}{n}} + \sqrt{\frac{38V_{\max}^2\log\frac{|\mathcal{F}||\Pi|}{\delta}}{n} + 14V_{\max}\sqrt{\frac{\log\frac{|\mathcal{F}||\Pi|}{\delta}}{n}}\varepsilon_{\mathcal{F},\mathcal{F}} + \varepsilon_{\mathcal{F},\mathcal{F}} + \varepsilon}$$

$$\leq 2V_{\max}\sqrt{\frac{\log\frac{|\mathcal{F}||\Pi|}{\delta}}{n}} + \sqrt{\frac{38V_{\max}^2\log\frac{|\mathcal{F}||\Pi|}{\delta}}{n}} + \sqrt{14V_{\max}\sqrt{\frac{\log\frac{|\mathcal{F}||\Pi|}{\delta}}{n}}\varepsilon_{\mathcal{F},\mathcal{F}}} + \sqrt{\varepsilon_{\mathcal{F},\mathcal{F}} + \varepsilon}$$

$$\leq V_{\max}\sqrt{\frac{67\log\frac{|\mathcal{F}||\Pi|}{\delta}}{n}} + \sqrt[4]{\frac{196V_{\max}^2\log\frac{|\mathcal{F}||\Pi|}{\delta}}{n}\varepsilon_{\mathcal{F},\mathcal{F}}} + \sqrt{\varepsilon_{\mathcal{F},\mathcal{F}} + \varepsilon}$$

$$\leq V_{\max}\sqrt{\frac{231\log\frac{|\mathcal{F}||\Pi|}{\delta}}{n}} + \sqrt{\varepsilon_{\mathcal{F},\mathcal{F}}} + \sqrt{\varepsilon_{\mathcal{F},\mathcal{F}} + \varepsilon},$$

where the last inequality follows from Cauchy–Schwarz inequality. This completes the proof. □

## A.2 Results for Linear Function Approximation

We now provide results regarding $\mathcal{E}(f, \pi; \mathcal{D})$ for linear function approximation.

The results for linear function approximation differ from those in general function approximations in two perspectives: (1) Since our linear function approximation setup are well specified, we have $\varepsilon_{\mathcal{F}_\Phi} = \varepsilon_{\mathcal{F}_\Phi,\mathcal{F}_\Phi} = 0$. It also implies that $Q^\pi \in \mathcal{F}, \forall \pi \in \Pi_\Phi$. (2) The uniform convergence argument for $\mathcal{F}_\Phi$ and $\Pi_\Phi$ can be studied more precisely (Lemma A.12 in Appendix A.2.2). We summrize the results in this section, and we defer the detailed proof to Appendix A.2.1 and Appendix A.2.2. We also define $K = |\mathcal{A}|$ over all the linear function approximation results.

**Corollary 6** (Alternative of Theorem A.1 in Linear Function Approximation). *For any $\pi \in \Pi_\Phi$, we have*

$$\mathcal{E}(Q^\pi, \pi; \mathcal{D}) \leq \frac{cV_{\max}^2 d\log\frac{V_{\max}Kd}{\delta}}{n}, \tag{A.18}$$

*where $c$ is an absolute constant.*

We define the RHS of Eq.(A.18) to be $\varepsilon_r$ in linear function approximation.

**Corollary 7** (Alternative of Theorem A.2 in Linear Function Approximation). *For any $\pi \in \Pi_\Phi$ and $f \in \mathcal{F}_\Phi$, if $\mathcal{E}(f, \pi; \mathcal{D}) \leq \varepsilon$ for any $\varepsilon > 0$, then,*

$$\|f - \mathcal{T}^\pi f\|_{2,\mu} \leq cV_{\max}\sqrt{\frac{d\log\frac{V_{\max}Kd}{\delta}}{n}} + \sqrt{\varepsilon},$$

*where $c$ is an absolute constant.*

**Theorem A.6.** *For any $\pi \in \Pi_\Phi$ and $f \in \mathcal{F}_\Phi$, if $\mathcal{E}(f, \pi; \mathcal{D}) \leq \varepsilon$ for any $\varepsilon > 0$, then,*

$$\|f - \mathcal{T}^\pi f\|_{2,\mathcal{D}} \leq cV_{\max}\sqrt{\frac{d\log\frac{V_{\max}Kd}{\delta}}{n}} + \sqrt{\varepsilon}, \tag{A.19}$$

*where $c$ is an absolute constant.*

We also define the RHS of Eq.(A.19) with $\varepsilon = \varepsilon_r$ to be $\sqrt{\varepsilon_b}$ in linear function approximation.

### A.2.1 Definitions and Basic Lemmas Used in Linear Function Approximation Results

**Lemma A.7** ([Daniely et al., 2011], Theorem 21). *Consider the hypothesis class $\mathcal{H}$ defined as follows,*

$$\mathcal{H} := \left\{ h_\theta(s) = \operatorname*{argmax}_{a \in \mathcal{A}} \phi(s,a)^\mathsf{T}\theta : \theta \in \Theta \right\}. \tag{A.20}$$

*Then, the Natarajan-dimension of $\mathcal{H}$ is bounded by, $\mathrm{Ndim}(\mathcal{H}) \le c_0 d \log d$, where $c_0 > 0$ is an absolute constant.*

**Lemma A.8** (Sauer's Lemma). *For the hypothesis class $\mathcal{H} \subset \mathcal{X} \to \{0,1\}$ with $\mathrm{VCdim}(\mathcal{H}) = d_1 \le \infty$, then for any $X = \{x_1, x_2, \dots, x_n\} \in \mathcal{X}^n$,*

$$|\mathcal{H}_X| \le (n+1)^{d_1},$$

*where $\mathcal{H}_X := \{(h(x_1), h(x_2), \dots, h(x_n)) : h \in \mathcal{H}\}$ is the restriction of $\mathcal{H}$ to $X$.*

**Lemma A.9** (Sauer's Lemma for Natarajan Dimension [Ben-David et al., 1995; Haussler and Long, 1995]). *Let $S = \{s_1, s_2, \dots, s_n\} \in \mathcal{S}^n$, $\mathcal{H}$ follow the same definition as Eq.(A.20), and $\mathcal{H}_S$ be the restriction of $\mathcal{H}$ to $S$. Then, for any $S \in \mathcal{S}^n$, if $\mathrm{Ndim}(\mathcal{H}) \le d_2$ and $|\mathcal{A}| = K$,*

$$|\mathcal{H}_\mathcal{S}| \le \left( \frac{ne(K+1)^2}{2d_2} \right)^{d_2}.$$

**Definition 3** ($L_1$ Covering Number). *Given hypothesis class $\mathcal{H} \subseteq \mathcal{X} \to \mathbb{R}$, $\varepsilon > 0$, and $X = \{x_1, x_2, \dots, x_n\} \in \mathcal{X}^n$. We define the $L_1$ covering number $\mathcal{N}(\varepsilon, \mathcal{H}, X)$ to be $\operatorname{argmin} |C_X|$, where $C_X$ satisfies*

(i) *$C_X \subseteq \mathbb{R}^n$;*
(ii) *For any $h \in \mathcal{H}$, there exists a $c_h = \{c_1, c_2, \dots, c_n\} \in C_X$, such that $\frac{1}{n}\sum_{i=1}^n |h(x_i) - c_i| \le \varepsilon$.*

**Lemma A.10** (Bounding $L_1$ Covering Number by Pseudo Dimension, [Haussler, 1995]). *Given hypothesis class $\mathcal{H} \subseteq \mathcal{X} \to [0,1]$ with $\mathrm{Pdim}(\mathcal{H}) \le d_\mathcal{H}$, we have for any $X \in \mathcal{X}^n$,*

$$\mathcal{N}(\varepsilon, \mathcal{H}, X) \le e\,(d_\mathcal{H}+1)\left( \frac{2e}{\varepsilon} \right)^{d_\mathcal{H}}.$$

### A.2.2 Detailed Proofs for Linear Function Approximation Results

**Lemma A.11.** *Let $\mathcal{X} := \mathcal{S} \times \mathcal{A}$ and $|\mathcal{A}| = K$. Let $\Pi \subseteq \mathcal{S} \to \mathcal{A}$ be a policy class with Natarajan dimension $\mathrm{Ndim}(\Pi) = d_\Pi$, and $\mathcal{G} \subseteq \mathcal{X} \to [0, V_{\max}]$ with pseudo dimension $\mathrm{Pdim}(\mathcal{G}) = d_\mathcal{G}$. Then,*

(i) *The hypothesis class $\mathcal{H}_1 = \{(x, x') \to f(x)g(x') : f, g \in \mathcal{G}\}$ has pseudo dimension $\mathrm{Pdim}(\mathcal{H}_3) \le c_1 d_\mathcal{G} \log(d_\mathcal{G})$.*
(ii) *The hypothesis class $\mathcal{H}_2 = \{(x, x') \to f(x) + g(x') : f, g \in \mathcal{G}\}$ has pseudo dimension $\mathrm{Pdim}(\mathcal{H}_3) \le c_2 d_\mathcal{G} \log(d_\mathcal{G})$.*
(iii) *The hypothesis class $\mathcal{H}_3 = \{(x, a, x') \to f(x)\mathbb{1}(a = \pi(x)) : f \in \mathcal{G}, \pi \in \Pi\}$ has pseudo dimension $\mathrm{Pdim}(\mathcal{H}_3) \le c_3 (d_\Pi + d_\mathcal{G}) \log(K(d_\Pi + d_\mathcal{G}))$.*

***Proof of Lemma A.11.*** We first prove *(i)*. Let $\mathcal{H}_1^+ := \{(x, x', \zeta) \to \mathbb{1}(f(x)g(x') > \zeta) : f, g \in \mathcal{G}\} \subseteq \mathcal{X} \times \mathcal{X} \times \mathbb{R} \to \{0,1\}$. It suffices to show that for any $X = \{(x_1, x_1', \zeta_1), (x_2, x_2', \zeta_2), \dots, (x_n, x_n', \zeta_n)\} \in (\mathcal{X} \times \mathcal{X} \times \mathbb{R})^n$, $\log |\mathcal{H}_{1,X}^+| < n$, where $n = c_1 d_\mathcal{G} \log(d_\mathcal{G})$ for some absolute constant $c_1 > 0$.

Let $\mathcal{G}^+ := \{(x, \xi) \to \mathbb{1}(f(x) > \xi) : f \in \mathcal{G}\} \subseteq \mathcal{X} \times \mathbb{R} \to \{0,1\}$. Since we have $\mathbb{1}(f(x)g(x') > \zeta) = \mathbb{1}(f(x) > \xi)\mathbb{1}(g(x') > \zeta/\xi)$, this implies that for any $X = \{(x_1, x_1', \zeta_1), (x_2, x_2', \zeta_2), \dots, (x_n, x_n', \zeta_n)\} \in (\mathcal{X} \times \mathcal{X} \times \mathbb{R})^n$, there exists

$$X' = \{(x_1, \xi_1), (x_2, \xi_2), \dots, (x_n, \xi_n)\} \in (\mathcal{X} \times \mathbb{R})^n,$$
$$X'' = \{(x_1', \xi_1'), (x_2', \xi_2'), \dots, (x_n', \xi_n')\} \in (\mathcal{X} \times \mathbb{R})^n,$$

where $\xi_i \xi_i' = \zeta_i, \forall i \in \{1, 2, \ldots, n\}$, such that,

$$\left| \mathcal{H}_{1,X}^+ \right| = \left| \mathcal{G}_{X'}^+ \right| \left| \mathcal{G}_{X''}^+ \right|.$$

By Lemma A.8 and the fact of $\mathrm{Pdim}(\mathcal{G}) = \mathrm{VCdim}(\mathcal{G}^+)$, we know

$$\left| \mathcal{G}_X^+ \right| \le (n+1)^{d_\mathcal{G}}, \ \forall X \in (\mathcal{X} \times \mathbb{R})^n.$$

Therefore,

$$\begin{aligned}
\log \left| \mathcal{H}_{1,X}^+ \right| &\le 2 \log (n+1)^{d_\mathcal{G}} \\
&\le 2 d_\mathcal{G} \log (c_1 d_\mathcal{G} \log(d_\mathcal{G}) + 1).
\end{aligned}$$

It is easy to verify that

$$2 d_\mathcal{G} \log (c_1 d_\mathcal{G} \log(d_\mathcal{G}) + 1) < c_1 d_\mathcal{G} \log(d_\mathcal{G})$$

if we choose $c_1$ properly.

The proof of *(ii)* can be derived similarly, as we have $\mathbb{1}(f(x)g(x') > \zeta) = \mathbb{1}(f(x) > \xi) + \mathbb{1}(g(x') > \zeta/\xi)$. Therefore, we obtain $\log |\mathcal{H}_{2,X}^+| \le c_2 d_\mathcal{G} \log(d_\mathcal{G})$ by following the same amendments above.

*(iii)* is obtained by [Jiang et al., 2017, Lemma 21]. This completes the proof. $\qquad\square$

We now provide an alternative of Lemma A.4 in linear function approximation.

**Lemma A.12** (Alternative of Lemma A.4 in Linear Function Approximation). *For any $f, g_1, g_2 \in \mathcal{F}_\Phi$ and $\pi \in \Pi_\Phi$, we have*

$$\begin{aligned}
&\left| \|g_1 - \mathcal{T}^\pi f\|_{2,\mu}^2 - \|g_2 - \mathcal{T}^\pi f\|_{2,\mu}^2 \right. \\
&\quad \left. - \frac{1}{n} \sum_{(s,a,r,s') \in \mathcal{D}} (g_1(s,a) - r - \gamma f(s',\pi))^2 + \frac{1}{n} \sum_{(s,a,r,s') \in \mathcal{D}} (g_2(s,a) - r - \gamma f(s',\pi))^2 \right| \\
&\le c V_{\max} \|g_1(s,a) - g_2(s,a)\|_{2,\mu} \sqrt{\frac{d \log \frac{V_{\max} K d}{\delta}}{n}} + c' \frac{V_{\max}^2 d \log \frac{V_{\max} K d}{\delta}}{n},
\end{aligned}$$

*where $c$ and $c'$ are absolute constants.*

***Proof of Lemma A.12.*** The only difference between this lemma and Lemma A.4 is the concentration of

$$\begin{aligned}
&\frac{1}{n} \sum_{(s,a,r,s') \in \mathcal{D}} (g_1(s,a) - r - \gamma f(s',\pi))^2 - \frac{1}{n} \sum_{(s,a,r,s') \in \mathcal{D}} (g_2(s,a) - r - \gamma f(s',\pi))^2 \\
&= \frac{1}{n} \sum_{(s,a,r,s') \in \mathcal{D}} ((g_1(s,a) - g_2(s,a))(g_1(s,a) + g_2(s,a) - 2r - 2\gamma f(s',\pi))) \\
&= \frac{1}{n} \sum_{(s,a,r,s') \in \mathcal{D}} \left( (g_1(s,a) - g_2(s,a)) \left( g_1(s,a) + g_2(s,a) - 2r - 2\gamma \sum_{a' \in \mathcal{A}} f(s',a') \mathbb{1}(a' = \pi(s')) \right) \right).
\end{aligned}$$

Therefore, we define the hypothesis class $\mathcal{H}$ as follows

$$\mathcal{H} = \left\{ (s,a,r,s') \rightarrow (g_1(s,a) - g_2(s,a)) \left( g_1(s,a) + g_2(s,a) - 2r - 2\gamma \sum_{a' \in \mathcal{A}} f(s',a') \mathbb{1}(a' = \pi(s')) \right) : \right.$$

$$\left. \forall g_1, g_2, f \in \mathcal{F}_\Phi, \pi \in \Pi_\Phi \right\}.$$

By definitions of $\mathcal{F}_\Phi$ and $\Pi_\Phi$, we have

$$\mathrm{Pdim}(\mathcal{F}_\Phi), \mathrm{Ndim}(\Pi_\Phi) \leq c_1 d \log d,$$

where $c_1 > 0$ is an absolute constant, $d$ is the feature dimension of linear function approximation, the upper bound of $\mathrm{Pdim}(\mathcal{F}_\Phi)$ follows from the standard argument of VC-dimension for the linear function class (e.g., [Mohri et al., 2018, Section 3.3]), and the upper bound of $\mathrm{Ndim}(\Pi_\Phi)$ is obtained by Lemma A.7.

Since $\mathcal{H}$ is an composited class of $\mathcal{F}_\Pi$ and $\Pi_\Phi$ and the composition only uses the operations studied in Lemma A.11, we can also obtain that $\mathrm{Pdim}(\mathcal{H}) \leq c_2 d \log(Kd)$, where $c_2 > 0$ is an absolute constant.

Then,

$$\left| \mathbb{E}_\mu \left[ (g_1(s,a) - (\mathcal{T}^\pi f)(s,a))^2 \right] - \mathbb{E}_\mu \left[ (g_2(s,a) - (\mathcal{T}^\pi f)(s,a))^2 \right] - \right.$$
$$\left. \frac{1}{n} \sum_{(s,a,r,s') \in \mathcal{D}} (g_1(s,a) - r - \gamma f(s',\pi))^2 + \frac{1}{n} \sum_{(s,a,r,s') \in \mathcal{D}} (g_2(s,a) - r - \gamma f(s',\pi))^2 \right|$$

$$= \left| \mathbb{E}_\mu \left[ (g_1(s,a) - (\mathcal{T}^\pi f)(s,a))^2 \right] - \mathbb{E}_\mu \left[ (g_2(s,a) - (\mathcal{T}^\pi f)(s,a))^2 \right] - \right.$$
$$\left. \frac{1}{n} \sum_{(s,a,r,s') \in \mathcal{D}} ((g_1(s,a) - g_2(s,a))(g_1(s,a) + g_2(s,a) - 2r - 2\gamma f(s',\pi))) \right|$$

$$\leq \sqrt{\frac{4\mathbb{V}_{\mu \times (\mathcal{P},R)} \left[ (g_1(s,a) - g_2(s,a))(g_1(s,a) + g_2(s,a) - 2r - 2\gamma f(s',\pi)) \right] \log \frac{\mathcal{N}(\varepsilon,\mathcal{H},X)}{\delta}}{n}}$$
$$+ \frac{2V_{\max}^2 \log \frac{\mathcal{N}(\varepsilon,\mathcal{H},X)}{\delta}}{3n},$$

where the last inequality follows from the Bernsiten's inequality and the definition of $\mathcal{N}(\varepsilon,\mathcal{H},X)$.

Similar to the proof of Lemma A.4, we also have

$$\mathbb{V}_{\mu \times (\mathcal{P},R)} \left[ (g_1(s,a) - g_2(s,a))(g_1(s,a) + g_2(s,a) - 2r - 2\gamma f(s',\pi)) \right]$$
$$\leq 4V_{\max}^2 \mathbb{E}_\mu [(g_1(s,a) - g_2(s,a))^2].$$

Then, we obtain

$$\left| \mathbb{E}_\mu \left[ (g_1(s,a) - (\mathcal{T}^\pi f)(s,a))^2 \right] - \mathbb{E}_\mu \left[ (g_2(s,a) - (\mathcal{T}^\pi f)(s,a))^2 \right] - \right.$$
$$\left. \frac{1}{n} \sum_{(s,a,r,s') \in \mathcal{D}} (g_1(s,a) - r - \gamma f(s',\pi))^2 + \frac{1}{n} \sum_{(s,a,r,s') \in \mathcal{D}} (g_2(s,a) - r - \gamma f(s',\pi))^2 \right|$$

$$\leq 4V_{\max} \|g_1(s,a) - g_2(s,a)\|_{2,\mu} \sqrt{\frac{\log \frac{\mathcal{N}(\varepsilon,\mathcal{H},X)}{\delta}}{n}} + \frac{2V_{\max}^2 \log \frac{\mathcal{N}(\varepsilon,\mathcal{H},X)}{\delta}}{3n}.$$

We now prove that, for any $\mathcal{H}' \subseteq \mathcal{X} \to [a,b]$,

$$\mathcal{N}(\varepsilon,\mathcal{H}',X) \leq e(d_{\mathcal{H}'} + 1) \left( \frac{2e(b-a)}{\varepsilon} \right)^{d_{\mathcal{H}'}},$$

Let $\mathcal{H}'' = \{(h(\cdot) - a)/(b-a) : h \in \mathcal{H}'\} \subseteq \mathcal{X} \to [0,1]$, then we have $d_{\mathcal{H}'} = d_{\mathcal{H}''}$ by Lemma A.11. By Definition 3 and Lemma A.10, it implies that

$$\mathcal{N}(\varepsilon,\mathcal{H}',X) = \mathcal{N}(\varepsilon/(b-a),\mathcal{H}'',X) = (d_{\mathcal{H}'} + 1) \left( \frac{2e(b-a)}{\varepsilon} \right)^{d_{\mathcal{H}'}},$$

by simple algebra. Since $\mathcal{H} \subseteq \mathcal{X} \to [-2V_{\max}^2, 2V_{\max}^2]$, we can just plug in $a = -2V_{\max}^2$ and $b = 2V_{\max}^2$, and obtain

$$\mathcal{N}(\varepsilon,\mathcal{H},X) = (d_{\mathcal{H}} + 1) \left( \frac{8eV_{\max}^2}{\varepsilon} \right)^{d_{\mathcal{H}}}, \tag{A.21}$$

for any $X \in \mathcal{S} \times \mathcal{A} \times \mathbb{R} \times \mathcal{S}$.

By plugging in the definition of $\mathcal{N}(\varepsilon, \mathcal{H}, X)$ in Eq.(A.21), we can verify that

$$
\Bigg| \mathbb{E}_\mu \left[ (g_1(s,a) - (\mathcal{T}^\pi f)(s,a))^2 \right] - \mathbb{E}_\mu \left[ (g_2(s,a) - (\mathcal{T}^\pi f)(s,a))^2 \right] -
$$
$$
\frac{1}{n} \sum_{(s,a,r,s') \in \mathcal{D}} (g_1(s,a) - r - \gamma f(s',\pi))^2 + \frac{1}{n} \sum_{(s,a,r,s') \in \mathcal{D}} (g_2(s,a) - r - \gamma f(s',\pi))^2 \Bigg|
$$
$$
\leq cV_{\max} \|g_1(s,a) - g_2(s,a)\|_{2,\mu} \sqrt{\frac{d \log \frac{V_{\max} K d}{\delta}}{n}} + c' \frac{V_{\max}^2 d \log \frac{V_{\max} K d}{\delta}}{n},
$$

where $c$ and $c'$ are absolute constants. This completes the proof. $\qquad\square$

**Corollary 8** (Alternative of Lemma A.3 in Linear Function Approximation). *For any $f_1, f_2 \in \mathcal{F}_\Phi$ and $\pi \in \Pi_\Phi$, w.p. $1 - \delta$,*

$$
\Big| \|f_1(s,a) - (\mathcal{T}^\pi f_2)(s,a)\|_{2,\mu} - \|f_1(s,a) - (\mathcal{T}^\pi f_2)(s,a)\|_{2,\mathcal{D}} \Big| \leq cV_{\max} \sqrt{\frac{d \log \frac{V_{\max} K d}{\delta}}{n}},
$$

*where $c$ is an absolute constant.*

***Proof of Theorem A.6.*** By combining Corollary 8 and Corollary 7, we complete the proof. $\qquad\square$

# B  Detailed Proofs in Section 3

## B.1  Detailed Proofs for General Function Approximation

Over this section, the definition of $\varepsilon_r$ follows from Eq.(A.2).

**Lemma B.1.** *For any $\pi \in \Pi$, $\min\limits_{f \in \mathcal{F}_{\pi,\varepsilon_r}} \|Q^\pi - f\|_{2,\nu} \leq \frac{\sqrt{\varepsilon_\mathcal{F}}}{1-\gamma}$ for any admissible distribution $\nu$.*

*Proof.* Let $f_\pi := \operatorname{argmin}_{f \in \mathcal{F}} \max_{\text{admissible } \nu} \|f - \mathcal{T}^\pi f\|_{2,\nu}$. By definition of $\mathcal{F}_{\pi,\varepsilon_r}$, we know $f_\pi \in \mathcal{F}_{\pi,\varepsilon_r}$. Then,

$$
\min_{f \in \mathcal{F}_{\pi,\varepsilon_r}} \|Q^\pi - f\|_{2,\nu} \leq \|Q^\pi - f_\pi\|_{2,\nu} \leq \frac{1}{1-\gamma} \max_{\text{admissible } \nu} \|f_\pi - \mathcal{T}^\pi f_\pi\|_{2,\nu} \leq \frac{\sqrt{\varepsilon_\mathcal{F}}}{1-\gamma}. \qquad\square
$$

**Lemma B.2.** *For any $\pi \in \Pi$, $\min\limits_{f \in \mathcal{F}_{\pi,\varepsilon_r}} f(s_0, \pi) \leq J(\pi) + \frac{\sqrt{\varepsilon_\mathcal{F}}}{1-\gamma}$.*

***Proof of Lemma B.2.*** Let $f_\pi := \operatorname{argmin}_{f \in \mathcal{F}} \max_{\text{admissible } \nu} \|f - \mathcal{T}^\pi f\|_{2,\nu}$.

$$
\min_{f \in \mathcal{F}_{\pi,\varepsilon_r}} f(s_0, \pi) \leq f_\pi(s_0, \pi) \leq Q^\pi(s_0, \pi) + \frac{\sqrt{\varepsilon_\mathcal{F}}}{1-\gamma} = J(\pi) + \frac{\sqrt{\varepsilon_\mathcal{F}}}{1-\gamma}. \qquad\square
$$

Therefore, the optimization objective is actually a valid lower bound of $J(\pi)$. Similarly, we have the following symmetrical result.

**Lemma B.3.** *For any $\pi \in \Pi$, $\max\limits_{f \in \mathcal{F}_{\pi,\varepsilon_r}} f(s_0, \pi) \geq J(\pi) - \frac{\sqrt{\varepsilon_\mathcal{F}}}{1-\gamma}$.*

***Proof of Lemma B.3.*** Let $f_\pi := \operatorname{argmin}_{f \in \mathcal{F}} \max_{\text{admissible } \nu} \|f - \mathcal{T}^\pi f\|_{2,\nu}$.

$$
\max_{f \in \mathcal{F}_{\pi,\varepsilon_r}} f(s_0, \pi) \geq f_\pi(s_0, \pi) \geq Q^\pi(s_0, \pi) - \frac{\sqrt{\varepsilon_\mathcal{F}}}{1-\gamma} = J(\pi) - \frac{\sqrt{\varepsilon_\mathcal{F}}}{1-\gamma}. \qquad\square
$$

We now ready to provide the proof of Theorem 3.1.

***Proof of Theorem 3.1.*** Using the optimality of $\widehat{\pi}$, we have

$$\max_{f\in\mathcal{F}_{\pi,\varepsilon_r}} f(s_0,\pi) - \min_{f\in\mathcal{F}_{\widehat{\pi},\varepsilon_r}} f(s_0,\widehat{\pi}) \le \max_{f\in\mathcal{F}_{\pi,\varepsilon_r}} f(s_0,\pi) - \min_{f\in\mathcal{F}_{\pi,\varepsilon_r}} f(s_0,\pi).$$

Now, let $f_{\pi,\min} := \operatorname{argmin}_{f\in\mathcal{F}_{\pi,\varepsilon_r}} f(s_0,\pi)$ and $f_{\pi,\max} := \operatorname{argmax}_{f\in\mathcal{F}_{\pi,\varepsilon_r}} f(s_0,\pi)$. By a standard telescoping argument (e.g., [Xie and Jiang, 2020, Lemma 1]), we can obtain

$$
\begin{aligned}
&f_{\pi,\max}(\pi,s_0) - f_{\pi,\min}(\pi,s_0) \\
&= f_{\pi,\max}(\pi,s_0) - J(\pi) + J(\pi) - f_{\pi,\min}(\pi,s_0) \\
&= \frac{1}{1-\gamma} \left( \mathbb{E}_{d_\pi}\left[f_{\pi,\max} - \mathcal{T}^\pi f_{\pi,\max}\right] - \mathbb{E}_{d_\pi}\left[f_{\pi,\min} - \mathcal{T}^\pi f_{\pi,\min}\right] \right) \\
&= \frac{1}{1-\gamma} \big( \mathbb{E}_\mu\left[\nu/\mu \cdot ((f_{\pi,\max} - \mathcal{T}^\pi f_{\pi,\max}) - (f_{\pi,\min} - \mathcal{T}^\pi f_{\pi,\min}))\right] \\
&\quad + \mathbb{E}_{d_\pi}\left[(f_{\pi,\max} - \mathcal{T}^\pi f_{\pi,\max}) - (f_{\pi,\min} - \mathcal{T}^\pi f_{\pi,\min})\right] \\
&\quad - \mathbb{E}_\nu\left[(f_{\pi,\max} - \mathcal{T}^\pi f_{\pi,\max}) - (f_{\pi,\min} - \mathcal{T}^\pi f_{\pi,\min})\right] \big) \\
&= \underbrace{\frac{1}{1-\gamma} \left( \mathbb{E}_\mu\left[\nu/\mu \cdot ((f_{\pi,\max} - \mathcal{T}^\pi f_{\pi,\max}) - (f_{\pi,\min} - \mathcal{T}^\pi f_{\pi,\min}))\right] \right)}_{\text{(I)}} \\
&\quad + \underbrace{\frac{1}{1-\gamma} \left( \mathbb{E}_{d_\pi}\left[\Delta f_\pi - \gamma \mathcal{P}^\pi \Delta f_\pi\right] - \mathbb{E}_\nu\left[\Delta f_\pi - \gamma \mathcal{P}^\pi \Delta f_\pi\right] \right)}_{\text{(II)}}, \qquad (\Delta f_\pi := f_{\pi,\max} - f_{\pi,\min})
\end{aligned}
$$

where $\nu$ is an arbitrary on-support state-action distribution. We now discuss these two terms above separately.

For the term (I),

$$
\begin{aligned}
\text{(I)} &\le \frac{1}{1-\gamma} \left| \mathbb{E}_\mu\left[\nu/\mu \cdot (f_{\pi,\max} - \mathcal{T}^\pi f_{\pi,\max})\right] \right| + \frac{1}{1-\gamma} \left| \mathbb{E}_\mu\left[\nu/\mu \cdot (f_{\pi,\min} - \mathcal{T}^\pi f_{\pi,\min})\right] \right| \\
&\le \frac{\sqrt{\mathscr{C}(\nu;\mu,\mathcal{F},\pi)}}{1-\gamma} \left( \|f_{\pi,\max} - \mathcal{T}^\pi f_{\pi,\max}\|_{2,\mu} + \|f_{\pi,\min} - \mathcal{T}^\pi f_{\pi,\min}\|_{2,\mu} \right),
\end{aligned}
$$

because of the Cauchy-Schwarz inequality for random variables ($|\mathbb{E}[XY]| \le \sqrt{\mathbb{E}[X^2]\mathbb{E}[Y^2]}$).

For the term (II),

$$
\begin{aligned}
\text{(II)} &= \frac{1}{1-\gamma} \left( \mathbb{E}_{d_\pi}\left[\Delta f_\pi - \gamma \mathcal{P}^\pi \Delta f_\pi\right] - \mathbb{E}_\nu\left[\Delta f_\pi - \gamma \mathcal{P}^\pi \Delta f_\pi\right] \right) \\
&= \frac{1}{1-\gamma} \sum_{(s,a)\in\mathcal{S}\times\mathcal{A}} \left[d_\pi(s,a) - \nu(s,a)\right]\left[\Delta f_\pi(s,a) - \gamma(\mathcal{P}^\pi \Delta f_\pi)(s,a)\right] \\
&\le \frac{1}{1-\gamma} \sum_{(s,a)\in\mathcal{S}\times\mathcal{A}} \mathbb{1}(d_\pi(s,a) \ge \nu(s,a))\left[d_\pi(s,a) - \nu(s,a)\right]\left[\Delta f_\pi(s,a) - \gamma(\mathcal{P}^\pi \Delta f_\pi)(s,a)\right] \\
&\quad + \frac{1}{1-\gamma} \left| \sum_{(s,a)\in\mathcal{S}\times\mathcal{A}} \mathbb{1}(\nu(s,a) > d_\pi(s,a))\left[\nu(s,a) - d_\pi(s,a)\right]\left[\Delta f_\pi(s,a) - \gamma(\mathcal{P}^\pi \Delta f_\pi)(s,a)\right] \right| \\
&\le \frac{1}{1-\gamma} \sum_{(s,a)\in\mathcal{S}\times\mathcal{A}} (d_\pi \setminus \nu)(s,a)\left[\Delta f_\pi(s,a) - \gamma(\mathcal{P}^\pi \Delta f_\pi)(s,a)\right] \\
&\quad + \frac{1}{1-\gamma} \sum_{(s,a)\in\mathcal{S}\times\mathcal{A}} \mathbb{1}(\nu(s,a) > d_\pi(s,a))\left[\nu(s,a) - d_\pi(s,a)\right]\left|\Delta f_\pi(s,a) - \gamma(\mathcal{P}^\pi \Delta f_\pi)(s,a)\right| \\
&\le \frac{1}{1-\gamma} \sum_{(s,a)\in\mathcal{S}\times\mathcal{A}} (d_\pi \setminus \nu)(s,a)\left[\Delta f_\pi(s,a) - \gamma(\mathcal{P}^\pi \Delta f_\pi)(s,a)\right]
\end{aligned}
$$

$$+ \frac{1}{1-\gamma}\mathbb{E}_\nu\left[|f_{\pi,\max} - \mathcal{T}^\pi f_{\pi,\max}| + |f_{\pi,\min} - \mathcal{T}^\pi f_{\pi,\min}|\right]$$

$$\leq \frac{1}{1-\gamma}\sum_{(s,a)\in\mathcal{S}\times\mathcal{A}}(d_\pi\setminus\nu)(s,a)\left[\Delta f_\pi(s,a) - \gamma(\mathcal{P}^\pi\Delta f_\pi)(s,a)\right]$$

$$+ \frac{\sqrt{\mathscr{C}(\nu;\mu,\mathcal{F},\pi)}}{1-\gamma}\left(\|f_{\pi,\max} - \mathcal{T}^\pi f_{\pi,\max}\|_{2,\mu} + \|f_{\pi,\min} - \mathcal{T}^\pi f_{\pi,\min}\|_{2,\mu}\right),$$

where the second last inequality follows from the fact of $\nu(s,a) \geq \mathbb{1}(\nu(s,a) > d_\pi(s,a))[\nu(s,a) - d_\pi(s,a)]$ for any $(s,a) \in \mathcal{S}\times\mathcal{A}$ and the triangle inequality for the absolute value, and the last inequality uses the Cauchy-Schwarz inequality for random variables (similar to the argument for the term (I)).

Combining the bounds of both term (I) and term (II), we have

$$f_{\pi,\max}(\pi,s_0) - f_{\pi,\min}(\pi,s_0)$$

$$\leq \frac{1}{1-\gamma}\sum_{(s,a)\in\mathcal{S}\times\mathcal{A}}(d_\pi\setminus\nu)(s,a)\left[\Delta f_\pi(s,a) - \gamma(\mathcal{P}^\pi\Delta f_\pi)(s,a)\right] \tag{B.1}$$

$$+ \frac{2\sqrt{\mathscr{C}(\nu;\mu,\mathcal{F},\pi)}}{1-\gamma}\left(\|f_{\pi,\max} - \mathcal{T}^\pi f_{\pi,\max}\|_{2,\mu} + \|f_{\pi,\min} - \mathcal{T}^\pi f_{\pi,\min}\|_{2,\mu}\right).$$

Since Eq.(B.1) holds for arbitrary on-support state-action distribution $\nu$, we take the minimal over the set of all $\{\nu : \mathscr{C}(\nu;\mu,\mathcal{F},\pi) \leq C_2\}$ ($C_2$ denotes the $L^2$ concentrability threshold), and obtain

$$f_{\pi,\max}(\pi,s_0) - f_{\pi,\min}(\pi,s_0)$$

$$\leq \min_{\nu:\mathscr{C}(\nu;\mu,\mathcal{F},\pi)\leq C_2}\left(\frac{2\sqrt{\mathscr{C}(\nu;\mu,\mathcal{F},\pi)}}{1-\gamma}\left(\|f_{\pi,\max} - \mathcal{T}^\pi f_{\pi,\max}\|_{2,\mu} + \|f_{\pi,\min} - \mathcal{T}^\pi f_{\pi,\min}\|_{2,\mu}\right)\right.$$

$$\left. + \frac{1}{1-\gamma}\sum_{(s,a)\in\mathcal{S}\times\mathcal{A}}(d_\pi\setminus\nu)(s,a)\left[\Delta f_\pi(s,a) - \gamma(\mathcal{P}^\pi\Delta f_\pi)(s,a)\right]\right)$$

$$\leq \frac{2\sqrt{C_2}}{1-\gamma}\left(\|f_{\pi,\max} - \mathcal{T}^\pi f_{\pi,\max}\|_{2,\mu} + \|f_{\pi,\min} - \mathcal{T}^\pi f_{\pi,\min}\|_{2,\mu}\right)$$

$$+ \min_{\nu:\mathscr{C}(\nu;\mu,\mathcal{F},\pi)\leq C_2}\left(\frac{1}{1-\gamma}\sum_{(s,a)\in\mathcal{S}\times\mathcal{A}}(d_\pi\setminus\nu)(s,a)\left[\Delta f_\pi(s,a) - \gamma(\mathcal{P}^\pi\Delta f_\pi)(s,a)\right]\right)$$

$$\leq \frac{4\sqrt{C_2}\sqrt{\varepsilon_b}}{1-\gamma}$$

$$+ \min_{\nu:\mathscr{C}(\nu;\mu,\mathcal{F},\pi)\leq C_2}\left(\frac{1}{1-\gamma}\sum_{(s,a)\in\mathcal{S}\times\mathcal{A}}(d_\pi\setminus\nu)(s,a)\left[\Delta f_\pi(s,a) - \gamma(\mathcal{P}^\pi\Delta f_\pi)(s,a)\right]\right).$$

This implies the bound of $J(\pi) - J(\widehat{\pi})$,

$$J(\pi) - J(\widehat{\pi}) \leq \max_{f\in\mathcal{F}_{\pi,\varepsilon_r}} f(s_0,\pi) - \min_{f\in\mathcal{F}_{\widehat{\pi},\varepsilon_r}} f(s_0,\widehat{\pi}) + \frac{2\sqrt{\varepsilon_\mathcal{F}}}{1-\gamma}$$

$$\leq \max_{f\in\mathcal{F}_{\pi,\varepsilon_r}} f(s_0,\pi) - \min_{f\in\mathcal{F}_{\pi,\varepsilon_r}} f(s_0,\pi) + \frac{2\sqrt{\varepsilon_\mathcal{F}}}{1-\gamma}$$

$$= f_{\pi,\max}(\pi,s_0) - f_{\pi,\min}(\pi,s_0) + \frac{2\sqrt{\varepsilon_\mathcal{F}}}{1-\gamma}$$

$$\leq \frac{4\sqrt{C_2}\sqrt{\varepsilon_b}}{1-\gamma} + \frac{2\sqrt{\varepsilon_\mathcal{F}}}{1-\gamma}$$

$$+ \min_{\nu:\mathscr{C}(\nu;\mu,\mathcal{F},\pi)\leq C_2}\left(\frac{1}{1-\gamma}\sum_{(s,a)\in\mathcal{S}\times\mathcal{A}}(d_\pi\setminus\nu)(s,a)\left[\Delta f_\pi(s,a) - \gamma(\mathcal{P}^\pi\Delta f_\pi)(s,a)\right]\right).$$

Plugging the definition of $\varepsilon_b$ (in Eq.(A.4)), we complete the proof. $\square$

## B.2 Detailed Proofs for Linear Function Approximation Results

***Proof of Theorem 3.2.*** We use $\Theta$ to denote the parameter space of $\mathcal{F}_\Phi$, i.e., $\mathcal{F}_\Phi = \{\phi^\mathsf{T}\theta : \theta \in \Theta\}$. And we also use $\Theta_{\pi,\varepsilon_r}$ to denote the version space in the parameter space accordingly, i.e., $\Theta_{\pi,\varepsilon_r} = \{\theta \in \Theta : \mathcal{E}(f,\pi;\mathcal{D}) \leq \varepsilon_r\}$. Now, using the optimality of $\widehat{\pi}$, we have

$$\max_{\theta \in \Theta_{\pi,\varepsilon_r}} \phi(s_0,\pi)^\mathsf{T}\theta - \min_{\theta \in \Theta_{\widehat{\pi},\varepsilon_r}} \phi(s_0,\widehat{\pi})^\mathsf{T}\theta \leq \max_{\theta \in \Theta_{\pi,\varepsilon_r}} \phi(s_0,\pi)^\mathsf{T}\theta - \min_{\theta \in \Theta_{\pi,\varepsilon_r}} \phi(s_0,\pi)^\mathsf{T}\theta. \quad \text{(B.2)}$$

Let $\theta_{\pi,\min} \coloneqq \operatorname{argmin}_{\theta \in \Theta_{\pi,\varepsilon_r}} \phi(s_0,\pi)^\mathsf{T}\theta$ and $\theta_{\pi,\max} \coloneqq \operatorname{argmax}_{\theta \in \Theta_{\pi,\varepsilon_r}} \phi(s_0,\pi)^\mathsf{T}\theta$. By a standard telescoping argument (e.g., [Xie and Jiang, 2020, Lemma 1]), we can obtain

$$\begin{aligned}
&\left|\phi(s_0,\pi)^\mathsf{T}\theta_{\pi,\min} - J(\pi)\right| \\
&= \frac{1}{1-\gamma}\left|\mathbb{E}_{d_\pi}\left[\phi(s,a)^\mathsf{T}\theta_{\pi,\min} - \left(\mathcal{T}^\pi\phi^\mathsf{T}\theta_{\pi,\min}\right)(s,a)\right]\right| \\
&\leq \frac{1}{1-\gamma}\mathbb{E}_{d_\pi}\left|\phi(s,a)^\mathsf{T}\theta_{\pi,\min} - \left(\mathcal{T}^\pi\phi^\mathsf{T}\theta_{\pi,\min}\right)(s,a)\right| \\
&= \frac{1}{1-\gamma}\mathbb{E}_{d_\pi}\left|\phi(s,a)^\mathsf{T}\left(\underbrace{\theta_{\pi,\min} - \theta'}_{=:\xi_{\pi,\min}}\right)\right|,
\end{aligned} \quad \text{(B.3)}$$

where $\theta'$ must exist by the linear completeness assumption.

We now define $\Sigma_\mathcal{D}$,

$$\Sigma_\mathcal{D} \coloneqq \mathbb{E}_\mathcal{D}\left[\phi(s,a)\phi(s,a)^\mathsf{T}\right],$$

and $\mathcal{E}_{\varepsilon_r}$,

$$\mathcal{E}_{\varepsilon_r}(s,a) \coloneqq \left|\phi(s,a)^\mathsf{T}\xi_{\pi,\min}\right|, \ \forall(s,a) \in \mathcal{S} \times \mathcal{A}.$$

By definition of $\xi_{\pi,\min}$ and Theorem A.6, we have

$$\begin{aligned}
&\xi_{\pi,\min}^\mathsf{T}\Sigma_\mathcal{D}\xi_{\pi,\min} \leq \varepsilon_b \\
&\implies \left\|\Sigma_\mathcal{D}^{1/2}\xi_{\pi,\min}\right\|_2 \leq \sqrt{\varepsilon_b}.
\end{aligned}$$

Then, for any $(s,a) \in \mathcal{S} \times \mathcal{A}$,

$$\begin{aligned}
\mathcal{E}_{\varepsilon_r}(s,a) &= \left|\phi(s,a)^\mathsf{T}\Sigma_\mathcal{D}^{-1/2}\Sigma_\mathcal{D}^{1/2}\xi_{\pi,\min}\right| \\
&\leq \left\|\phi(s,a)^\mathsf{T}\Sigma_\mathcal{D}^{-1/2}\right\|_2 \left\|\Sigma_\mathcal{D}^{1/2}\xi_{\pi,\min}\right\|_2 \\
&\leq \sqrt{\phi(s,a)^\mathsf{T}\Sigma_\mathcal{D}^{-1}\phi(s,a)}\sqrt{\varepsilon_b}.
\end{aligned} \quad \text{(B.4)}$$

Plugging Eq.(B.4) into Eq.(B.3), we obtain

$$\begin{aligned}
&\left|\phi(s_0,\pi)^\mathsf{T}\theta_{\pi,\min} - J(\pi)\right| \\
&\leq \frac{1}{1-\gamma}\mathbb{E}_{d_\pi}\left[\mathcal{E}_{\varepsilon_r}(s,a)\right] \\
&\leq \frac{\sqrt{\varepsilon_b}}{1-\gamma}\mathbb{E}_{d_\pi}\left[\sqrt{\phi(s,a)^\mathsf{T}\Sigma_\mathcal{D}^{-1}\phi(s,a)}\right].
\end{aligned} \quad \text{(B.5)}$$

Similarly, we also have

$$\left|\phi(s_0,\pi)^\mathsf{T}\theta_{\pi,\max} - J(\pi)\right| \leq \frac{\sqrt{\varepsilon_b}}{1-\gamma}\mathbb{E}_{d_\pi}\left[\sqrt{\phi(s,a)^\mathsf{T}\Sigma_\mathcal{D}^{-1}\phi(s,a)}\right]. \quad \text{(B.6)}$$

Combining Eq.(B.2), Eq.(B.5), and Eq.(B.6),

$$\max_{\theta \in \Theta_{\pi,\varepsilon_r}} \phi(s_0,\pi)^\mathsf{T}\theta - \min_{\theta \in \Theta_{\widehat{\pi},\varepsilon_r}} \phi(s_0,\widehat{\pi})^\mathsf{T}\theta \leq \max_{\theta \in \Theta_{\pi,\varepsilon_r}} \phi(s_0,\pi)^\mathsf{T}\theta - \min_{\theta \in \Theta_{\pi,\varepsilon_r}} \phi(s_0,\pi)^\mathsf{T}\theta$$

$$\leq \frac{2\sqrt{\varepsilon_b}}{1-\gamma} \mathbb{E}_{d_\pi}\left[\sqrt{\phi(s,a)^\mathsf{T}\Sigma_{\mathcal{D}}^{-1}\phi(s,a)}\right]. \qquad (B.7)$$

By the definition of $\varepsilon_r$, we know $\theta_\pi \in \Theta_{\pi,\varepsilon_r}$ for any $\pi \in \Pi$. This implies

$$
\begin{aligned}
J(\pi) - J(\widehat{\pi}) &= Q^\pi(s_0,\pi) - Q^{\widehat{\pi}}(s_0,\widehat{\pi}) \\
&= \phi(s_0,\pi)^\mathsf{T}\theta_\pi - \phi(s_0,\widehat{\pi})^\mathsf{T}\theta_{\widehat{\pi}} \\
&\leq \max_{\theta \in \Theta_{\pi,\varepsilon_r}} \phi(s_0,\pi)^\mathsf{T}\theta - \min_{\theta \in \Theta_{\widehat{\pi},\varepsilon_r}} \phi(s_0,\widehat{\pi})^\mathsf{T}\theta \\
&\leq \frac{2\sqrt{\varepsilon_b}}{1-\gamma} \mathbb{E}_{d_\pi}\left[\sqrt{\phi(s,a)^\mathsf{T}\Sigma_{\mathcal{D}}^{-1}\phi(s,a)}\right],
\end{aligned}
$$

where the last inequality follows from Eq.(B.7). Plugging the definition of $\varepsilon_b$ (defined in Theorem A.6), we completes the proof. $\qquad\square$

## C   Detailed Proofs in Section 4

### C.1   Some Lemmas

We first introduced the necessary lemmas that used in our proofs.

In the following lemma, we show that at every iteration $t$ of Algorithm 1, the estimated Q-function ($f_t$ obtained at the step 3 of Algorithm 1) is actually the true Q-value of $\pi_t$ in a specific MDP $\mathcal{M}_t$, denoted by $Q^{\pi_t}_{\mathcal{M}_t}$, where dynamic of $\mathcal{M}_t$ is same as the ground-truth MDP $\mathcal{M}$ and the difference between the reward functions of $\mathcal{M}$ and $\mathcal{M}_t$ can be controlled.

**Lemma C.1.** *Let $f_t$ satisfies $\mathcal{E}(f_t,\pi_t;\mathcal{D}) \leq \varepsilon$ for some $\pi_t$. Then, there exists an MDP $\mathcal{M}_t = (\mathcal{P}_t, \mathcal{R}_t)$ (the other elements of $\mathcal{M}_t$ are same as the environment MDP $\mathcal{M}$, and also let $R_t(s,a) = \mathbb{E}[\mathcal{R}_t(s,a)]$) with $\mathcal{P}_t = \mathcal{P}$ and $\|R_t(s,a) - R(s,a)\|^2_{2,\mu} \leq \varepsilon_b$, such that $f_t = Q^{\pi_t}_{\mathcal{M}_t}$, where $\varepsilon_b$ is defined in Theorem A.2.*

***Proof of Lemma C.1.*** We can simply set $\mathcal{R}_t = R_t$ is deterministic as,

$$R_t(s,a) := f_t(s,a) - \gamma \mathop{\mathbb{E}}_{s' \sim \mathcal{P}_t(\cdot|s,a)}\left[\sum_{a' \in \mathcal{A}} \pi_t(a'|s')f_t(s',\pi)\right]. \qquad (C.1)$$

Note that, this $R_t$ always exist because the definition above is equivalent to $R_t = (I - \gamma\mathcal{P})f_t$.

With this $\mathcal{P}_t$ and $\mathcal{R}_t$ ($\mathcal{M}_t = (\mathcal{P}_t, \mathcal{R}_t)$), it directly implies that

$$f_t(s,a) = R_t(s,a) + \gamma \mathop{\mathbb{E}}_{s' \sim \mathcal{P}_t(\cdot|s,a)}\left[\sum_{a' \in \mathcal{A}} \pi_t(a'|s')f_t(s',\pi)\right] = (\mathcal{T}^{\pi_t}_{\mathcal{M}_t}f_t)(s,a),$$

which means that $f_t$ is the Q-function of $\pi_t$ in MDP $\mathcal{M}_t$, i.e., $f_t(s,a) = Q^\pi_{\mathcal{M}_t}(s,a)$ for any $(s,a) \in \mathcal{S} \times \mathcal{A}$.

For $\|R_t(s,a) - R(s,a)\|^2_{2,\mu}$, we have

$$\|R_t(s,a) - R(s,a)\|^2_{2,\mu} = \left\|f_t(s,a) - \gamma \mathop{\mathbb{E}}_{s' \sim \mathcal{P}_t(\cdot|s,a)}\left[\sum_{a' \in \mathcal{A}} \pi_t(a'|s')f_t(s',\pi)\right] - R(s,a)\right\|^2_{2,\mu}$$

$$\text{(by Eq.(C.1))}$$

$$= \left\|f_t(s,a) - \gamma \mathop{\mathbb{E}}_{s' \sim \mathcal{P}(\cdot|s,a)}\left[\sum_{a' \in \mathcal{A}} \pi_t(a'|s')f_t(s',\pi)\right] - R(s,a)\right\|^2_{2,\mu}$$

$$= \|f_t - \mathcal{T}^{\pi_t} f_t\|_{2,\mu}^2$$

$$\leq \varepsilon_b,$$

where the last inequality follows from the definition of $\varepsilon_b$ (in Appendix A.1). This completes the proof. $\square$

**Definition 4.** *Consider following procedure: for any $t \in [T]$*

*1. $f_t = \operatorname{argmin}_{f \in \mathcal{F}} \left(f(s_0, \pi_t) + \lambda \mathcal{E}(f, \pi_t; \mathcal{D})\right)$ (step 3 of Algorithm 1)*

*2. $\pi_{t+1}(a|s) \propto \pi_t(a|s) \exp\left(\eta f_t(s,a)\right), \ \forall s, a \in \mathcal{S} \times \mathcal{A}.$*

*Let $J_{\mathcal{M}}(\pi)$ denotes the policy return under MDP $\mathcal{M}$. Then, we define the total regret of the procedure above as*

$$\mathsf{Regret}_T := \max_{\pi \in \Pi} \sum_{i=1}^{T} J_{\mathcal{M}_t}(\pi) - J_{\mathcal{M}_t}(\pi_t).$$

Over this section, we define $\ell_s(\pi)$ as

$$\ell_s(\pi) := \frac{1}{\eta} \sum_{a \in \mathcal{A}} \pi(a|s) \log \pi(a|s).$$

**Lemma C.2.** *For any $\pi \in \Pi$ and $s \in \mathcal{S}$,*

$$\sum_{t=1}^{T} \langle \pi_{t+1}(\cdot|s), f_t(s, \cdot) \rangle - \ell_s(\pi_1) \geq \sum_{t=1}^{T} \langle \pi(\cdot|s), f_t(s, \cdot) \rangle - \ell_s(\pi). \tag{C.2}$$

***Proof of Lemma C.2.*** We establish our proof by induction. The case of $T = 0$ holds as $\pi_1$ is the uniform policy. We assume Eq.(C.2) holds at $T = T'$, then for the case of $T' + 1$, we have the follows for any $\pi \in \Pi$ and $s \in \mathcal{S}$

$$\sum_{t=1}^{T'+1} \langle \pi(\cdot|s), f_t(s, \cdot) \rangle - \ell_s(\pi)$$

$$\overset{(a)}{\leq} \sum_{t=1}^{T'+1} \langle \pi_{T'+2}(\cdot|s), f_t(s, \cdot) \rangle - \ell_s(\pi_{T'+2}(\cdot|s))$$

$$= \sum_{t=1}^{T'} \langle \pi_{T'+2}(\cdot|s), f_t(s, \cdot) \rangle - \ell_s(\pi_{T'+2}(\cdot|s)) + \langle \pi_{T'+2}(\cdot|s), f_{T'+1}(s, \cdot) \rangle$$

$$\overset{(b)}{\leq} \sum_{t=1}^{T'} \langle \pi_{t+1}(\cdot|s), f_t(s, \cdot) \rangle - \ell_s(\pi_1) + \langle \pi_{T'+2}(\cdot|s), f_{T'+1}(s, \cdot) \rangle$$

$$= \sum_{t=1}^{T'+1} \langle \pi_{t+1}(\cdot|s), f_t(s, \cdot) \rangle - \ell_s(\pi_1),$$

where (a) follows from the fact that $\pi_{T'+2}(\cdot|s)$ maximizes $\sum_{t=1}^{T'+1} \langle \pi(\cdot|s), f_t(s, \cdot) \rangle - \ell_s(\pi)$ over any $\pi \in \Pi$, and (b) uses the induction hypothesis that Eq.(C.2) holds at $T = T'$. This completes the proof. $\square$

**Lemma C.3.** *For any $\pi \in \Pi$ and $s \in \mathcal{S}$,*

$$\sum_{t=1}^{T} \langle \pi(\cdot|s) - \pi_t(\cdot|s), f_t(s, \cdot) \rangle \leq \sum_{t=1}^{T} \langle \pi_{t+1}(\cdot|s) - \pi_t(\cdot|s), f_t(s, \cdot) \rangle - \ell_s(\pi_1).$$

**Proof of Lemma C.3.** We use the result of Lemma C.2 to establish the proof.

$$\sum_{t=1}^{T} \langle \pi(\cdot|s) - \pi_t(\cdot|s), f_t(s, \cdot) \rangle$$

$$\leq \sum_{t=1}^{T} \langle \pi(\cdot|s), f_t(s, \cdot) \rangle - \ell_s(\pi) + \ell_s(\pi) - \sum_{t=1}^{T} \langle \pi_t(\cdot|s), f_t(s, \cdot) \rangle$$

$$\leq \sum_{t=1}^{T} \langle \pi_{t+1}(\cdot|s), f_t(s, \cdot) \rangle - \ell_s(\pi_1) - \sum_{t=1}^{T} \langle \pi_t(\cdot|s), f_t(s, \cdot) \rangle + \ell_s(\pi) \qquad \text{(by Lemma C.2)}$$

$$\leq \sum_{t=1}^{T} \langle \pi_{t+1}(\cdot|s) - \pi_t(\cdot|s), f_t(s, \cdot) \rangle - \ell_s(\pi_1). \qquad \qquad \square$$

**Lemma C.4.** *For any $\pi \in \Pi$ and $s \in \mathcal{S}$,*

$$\sum_{t=1}^{T} \langle \pi(\cdot|s) - \pi_t(\cdot|s), f_t(s, \cdot) \rangle \leq 2 V_{\max} \sqrt{2 \log |\mathcal{A}| T},$$

*if we take $\eta = \sqrt{\frac{\log |\mathcal{A}|}{2 V_{\max}^2 T}}$.*

**Proof of Lemma C.4.** We define $\mathcal{L}_{s,t}$ as

$$\mathcal{L}_{s,t}(\pi) := \sum_{t'=1}^{t} \langle \pi(\cdot|s), f_{t'}(s, \cdot) \rangle - \ell_s(\pi).$$

Let $B_{\mathcal{L}_{s,t}}(\cdot \| \cdot)$ and $B_{\ell_s}(\cdot \| \cdot)$ be the Bregman divergence w.r.t. $\mathcal{L}_{s,t}(\cdot)$ and $\ell_s(\cdot)$, then we have

$$\mathcal{L}_{s,t}(\pi_t) \overset{(a)}{=} \mathcal{L}_{s,t}(\pi_{t+1}) + \langle \pi_t(\cdot|s) - \pi_{t+1}(\cdot|s), \nabla \mathcal{L}_{s,t}(\pi)|_{\pi=\pi_{t+1}} \rangle + B_{\mathcal{L}_{s,t}}(\pi_t(\cdot|s) \| \pi_{t+1}(\cdot|s))$$

$$\overset{(b)}{\leq} \mathcal{L}_{s,t}(\pi_{t+1}) + B_{\mathcal{L}_{s,t}}(\pi_t(\cdot|s) \| \pi_{t+1}(\cdot|s))$$

$$\overset{(c)}{=} \mathcal{L}_{s,t}(\pi_{t+1}) - B_{\ell_s}(\pi_t(\cdot|s) \| \pi_{t+1}(\cdot|s)),$$

where (a) is obtained the the definition of Bregman divergence, (b) follows from the fact that $\pi_{t+1}$ maximizes $\mathcal{L}_{s,t}(\pi)$ by definition, and (c) is because $\mathcal{L}_{s,t}(\pi) + \ell_s(\pi)$ is linear, which does not affect the Bregman divergence.

By reordering the inequality above, we obtain

$$B_{\ell_s}(\pi_t(\cdot|s) \| \pi_{t+1}(\cdot|s)) \leq (\mathcal{L}_{s,t}(\pi_{t+1}) - \mathcal{L}_{s,t}(\pi_t))$$

$$= [\mathcal{L}_{s,t-1}(\pi_{t+1}) - \mathcal{L}_{s,t-1}(\pi_t) + \langle \pi_{t+1}(\cdot|s) - \pi_t(\cdot|s), f_t(s, \cdot) \rangle]$$

$$\leq \langle \pi_{t+1}(\cdot|s) - \pi_t(\cdot|s), f_t(s, \cdot) \rangle, \qquad (C.3)$$

where the last inequality is because $\pi_t$ maximizes $\mathcal{L}_{s,t-1}(\cdot)$.

By applying the Taylor expansion and the mean-value theorem on $B_{\ell_s}$, we can rewrite $B_{\ell_s}$ as

$$B_{\ell_s}(\pi_t(\cdot|s) \| \pi_{t+1}(\cdot|s)) = \frac{1}{2} \| \pi_t(\cdot|s) - \pi_{t+1}(\cdot|s) \|^2_{(H\ell_s)(\pi_{t'})} \qquad (C.4)$$

$$:= \frac{1}{2} (\pi_t(\cdot|s) - \pi_{t+1}(\cdot|s))^{\mathsf{T}} [(H\ell_s)(\pi_{t'})] (\pi_t(\cdot|s) - \pi_{t+1}(\cdot|s)),$$

where $\pi_t' = \alpha \pi_t + (1 - \alpha) \pi_{t+1}$ for some $\alpha \in [0, 1]$, and $H\ell_s$ denotes the Hessian matrix of $\ell_s$.

We now bound $\langle \pi_{t+1}(\cdot|s) - \pi_t(\cdot|s), f_t(s, \cdot) \rangle$ using the results above. By the generalized Cauchy-Schwarz theorem,

$$\langle \pi_{t+1}(\cdot|s) - \pi_t(\cdot|s), f_t(s, \cdot) \rangle \leq \| \pi_{t+1}(\cdot|s) - \pi_t(\cdot|s) \|_{(H\ell_s)(\pi_{t'})} \| f_t(s, \cdot) \|_{(H\ell_s)^{-1}(\pi_{t'})}$$

$$= \sqrt{2 B_{\ell_s}(\pi_t(\cdot|s) \| \pi_{t+1}(\cdot|s))} \| f_t(s, \cdot) \|_{(H\ell_s)^{-1}(\pi_{t'})}$$

$$\text{(by Eq.(C.4))}$$

$$\leq \sqrt{2B_{\ell_s}(\pi_t(\cdot|s)\|\pi_{t+1}(\cdot|s))}\sqrt{\eta}\,\|f_t(s,\cdot)\|_\infty$$
$$\leq \sqrt{2\langle\pi_{t+1}(\cdot|s)-\pi_t(\cdot|s),f_t(s,\cdot)\rangle}\sqrt{\eta}\,\|f_t(s,\cdot)\|_\infty$$
$$\text{(by Eq.(C.3))}$$
$$\leq V_{\max}\sqrt{2\eta\langle\pi_{t+1}(\cdot|s)-\pi_t(\cdot|s),f_t(s,\cdot)\rangle}$$
$$\implies \langle\pi_{t+1}(\cdot|s)-\pi_t(\cdot|s),f_t(s,\cdot)\rangle \leq 2\eta V_{\max}^2.$$

As $\pi_1$ is the uniform policy, we know $\ell_s(\pi_1) = -\log|\mathcal{A}|/\eta$. Therefore, by applying Lemma C.3, we obtain

$$\sum_{t=1}^T \langle\pi-\pi_t(\cdot|s),f_t(s,\cdot)\rangle \leq \sum_{t=1}^T \langle\pi_{t+1}(\cdot|s)-\pi_t(\cdot|s),f_t(s,\cdot)\rangle - \ell_s(\pi_1)$$

$$\leq 2\eta V_{\max}^2 T + \frac{\log|\mathcal{A}|}{\eta}$$

$$= 2V_{\max}\sqrt{2T\log|\mathcal{A}|},$$

where the last step is attained by taking $\eta = \sqrt{\frac{\log|\mathcal{A}|}{2V_{\max}^2 T}}$. $\qquad\square$

**Theorem C.5.** *Let* $\widetilde{\pi} = \text{argmax}_{\pi\in\Pi}\sum_{t=1}^T J_{\mathcal{M}_t}(\pi) - J_{\mathcal{M}_t}(\pi_t)$ *and* $\eta = \sqrt{\frac{\log|\mathcal{A}|}{2V_{\max}^2 T}}$, *we have*

$$\mathsf{Regret}_T \leq \frac{2V_{\max}\sqrt{2T\log|\mathcal{A}|}}{1-\gamma}.$$

*Proof of Theorem C.5.* Using the performance difference lemma, we have

$$\mathsf{Regret}_T \coloneqq \sum_{t=1}^T J_{\mathcal{M}_t}(\widetilde{\pi}) - J_{\mathcal{M}_t}(\pi_t)$$

$$= \frac{1}{1-\gamma}\sum_{t=1}^T \mathbb{E}_{s\sim d_{\widetilde{\pi},\mathcal{M}_t}}\left[Q_{\mathcal{M}_t}^{\pi_t}(s,\widetilde{\pi}) - Q_{\mathcal{M}_t}^{\pi_t}(s,\pi_t)\right]$$
$$\text{(by performance difference lemma [Kakade and Langford, 2002])}$$

$$= \frac{1}{1-\gamma}\sum_{t=1}^T \mathbb{E}_{s\sim d_{\widetilde{\pi},\mathcal{M}_t}}\left[f_t(s,\widetilde{\pi}) - f_t(s,\pi_t)\right] \qquad\text{(by Lemma C.1)}$$

$$= \frac{1}{1-\gamma}\sum_{t=1}^T \mathbb{E}_{s\sim d_{\widetilde{\pi},\mathcal{M}_t}}\left[\langle\widetilde{\pi}(\cdot|s)-\pi_t(\cdot|s),f_t(s,\cdot)\rangle\right]$$

By Lemma C.1 and its proof, we know the dynamics of $\mathcal{M}_t, t\in[T]$ are identical (same as that of the true environment MDP). Let $d_{\widetilde{\pi}} = d_{\widetilde{\pi},\mathcal{M}_t}$ which holds for any $t\in[T]$, and we have

$$\mathsf{Regret}_T = \frac{1}{1-\gamma}\sum_{t=1}^T \mathbb{E}_{s\sim d_{\widetilde{\pi}}}\left[\langle\widetilde{\pi}(\cdot|s)-\pi_t(\cdot|s),f_t(s,\cdot)\rangle\right]$$

$$= \frac{1}{1-\gamma}\mathbb{E}_{s\sim d_{\widetilde{\pi}}}\left[\sum_{t=1}^T \langle\widetilde{\pi}(\cdot|s)-\pi_t(\cdot|s),f_t(s,\cdot)\rangle\right]$$

$$\leq \frac{1}{1-\gamma}\mathbb{E}_{s\sim d_{\widetilde{\pi}}}\left[2V_{\max}\sqrt{2T\log|\mathcal{A}|}\right] \qquad\text{(by Lemma C.4)}$$

$$= \frac{2V_{\max}\sqrt{2T\log|\mathcal{A}|}}{1-\gamma}.$$

This completes the proof. $\qquad\square$

## C.2 Proof of Theorem 4.1

**Lemma C.6.** *For any $\pi \in \Pi$, we have,*

$$\min_{f \in \mathcal{F}}(f(s_0, \pi) + \lambda \mathcal{E}(f, \pi; \mathcal{D})) \leq J(\pi) + \frac{\sqrt{\varepsilon_{\mathcal{F}}}}{1 - \gamma} + \lambda \varepsilon_r,$$

*and,*

$$\max_{f \in \mathcal{F}}(f(s_0, \pi) - \lambda \mathcal{E}(f, \pi; \mathcal{D})) \geq J(\pi) - \frac{\sqrt{\varepsilon_{\mathcal{F}}}}{1 - \gamma} - \lambda \varepsilon_r,$$

*where $\varepsilon_r$ is defined in Eq.(A.2), i.e.,*

$$\varepsilon_r := \frac{139 V_{\max}^2 \log \frac{|\mathcal{F}||\Pi|}{\delta}}{n} + 39 \varepsilon_{\mathcal{F}}.$$

***Proof of Lemma C.6.*** For any $\pi \in \Pi$, let

$$f_\pi := \operatorname*{argmin}_{f \in \mathcal{F}} \sup_{\text{admissible } \nu} \|f - \mathcal{T}^\pi f\|_{2,\nu}^2,$$

then we know $\|f - \mathcal{T}^\pi f\|_{2,\nu}^2 \leq \varepsilon_{\mathcal{F}}$ for any admissible $\nu$. We now also the following arguments

$$\begin{aligned}
J(\pi) &= J(\pi) - (f_\pi(s_0, \pi) - \lambda \mathcal{E}(f_\pi, \pi; \mathcal{D})) + (f_\pi(s_0, \pi) - \lambda \mathcal{E}(f_\pi, \pi; \mathcal{D})) \\
&= (f_\pi(s_0, \pi) - \lambda \mathcal{E}(f_\pi, \pi; \mathcal{D})) + (J(\pi) - f_\pi(s_0, \pi)) + \lambda \mathcal{E}(f_\pi, \pi; \mathcal{D}) \\
&\leq (f_\pi(s_0, \pi) - \lambda \mathcal{E}(f_\pi, \pi; \mathcal{D})) + \frac{\|f_\pi - \mathcal{T}^\pi f_\pi\|_{2, d_\pi}}{1 - \gamma} + \lambda \mathcal{E}(f_\pi, \pi; \mathcal{D}) \\
&\hspace{4cm} \text{(by [Xie and Jiang, 2020, Lemma 1])} \\
&\leq (f_\pi(s_0, \pi) - \lambda \mathcal{E}(f_\pi, \pi; \mathcal{D})) + \frac{\sqrt{\varepsilon_{\mathcal{F}}}}{1 - \gamma} + \lambda \varepsilon_r \hspace{2cm} \text{(by Eq.(A.4))} \\
&\leq \max_{f \in \mathcal{F}}(f(s_0, \pi) - \lambda \mathcal{E}(f, \pi; \mathcal{D})) + \frac{\sqrt{\varepsilon_{\mathcal{F}}}}{1 - \gamma} + \lambda \varepsilon_r
\end{aligned}$$

and

$$\begin{aligned}
J(\pi) &= J(\pi) - (f_\pi(s_0, \pi) + \lambda \mathcal{E}(f_\pi, \pi; \mathcal{D})) + (f_\pi(s_0, \pi) + \lambda \mathcal{E}(f_\pi, \pi; \mathcal{D})) \\
&= (f_\pi(s_0, \pi) + \lambda \mathcal{E}(f_\pi, \pi; \mathcal{D})) + (J(\pi) - f_\pi(s_0, \pi)) - \lambda \mathcal{E}(f_\pi, \pi; \mathcal{D}) \\
&\geq (f_\pi(s_0, \pi) + \lambda \mathcal{E}(f_\pi, \pi; \mathcal{D})) - \frac{\|f_\pi - \mathcal{T}^\pi f_\pi\|_{2, d_\pi}}{1 - \gamma} - \lambda \mathcal{E}(f_\pi, \pi; \mathcal{D}) \\
&\hspace{4cm} \text{(by [Xie and Jiang, 2020, Lemma 1])} \\
&\geq (f_\pi(s_0, \pi) + \lambda \mathcal{E}(f_\pi, \pi; \mathcal{D})) - \frac{\sqrt{\varepsilon_{\mathcal{F}}}}{1 - \gamma} - \lambda \varepsilon_r \hspace{2cm} \text{(by Theorem A.1)} \\
&\geq \min_{f \in \mathcal{F}}(f(s_0, \pi) + \lambda \mathcal{E}(f, \pi; \mathcal{D})) - \frac{\sqrt{\varepsilon_{\mathcal{F}}}}{1 - \gamma} - \lambda \varepsilon_r.
\end{aligned}$$

This completes the proof. $\qquad \square$

***Proof of Theorem 4.1.*** We use $\mathcal{M}_t$ to denote the corresponding MDP of $f_t$ (see Lemma C.1). For any $\pi \in \Pi$, let

$$f_\pi := \operatorname*{argmin}_{f \in \mathcal{F}} \sup_{\text{admissible } \nu} \|f - \mathcal{T}^\pi f\|_{2,\nu}^2.$$

By Lemma C.6, we know

$$\begin{aligned}
J(\pi_t) &\geq \min_{f \in \mathcal{F}}(f(s_0, \pi_t) + \lambda \mathcal{E}(f, \pi_t; \mathcal{D})) - \frac{\sqrt{\varepsilon_{\mathcal{F}}}}{1 - \gamma} - \lambda \varepsilon_r \\
&= f_t(s_0, \pi_t) + \lambda \mathcal{E}(f_t, \pi_t; \mathcal{D}) - \frac{\sqrt{\varepsilon_{\mathcal{F}}}}{1 - \gamma} - \lambda \varepsilon_r
\end{aligned}$$

$$\geq f_t(s_0, \pi_t) - \frac{\sqrt{\varepsilon_{\mathcal{F}}}}{1-\gamma} - \lambda\varepsilon_r$$

$$= J_{\mathcal{M}_t}(\pi_t) - \frac{\sqrt{\varepsilon_{\mathcal{F}}}}{1-\gamma} - \lambda\varepsilon_r. \tag{C.5}$$

Now, we have

$$J(\pi) - J(\bar{\pi}) = \frac{1}{T}\sum_{t=1}^{T}(J(\pi) - J(\pi_t))$$

$$\leq \frac{1}{T}\sum_{t=1}^{T}(J(\pi) - J_{\mathcal{M}_t}(\pi) + J_{\mathcal{M}_t}(\pi) - J_{\mathcal{M}_t}(\pi_t)) + \frac{\sqrt{\varepsilon_{\mathcal{F}}}}{1-\gamma} + \lambda\varepsilon_r \quad \text{(by Eq.(C.5))}$$

$$\leq \frac{1}{T}\sum_{t=1}^{T}(J_{\mathcal{M}_t}(\pi) - J_{\mathcal{M}_t}(\pi_t)) + \frac{1}{T}\sum_{t=1}^{T}(J(\pi) - J_{\mathcal{M}_t}(\pi)) + \frac{\sqrt{\varepsilon_{\mathcal{F}}}}{1-\gamma} + \lambda\varepsilon_r$$

$$\leq \frac{2V_{\max}}{1-\gamma}\sqrt{\frac{2\log|\mathcal{A}|}{T}} + \frac{1}{T}\sum_{t=1}^{T}(J(\pi) - J_{\mathcal{M}_t}(\pi)) + \frac{\sqrt{\varepsilon_{\mathcal{F}}}}{1-\gamma} + \lambda\varepsilon_r.$$

$$\text{(by Lemma C.5)}$$

We now provide the bound on $J(\pi) - J_{\mathcal{M}_t}(\pi)$ for any $t \in [T]$. By a standard telescoping argument (e.g., [Xie and Jiang, 2020, Lemma 1]), we can obtain

$$J(\pi) - J_{\mathcal{M}_t}(\pi)$$
$$= Q^{\pi}(s_0, \pi) - J_{\mathcal{M}_t}(\pi)$$
$$= \frac{1}{1-\gamma}\left|\mathbb{E}_{\mu}\left[\nu/\mu \cdot (Q^{\pi} - \mathcal{T}_{\mathcal{M}_t}^{\pi}Q^{\pi})\right] + \mathbb{E}_{d_{\pi}}\left[Q^{\pi} - \mathcal{T}_{\mathcal{M}_t}^{\pi}Q^{\pi}\right] - \mathbb{E}_{\nu}\left[Q^{\pi} - \mathcal{T}_{\mathcal{M}_t}^{\pi}Q^{\pi}\right]\right|$$
$$\leq \underbrace{\frac{1}{1-\gamma}\left|\mathbb{E}_{\mu}\left[\nu/\mu \cdot (Q^{\pi} - \mathcal{T}_{\mathcal{M}_t}^{\pi}Q^{\pi})\right]\right|}_{\text{(I)}} + \underbrace{\frac{1}{1-\gamma}\left|\mathbb{E}_{d_{\pi}}\left[Q^{\pi} - \mathcal{T}_{\mathcal{M}_t}^{\pi}Q^{\pi}\right] - \mathbb{E}_{\nu}\left[Q^{\pi} - \mathcal{T}_{\mathcal{M}_t}^{\pi}Q^{\pi}\right]\right|}_{\text{(II)}}$$

where $\nu$ is an arbitrary on-support state-action distribution. We now discuss these two terms above separately. Note that, the $d_{\pi}$ we used above is defined to be the distribution under the true environment MDP $\mathcal{M}$, which is equal to $d_{\pi,\mathcal{M}_t}$ as we define $\mathcal{M}_t$ to have the same dynamic as $\mathcal{M}_t$ (details in Lemma C.1).

For the term (I),

$$\text{(I)} = \frac{1}{1-\gamma}\left|\mathbb{E}_{\mu}\left[\nu/\mu \cdot (Q^{\pi} - \mathcal{T}_{\mathcal{M}_t}^{\pi}Q^{\pi})\right]\right|$$

$$\leq \frac{1}{1-\gamma}\|Q^{\pi} - \mathcal{T}_{\mathcal{M}_t}^{\pi}Q^{\pi}\|_{2,\nu},$$

because of the Cauchy-Schwarz inequality for random variables ($|\mathbb{E}[XY]| \leq \sqrt{\mathbb{E}[X^2]\mathbb{E}[Y^2]}$).

We define $(d_{\pi} \setminus \nu)$ as $(d_{\pi} \setminus \nu)(s,a) := \max(d_{\pi}(s,a) - \nu(s,a), 0)$ for any $(s,a) \in \mathcal{S} \times \mathcal{A}$. Then, for the term (II)

$$\text{(II)} = \frac{1}{1-\gamma}\left|\mathbb{E}_{d_{\pi}}\left[Q^{\pi} - \mathcal{T}_{\mathcal{M}_t}^{\pi}Q^{\pi}\right] - \mathbb{E}_{\nu}\left[Q^{\pi} - \mathcal{T}_{\mathcal{M}_t}^{\pi}Q^{\pi}\right]\right|$$

$$= \frac{1}{1-\gamma}\left|\sum_{(s,a)\in\mathcal{S}\times\mathcal{A}}[d_{\pi}(s,a) - \nu(s,a)]\left[Q^{\pi}(s,a) - (\mathcal{T}_{\mathcal{M}_t}^{\pi}Q^{\pi})(s,a)\right]\right|$$

$$= \frac{1}{1-\gamma}\left|\sum_{(s,a)\in\mathcal{S}\times\mathcal{A}}\mathbb{1}(d_{\pi}(s,a) \geq \nu(s,a))[d_{\pi}(s,a) - \nu(s,a)]\left[Q^{\pi}(s,a) - (\mathcal{T}_{\mathcal{M}_t}^{\pi}Q^{\pi})(s,a)\right]\right|$$

$$+ \frac{1}{1-\gamma} \left| \sum_{(s,a)\in\mathcal{S}\times\mathcal{A}} \mathbb{1}(\nu(s,a) > d_\pi(s,a)) \left[ \nu(s,a) - d_\pi(s,a) \right] \left[ Q^\pi(s,a) - (\mathcal{T}^\pi_{\mathcal{M}_t} Q^\pi)(s,a) \right] \right|$$

$$\leq \frac{1}{1-\gamma} \left| \sum_{(s,a)\in\mathcal{S}\times\mathcal{A}} (d_\pi \setminus \nu)(s,a) \left[ Q^\pi(s,a) - (\mathcal{T}^\pi_{\mathcal{M}_t} Q^\pi)(s,a) \right] \right|$$

$$+ \frac{1}{1-\gamma} \sum_{(s,a)\in\mathcal{S}\times\mathcal{A}} \mathbb{1}(\nu(s,a) > d_\pi(s,a)) \left[ \nu(s,a) - d_\pi(s,a) \right] \left| Q^\pi(s,a) - (\mathcal{T}^\pi_{\mathcal{M}_t} Q^\pi)(s,a) \right|$$

$$\leq \frac{1}{1-\gamma} \left| \sum_{(s,a)\in\mathcal{S}\times\mathcal{A}} (d_\pi \setminus \nu)(s,a) \left[ Q^\pi(s,a) - (\mathcal{T}^\pi_{\mathcal{M}_t} Q^\pi)(s,a) \right] \right| + \frac{1}{1-\gamma} \mathbb{E}_\nu \left[ \left| Q^\pi - \mathcal{T}^\pi_{\mathcal{M}_t} Q^\pi \right| \right]$$

$$\leq \frac{1}{1-\gamma} \underbrace{\left| \sum_{(s,a)\in\mathcal{S}\times\mathcal{A}} (d_\pi \setminus \nu)(s,a) \left[ Q^\pi(s,a) - (\mathcal{T}^\pi_{\mathcal{M}_t} Q^\pi)(s,a) \right] \right|}_{\text{(IIa)}} + \frac{1}{1-\gamma} \| Q^\pi - \mathcal{T}^\pi_{\mathcal{M}_t} Q^\pi \|_{2,\nu},$$

where the second inequality follows from the fact of $\nu(s,a) \geq \mathbb{1}(\nu(s,a) > d_\pi(s,a))[\nu(s,a) - d_\pi(s,a)]$ for any $(s,a) \in \mathcal{S} \times \mathcal{A}$, and the last inequality uses the Cauchy-Schwarz inequality for random variables (similar to the argument about the term (I)).

We now discuss the term (IIa),

$$\text{(IIa)} = \left| \sum_{(s,a)\in\mathcal{S}\times\mathcal{A}} (d_\pi \setminus \nu)(s,a) \left[ Q^\pi(s,a) - (\mathcal{T}^\pi_{\mathcal{M}_t} Q^\pi)(s,a) \right] \right|$$

$$= \left| \sum_{(s,a)\in\mathcal{S}\times\mathcal{A}} (d_\pi \setminus \nu)(s,a) \left[ Q^\pi(s,a) - R_{\mathcal{M}_t}(s,a) - \gamma (\mathcal{P}^\pi_{\mathcal{M}_t} Q^\pi)(s,a) \right] \right|$$

$$= \left| \sum_{(s,a)\in\mathcal{S}\times\mathcal{A}} (d_\pi \setminus \nu)(s,a) \left[ Q^\pi(s,a) - (f_t(s,a) - \gamma(\mathcal{P}^{\pi_t} f_t)(s,a)) - \gamma(\mathcal{P}^\pi Q^\pi)(s,a) \right] \right|$$

$$= \left| \sum_{(s,a)\in\mathcal{S}\times\mathcal{A}} (d_\pi \setminus \nu)(s,a) \left[ f_t(s,a) - R(s,a) - \gamma(\mathcal{P}^{\pi_t} f_t)(s,a) \right] \right|$$

$$= \left| \sum_{(s,a)\in\mathcal{S}\times\mathcal{A}} (d_\pi \setminus \nu)(s,a) \left[ f_t(s,a) - (\mathcal{T}^{\pi_t} f_t)(s,a) \right] \right|,$$

where the third equation follows from the definition of $R_{\mathcal{M}_t}(s,a) := f_t(s,a) - \gamma(\mathcal{P}^{\pi_t}_{\mathcal{M}_t} f_t)(s,a) = f_t(s,a) - \gamma(\mathcal{P}^{\pi_t} f_t)(s,a)$ (refer to Lemma C.1 and its proof).

Thus,

$$\text{(II)} \leq \left| \sum_{(s,a)\in\mathcal{S}\times\mathcal{A}} (d_\pi \setminus \nu)(s,a) \left[ f_t(s,a) - (\mathcal{T}^{\pi_t} f_t)(s,a) \right] \right| + \frac{1}{1-\gamma} \| Q^\pi - \mathcal{T}^\pi_{\mathcal{M}_t} Q^\pi \|_{2,\nu}.$$

Combining the bounds of both term (I) and term (II), we have

$$Q^\pi(\pi, s_0) - J_{\mathcal{M}_t}(\pi)$$

$$\leq \frac{2}{1-\gamma}\|Q^\pi - \mathcal{T}_{\mathcal{M}_t}^\pi Q^\pi\|_{2,\nu} + \left|\sum_{(s,a)\in\mathcal{S}\times\mathcal{A}} (d_\pi \setminus \nu)(s,a)\left[f_t(s,a) - (\mathcal{T}^{\pi_t}f_t)(s,a)\right]\right|. \quad \text{(C.6)}$$

For the term $\|Q^\pi - \mathcal{T}_{\mathcal{M}_t}^\pi Q^\pi\|_{2,\nu}$,

$$
\begin{aligned}
\left\|Q^\pi - \mathcal{T}_{\mathcal{M}_t}^\pi Q^\pi\right\|_{2,\nu} &= \left\|Q^\pi - R_t - \gamma\mathcal{P}_{\mathcal{M}_t}^\pi Q^\pi\right\|_{2,\nu} \\
&= \left\|Q^\pi - R_t - \gamma\mathcal{P}^\pi Q^\pi\right\|_{2,\nu} &&\text{(by Lemma C.1)} \\
&= \|R - R_t\|_{2,\nu} \\
&= \|R + \gamma\mathcal{P}^{\pi_t}f_t - f_t\|_{2,\nu} &&\text{(by Lemma C.1)} \\
&= \|\mathcal{T}^{\pi_t}f_t - f_t\|_{2,\nu} \\
&\leq \sqrt{\mathscr{C}(\nu;\mu,\mathcal{F},\pi_t)}\,\|\mathcal{T}^{\pi_t}f_t - f_t\|_{2,\mu} \\
&\leq \sqrt{\mathscr{C}(\nu;\mu,\mathcal{F},\pi_t)}(\sqrt{\varepsilon_b} + \sqrt{V_{\max}/\lambda}),
\end{aligned}
$$

where the last step is obtained by the following argument:

$$
\begin{aligned}
f_t(s_0,\pi_t) + \lambda\mathcal{E}(f_t,\pi_t;\mathcal{D}) &= \min_{f\in\mathcal{F}}(f(s_0,\pi_t) + \lambda\mathcal{E}(f,\pi_t;\mathcal{D})) \\
&\leq f_{\pi_t}(s_0,\pi_t) + \lambda\mathcal{E}(f_{\pi_t},\pi_t;\mathcal{D}) \\
&\leq V_{\max} + \lambda\varepsilon_r. &&\text{(by Theorem A.1)} \\
\implies \mathcal{E}(f_t,\pi_t;\mathcal{D}) &\leq \varepsilon_r + \frac{V_{\max}}{\lambda}
\end{aligned}
$$

Then, applying Theorem A.2, we transfer the bound on $\mathcal{E}(f_t,\pi_t;\mathcal{D})$ to the bound on $\|\mathcal{T}^{\pi_t}f_t - f_t\|_{2,\mu}$.

Since Eq.(C.6) holds for arbitrary on-support state-action distribution $\nu$, we take the minimal over the set of all $\{\nu : \mathscr{C}(\nu;\mu,\mathcal{F},\pi_t) \leq C_{2,t}\}$ ($C_{2,t}$ denotes the $L^2$ concentrability threshold), and obtain

$$
\begin{aligned}
& Q^\pi(s_0,\pi) - J_{\mathcal{M}_t}(\pi) \\
&\leq \min_{\nu:\mathscr{C}(\nu;\mu,\mathcal{F},\pi_t)\leq C_{2,t}} \left(\frac{2\mathscr{C}(\nu;\mu,\mathcal{F},\pi_t)(\sqrt{\varepsilon_b}+\sqrt{V_{\max}/\lambda})}{1-\gamma}\right. \\
&\quad \left. + \left|\sum_{(s,a)\in\mathcal{S}\times\mathcal{A}}(d_\pi\setminus\nu)(s,a)\left[f_t(s,a)-(\mathcal{T}^{\pi_t}f_t)(s,a)\right]\right|\right) \\
&\leq \frac{2\sqrt{C_{2,t}}(\sqrt{\varepsilon_b}+\sqrt{V_{\max}/\lambda})}{1-\gamma} + \min_{\nu:\mathscr{C}(\nu;\mu,\mathcal{F},\pi_t)\leq C_{2,t}}\left|\sum_{(s,a)\in\mathcal{S}\times\mathcal{A}}(d_\pi\setminus\nu)(s,a)\left[f_t(s,a)-(\mathcal{T}^{\pi_t}f_t)(s,a)\right]\right|.
\end{aligned}
$$

Therefore, we complete the proof as follows.

$$
\begin{aligned}
& J(\pi) - J(\bar\pi) \\
&\leq \frac{2V_{\max}}{1-\gamma}\sqrt{\frac{2\log|\mathcal{A}|}{T}} + \frac{1}{T}\sum_{t=1}^T (J(\pi) - J_{\mathcal{M}_t}(\pi)) + \frac{\sqrt{\varepsilon_\mathcal{F}}}{1-\gamma} + \lambda\varepsilon_r \\
&\leq \frac{2V_{\max}}{1-\gamma}\sqrt{\frac{2\log|\mathcal{A}|}{T}} + \frac{\sqrt{\varepsilon_\mathcal{F}}}{1-\gamma} + \lambda\varepsilon_r + \frac{1}{T}\sum_{t=1}^T\left(\frac{2\sqrt{C_{2,t}}(\sqrt{\varepsilon_b}+\sqrt{V_{\max}/\lambda})}{1-\gamma}\right. \\
&\quad \left. + \min_{\nu:\mathscr{C}(\nu;\mu,\mathcal{F},\pi_t)\leq C_{2,t}}\left|\sum_{(s,a)\in\mathcal{S}\times\mathcal{A}}(d_\pi\setminus\nu)(s,a)\left[f_t(s,a)-(\mathcal{T}^{\pi_t}f_t)(s,a)\right]\right|\right) \\
&= \frac{2V_{\max}}{1-\gamma}\sqrt{\frac{2\log|\mathcal{A}|}{T}} + \frac{\sqrt{\varepsilon_\mathcal{F}}}{1-\gamma} + \lambda\varepsilon_r + \frac{1}{T}\sum_{t=1}^T\left(\frac{2\sqrt{C_{2,t}}(\sqrt{\varepsilon_b}+\sqrt{V_{\max}/\lambda})}{1-\gamma}\right.
\end{aligned}
$$

$$+ \min_{\nu:\mathscr{C}(\nu;\mu,\mathcal{F},\pi_t)\leq C_{2,t}} \left| \sum_{(s,a)\in\mathcal{S}\times\mathcal{A}} (d_\pi \setminus \nu)(s,a)\left[f_t(s,a) - (\mathcal{T}^{\pi_t} f_t)(s,a)\right] \right| \Bigg),$$

where $C_{2,t}$ can be chosen arbitrarily for any $t \in [T]$. Since the complexity of $\Pi_{\mathrm{SPI}}$ is at most $|\mathcal{F}|^T$ by it definition, setting $\lambda = \sqrt[3]{V_{\max}/(1-\gamma)^2 \varepsilon_r^2}$ and plugging the definition of $\varepsilon_b$ and $\varepsilon_r$ (defined in Appendix A) completes the proof. $\qquad\square$

## D  Linear Implementation of PSPI

In this section, we provide the details of implementing PSPI with linear function approximation, that is, $\mathcal{F} := \{\phi(\cdot,\cdot)^\mathsf{T}\theta : \theta \in \mathbb{R}^d\}$, where $\phi : \mathcal{S} \times \mathcal{A} \in \mathbb{R}^d$ is a given feature map.

Recall that Eq.(4.1) is

$$f(s_0,\pi) + \lambda\mathcal{E}(f,\pi;\mathcal{D})$$
$$:= \phi(s_0,\pi)^\mathsf{T}\theta$$
$$+ \lambda\left(\mathbb{E}_\mathcal{D}\left[\left(\phi(s,a)^\mathsf{T}\theta - r - \gamma\phi(s',\pi)^\mathsf{T}\theta\right)^2\right] - \min_{\theta'\in\mathbb{R}^d}\mathbb{E}_\mathcal{D}\left[\left(\phi(s,a)^\mathsf{T}\theta' - r - \gamma\phi(s',\pi)^\mathsf{T}\theta\right)^2\right]\right).$$

We first provide a closed-form solution to the inner $\min_{\theta'\in\mathbb{R}^d}$. Note that this inner $\min$ is a linear regression objective, so the minimal value can be achieved with

$$\theta' = \Sigma^\dagger\mathbb{E}_\mathcal{D}[\phi(s,a)(r + \gamma\phi(s',\pi)^\top\theta)].$$

where $\Sigma := \mathbb{E}_\mathcal{D}[\phi(s,a)\phi(s,a)^\mathsf{T}]$ is the sample covariance matrix and $\Sigma^\dagger$ is its pseudo-inverse. Here, we do not require the invertibility of $\Sigma$, since we only care about the $\min$ value instead of the $\arg\min_{\theta'}$. The $\min$ value is therefore

$$\mathbb{E}_\mathcal{D}\left[\left(\phi(s,a)^\mathsf{T}\Sigma^\dagger\mathbb{E}_\mathcal{D}[\phi(s,a)(r + \gamma\phi(s',\pi)^\top\theta)] - r - \gamma\phi(s',\pi)^\mathsf{T}\theta\right)^2\right]. \tag{D.1}$$

It should be clear at this point that Eq.(4.1) is quadratic in $\theta$. The rest of the derivation provides a simplified closed-form expression. Define shorthand notation

$$\phi := \phi(s,a), \qquad \psi := \phi(s',\pi),$$
$$B := \mathbb{E}_\mathcal{D}\left[\phi\psi^\mathsf{T}\right], \qquad C := \mathbb{E}_\mathcal{D}\left[\psi\psi^\mathsf{T}\right], \quad b := \mathbb{E}_\mathcal{D}\left[\phi r\right], \quad c := \mathbb{E}_\mathcal{D}\left[\psi r\right].$$

Then, Eq.(D.1) is

$$\mathbb{E}_\mathcal{D}\left[\left(\phi^\mathsf{T}\Sigma^\dagger\mathbb{E}_\mathcal{D}[\phi(r + \gamma\psi^\top\theta)] - r - \gamma\psi^\mathsf{T}\theta\right)^2\right]$$
$$= \mathbb{E}_\mathcal{D}\left[\left(\phi^\mathsf{T}\Sigma^\dagger(b + \gamma B\theta) - r - \gamma\psi^\mathsf{T}\theta\right)^2\right]$$
$$= \mathbb{E}_\mathcal{D}\left[\left(\gamma\left(\phi^\mathsf{T}\Sigma^\dagger B - \psi^\mathsf{T}\right)\theta + \phi^\top\Sigma^\dagger b - r\right)^2\right]$$
$$= \gamma^2\mathbb{E}_\mathcal{D}[\theta^\mathsf{T}(B^\mathsf{T}\Sigma^\dagger\phi - \psi)(\phi^\mathsf{T}\Sigma^\dagger B - \psi^\mathsf{T})\theta] + 2\gamma\mathbb{E}_\mathcal{D}[\theta^\mathsf{T}(B^\mathsf{T}\Sigma^\dagger\phi - \psi)(\phi^\mathsf{T}\Sigma^\dagger b - r)] + \text{constant},$$

where "constant" is any term that is independent of $\theta$ and will not affect the optimization. Dropping the constant, the above is equal to

$$\gamma^2\mathbb{E}_\mathcal{D}[\theta^\mathsf{T}(B^\mathsf{T}\Sigma^\dagger\phi\phi^\mathsf{T}\Sigma^\dagger B - \psi\phi^\mathsf{T}\Sigma^\dagger B - B^\mathsf{T}\Sigma^\dagger\phi\psi^T + \psi\psi^\mathsf{T})\theta]$$
$$+ 2\gamma\mathbb{E}_\mathcal{D}[\theta^\mathsf{T}(B^\mathsf{T}\Sigma^\dagger\phi\phi^\mathsf{T}\Sigma^\dagger b - \psi\phi^\mathsf{T}\Sigma^\dagger b - B^\mathsf{T}\Sigma^\dagger\phi r + \psi r)]$$
$$= \gamma^2\theta^\mathsf{T}(B^\mathsf{T}\Sigma^\dagger\Sigma\Sigma^\dagger B - 2B^\mathsf{T}\Sigma^\dagger B + C)\theta + 2\gamma\theta^\mathsf{T}(B^\mathsf{T}\Sigma^\dagger\Sigma\Sigma^\dagger b - 2B^\mathsf{T}\Sigma^\dagger b + c)$$
$$= \gamma^2\theta^\mathsf{T}(B^\mathsf{T}\Sigma^\dagger B - 2B^\mathsf{T}\Sigma^\dagger B + C)\theta + 2\gamma\theta^\mathsf{T}(c - B^\mathsf{T}\Sigma^\dagger b)$$
$$\qquad\qquad\qquad\qquad \text{(Property of pseudo-inverse: } \Sigma^\dagger\Sigma\Sigma^\dagger = \Sigma^\dagger)$$
$$= \gamma^2\theta^\mathsf{T}(C - B^\mathsf{T}\Sigma^\dagger B)\theta + 2\gamma\theta^\mathsf{T}(c - B^\mathsf{T}\Sigma^\dagger b).$$

We now handle the first expectation in Eq.(4.1):

$$\mathbb{E}_\mathcal{D}\left[\left(\phi(s,a)^\mathsf{T}\theta - r - \gamma\phi(s',\pi)^\mathsf{T}\theta\right)^2\right]$$

$$\begin{aligned}
&= \mathbb{E}_{\mathcal{D}}\left[\left((\phi^\mathsf{T} - \gamma\psi^\mathsf{T})\theta - r\right)^2\right] \\
&= \mathbb{E}_{\mathcal{D}}\left[\theta^\mathsf{T}(\phi - \gamma\psi)(\phi^\mathsf{T} - \gamma\psi^\mathsf{T})\theta\right] - 2\mathbb{E}_{\mathcal{D}}[r \cdot (\phi - \gamma\psi)^\mathsf{T}\theta] + \text{constant} \\
&= \theta^\mathsf{T}(\Sigma - \gamma B^\mathsf{T} - \gamma B + \gamma^2 C)\theta - 2(b - \gamma c)\theta + \text{constant}.
\end{aligned}$$

We now combine the two terms and consider the quadratic term $\theta^\mathsf{T}(\cdot)\theta$ and the linear term $\theta^\mathsf{T}(\cdot)$ separately. The matrix in the quadratic term is

$$\begin{aligned}
&\Sigma - \gamma B^\mathsf{T} - \gamma B + \gamma^2 C - \gamma^2(C - B^\mathsf{T}\Sigma^\dagger B) \\
&= \Sigma - \gamma B^\mathsf{T} - \gamma B + \gamma^2 B^\mathsf{T}\Sigma^\dagger B \\
&= (I - \gamma\Sigma^\dagger B)^\mathsf{T}\Sigma(I - \gamma\Sigma^\dagger B).
\end{aligned}$$

To verify the last step, we can expand the last expression and obtain $\Sigma - \gamma\Sigma\Sigma^\dagger B - \gamma(\Sigma\Sigma^\dagger B)^\mathsf{T} + \gamma^2 B^\mathsf{T}\Sigma^\dagger B$, and the identity holds due to $\Sigma\Sigma^\dagger B = B$; we defer the proof of this fact to the end of this section.

The vector in the linear term is

$$-2(b - \gamma c) - 2\gamma(c - B^\mathsf{T}\Sigma^\dagger b) = -2(I - \gamma B^\mathsf{T}\Sigma^\dagger)b.$$

Putting all together, Eq.(4.1) divided by $\lambda$ is equal to (up to a constant independent of $\theta$):

$$\theta^\mathsf{T}(I - \gamma\Sigma^\dagger B)^\mathsf{T}\Sigma(I - \gamma\Sigma^\dagger B)\theta - \theta^\mathsf{T}\left(2(I - \Sigma^\dagger\gamma B)^\mathsf{T}b - \phi(s_0,\pi)/\lambda\right). \tag{D.2}$$

Note that this objective is quadratic in $\theta$, and the Hessian is always positive semi-definite.

**Closed-form solution under invertibility and connection to LSTDQ** We show that the above objective is intimately connected to LSTDQ [Lagoudakis and Parr, 2003]. In particular, assuming $\Sigma$ and $\Sigma - \gamma B$ are both invertible, Eq.(D.2) becomes

$$\theta^\mathsf{T}(I - \gamma\Sigma^{-1}B)^\mathsf{T}\Sigma(I - \gamma\Sigma^{-1}B)\theta - \theta^\mathsf{T}\left(2(I - \gamma\Sigma^{-1}B)^\mathsf{T}b - \phi(s_0,\pi)/\lambda\right).$$

Note that the quadratic term is now positive definite, and we are minimizing the objective, so the minimizer can be found simply by setting the gradient to $0$, i.e.,

$$2(I - \gamma\Sigma^{-1}B)^\mathsf{T}\Sigma(I - \gamma\Sigma^{-1}B)\theta = 2(I - \gamma\Sigma^{-1}B)^\mathsf{T}b - \phi(s_0,\pi)/\lambda.$$

Define $A := I - \gamma\Sigma^{-1}B$, the closed-form solution is

$$\begin{aligned}
\theta &= A^{-1}\Sigma^{-1}(A^{-1})^\mathsf{T}(A^\mathsf{T}b - \phi(s_0,\pi)/2\lambda) \\
&= A^{-1}\Sigma^{-1}b - A^{-1}\Sigma^{-1}(A^{-1})^\mathsf{T}\phi(s_0,\pi)/2\lambda.
\end{aligned}$$

Note that when we drop the pessimistic term $f(s_0,\pi)$ (i.e., setting $\lambda \to \infty$), the solution becomes $A^{-1}\Sigma^{-1}b$, which is exactly LSTDQ Lagoudakis and Parr [2003]. Antos et al. [2008, Proposition 2] shows a similar result, but their proof is restricted to the invertible case and directly verifies that Eq.(4.1) achieves its minimal possible value $0$ when the LSTDQ solution is plugged in. In contrast, our result in Eq.(D.2) is substantially more general as it does not rely on the invertibility assumptions.

**Proof of $\Sigma\Sigma^\dagger B = B$** We rewrite $\Sigma = \mathbb{E}_{\mathcal{D}}[\phi\phi^\mathsf{T}] = \Phi^\mathsf{T}\mathcal{D}\Phi$, where $\mathcal{D}$ is a diagonal matrix representing the empirical measure of $\mathcal{D}$ over $\mathcal{S} \times \mathcal{A}$, and $\Phi \in \mathbb{R}^{|\mathcal{S}\times\mathcal{A}|\times d}$ is the matrix representation of the entire feature map. Similarly we may write $B = \mathbb{E}_{\mathcal{D}}[\phi\psi^\mathsf{T}] = \Phi^\mathsf{T}\mathcal{D}\Psi$ for some suitable matrix $\Psi$.[3] Define $X = \mathcal{D}^{1/2}\Phi$ and $Y = \mathcal{D}^{1/2}\Psi$, and $\Sigma\Sigma^\dagger B = B$ becomes $(X^\mathsf{T}X)(X^\mathsf{T}X)^\dagger(X^\mathsf{T}Y) = X^\mathsf{T}Y$, which is what we need to show.

To show this, let $X = UZV^\mathsf{T}$ be the SVD of $X$, where $Z \in \mathbb{R}^{r\times r}$ is an invertible diagonal matrix with $r$ being the rank of $X$. Note that $U^\mathsf{T}U = V^\mathsf{T}V = I_r$, i.e., the $r \times r$ identity matrix. Then,

$$\begin{aligned}
&(X^\mathsf{T}X)(X^\mathsf{T}X)^\dagger(X^\mathsf{T}Y) \\
&= VZ^2V^\mathsf{T}VZ^{-2}V^\mathsf{T}VZU^\mathsf{T}Y \\
&= VV^\mathsf{T}VZU^\mathsf{T}Y \\
&= VZU^\mathsf{T}Y = X^\mathsf{T}Y.
\end{aligned}$$

This completes the proof.

---

[3]The $(s,a)$-th row of $\Psi$ is $\mathbb{E}_{\mathcal{D}}[\phi(s',\pi)^\mathsf{T}|s,a]$ for $(s,a)$ in the data, and does not matter otherwise.