# OpenReview forum: "Bellman-consistent Pessimism for Offline Reinforcement Learning"
_NeurIPS.cc/2021/Conference — NeurIPS 2021 Oral_

### Official Review · Reviewer_BXJd · 2021-07-11

**Rating:** 7
**Confidence:** 3

**Summary:**

This paper studies offline reinforcement with function approximation, where the function class is general (but finite), without the strong assumption of coverage of the dataset. It proposes an algorithm based on a new notion of Bellman-consistency combined with pessimism that is applied globally (at the initial state) and not point-wise. Under realizability and completeness, the authors prove general suboptimality guarantees against any policy that depends on the level of "mismatch" with the data distribution. This bias-variance trade-off is optimized implicitly in the analysis without requiring tuning of specific hyperparameters. Finally, the authors provide a computationally efficient version of the algorithm that preserves the nature of these theoretical guarantees.

**Limitations And Societal Impact:**

Yes.

**Main Review:**

This is a good paper. It addresses an important subject - offline RL with general function approximation (actually general without assumptions like bounded Eluder dimension), and attempts to eliminate a problematic assumption - coverage of the dataset is clearly not achievable in many settings. I really like the approach for dealing with limited coverage - the implicit trade-off using the $C_2$ constant is a powerful result that is also very intuitive. I also like the practical variant of the algorithm although its convergence rate is quite slow, since it manages to leverage mirror descent techniques in the offline setting. I am familiar with these only in the online setting, were they previously used for offline RL as well? If so, it should be discussed.

However, I still think that there are some concerns that must be addressed before I can vote this paper a clear accept.
One thing that really bothered me is that $\epsilon_r$ depends on $\epsilon_{\mathcal{F}} $. The problem is that $\epsilon_{\mathcal{F}} $ cannot be computed (even without thinking of efficiency) since we do not have access to the Bellman operator. I understand that in many related work this is completely circumvented by assuming strong realizability (i.e., $\epsilon_{\mathcal{F}} =0 $), but I feel that it is a little bit of a hidden assumption - one cannot really instantiate this algorithm and get the guarantees of theorem 3.1. Do the authors have a suggestion about unknown $\epsilon_{\mathcal{F}} $?

While the authors provide a good amount of discussion, there are a few points where I needed more intuition and explanation.
The first point is definition 1 which is hard to process so early in the paper and could use better explanation/intuition in the lines that follow. For example, is there a good reason for this parameter to be small for many policies?
Another point is in theorem 3.1 where the second term in the bound is very hard to comprehend. Although the authors try to explain this in the discussions that follow, I was still missing a basic description of this term (in words) and maybe some examples.
Moreover, I couldn't understand how figure 1 helps to illustrate it.

As a non-expert in offline reinforcement learning, I was missing more comparisons to related work and clear statements about what a previous paper did compared to what the authors did - both in terms of results and more importantly techniques. For example, what is state-of-the-art for offline RL with coverage explicitly? what techniques did they use? why does it break when there is no coverage? how does your approach deal with that?
A specific point that bothered in the context of related work is line 120 where the authors say that completeness assumption can be only avoided in some rare cases and cite [22]. I looked at that paper and it looks like this case is not that rare - general function class with only realizability assumption albeit with some coverage assumption. Does the "rare" term reflect the coverage assumption? Is it because this is the only paper? I feel like this should be described a lot more clearly.

To me it looks like the bound on the complexity of $\Pi_{SPI} $ in the efficient algorithm is way too trivial. Did you attempt to prove better bounds on it in the general case (of course in many examples it is going to be small)? Is it not possible in the general case? If so, you should provide an example where this complexity really grows exponentially with $T $.

Finally, just two typos I found. In the equation after line 151 you use $\epsilon_r $, but it is defined only a few lines afterwards. Is it supposed to be $\epsilon$ there? Also, I couldn't find equation 4.1. Looks like it should be the equation after line 284. Is that correct?

Post rebuttal: I thank the authors for these clarifications and raise my score to 7.


**Time Spent Reviewing:**

3

---

> ### Author Response · Authors · 2021-08-10
> **Authors' response**
>
> We thank the reviewer for their valuable comments and respond to the questions below.
>
> *****
>
> > Mirror descent in offline RL?
>
> We discussed in Line 277, where indeed mirror descent is mostly used in online RL. To the best of our knowledge we are the first to use mirror descent in offline RL. If you run into other offline RL works using mirror descent, please let us know and we will add them to our references.
>
> *****
>
> > $\varepsilon_r$ depends on $\varepsilon_{\mathcal F}$, but $\varepsilon_{\mathcal F}$ cannot be computed.
>
> This is a common characteristic of version-space-based algorithms (give another example, e.g., OLIVE in [Jiang et al '17]). We will acknowledge and clarify in revision. In practice, we can treat it as a scalar hyperparameter and tune it (among many other hyperparameters that need to be tuned in any offline RL algorithm).
>
> *****
>
> > Definition 1 which is hard to process so early in the paper and could use better explanation/intuition in the lines that follow. For example, is there a good reason for this parameter to be small for many policies?
>
> Roughly speaking, it measures how well our data covers a particular distribution $\nu$ of interest. A simplified form is $|| \nu / \mu ||\_{\infty}$ on Line 132, and our definition is more complicated because we use the structure of the function class $\mathcal{F}$ to tighten such a definition, which is important to obtaining the “doubly robust” result in Corollary 1. In our analysis, we do not need this term to be small for _many_ policies; in fact, in Theorem 3.1, it suffices to have this quantity to be small for only 1 good policy in $\Pi$ (and we will be able to compete with such a policy $\pi$, i.e., $J(\pi) - J(\hat \pi)$ will be small), and this can be easily satisfied if our data covers the $d^\pi$ of this policy we wish to compete with (because we can let $\nu=d^\pi$ in $err_{off}(\pi)$, and $d^\pi \setminus \nu$ becomes $0$, hence $err_{off}(\pi)=0$).
>
> *****
>
> > In theorem 3.1 where the second term in the bound is very hard to comprehend. … Moreover, I couldn't understand how figure 1 helps to illustrate it.
>
> The second term roughly corresponds to the off-policy mass $d^\pi \setminus \nu$. This is a robustness result so that we can still enjoy nice guarantees even if our data does not _exactly_ cover any good policy---we can still compete with any policy whose mass is mostly covered by our data (which corresponds to the right part of $d_\pi$ of the either vertical lines in figure 1, as our result automatically adapts to the best on-support and off-support splitting), and we pay this additional term that depends on how much mass is not covered. The complicated expression is because we further tighten this penalty term using information of the function class (L185).
>
> *****
>
> > What is state-of-the-art for offline RL with coverage explicitly? What techniques did they use? Why does it break when there is no coverage? How does your approach deal with that?
>
> When assuming coverage explicitly, AVI- or API-style algorithms are typically used. Without data coverage, these approaches will run into issues because the value of the uncovered/under-explored states may be over-estimated, and the learned policy may visit such states even if those states are bad. [Liu et al '20] ("Provably Good Batch Reinforcement Learning Without Great Exploration") has an explicit section (sec. 3) discussing that issue. We avoid this by using pessimism, which is a dominant principle to handle data without coverage (e.g., see the offline RL tutorial in NeurIPS 2020).
>
> ***
>
> > Why is [22] rare? Does the "rare" term reflect the coverage assumption? Is it because this is the only paper?
>
> Yes. First, this is the only paper that only requires realizability to our knowledge. Second, the coverage assumption they need is even stronger than the standard version of concentrability that provides full coverage. (By the way, their appendix includes examples of how their approach fails when the strong coverage assumption is violated.) Third, their algorithm requires an indifferentiable partitioning step on top of double enumeration over the function space, and does not lend itself to optimization (see the last paragraph of their intro), so the algorithm remains information-theoretic, whereas our PSPI is both computationally tractable and statistically efficient.
>
> *****
>
> > Complexity of $\Pi_{SPI}$
>
> As shown in the paper, we show how this bound can be improved in concrete settings (e.g., linear). For the general setting, we indeed have considered a few possible techniques that improve the complexity bound (e.g., using knowledge of how $\eta$ decays), but none of them works out cleanly. Anecdotally we are aware that some other research groups have encountered similar problems and it is unclear if anyone has a clean solution to this problem.
>
> *****
>
> > Finally, just two typos I found. In the equation after line 151 you use $\varepsilon_r$, but it is defined only a few lines afterwards. Is it supposed to be $\varepsilon$ there? Also, I couldn't find equation 4.1. Looks like it should be the equation after line 284. Is that correct?
>
> Apologies for the typos and thanks for catching them. Yes, line 151 should be $\varepsilon$, and equation 4.1 should be the one after line 284. We will fix them in revision.

---

### Official Review · Reviewer_uzvJ · 2021-07-15

**Rating:** 7
**Confidence:** 4

**Summary:**

This paper studies the pessimism in offline RL with general function approximation given i.i.d. transition data. An information-theoretic analysis is established for an inefficient method, where instead of point-wise pessimism, a notion of Bellman-consistent pessimism is considered. In this way, the theoretical guarantee is established only under closedness assumptions. An instantiation to linear MDP is considered and improved upon literature. Besides, a practical and efficient policy iteration method is proposed, whose performance is analyzed as well.

**Limitations And Societal Impact:**

Yes

**Main Review:**

Originality:
The work falls under the umbrella of pessimism for offline RL. The notion of Bellman consistent pessimism is novel and settles the problem of offline RL with general function approximation in an elegant and standard way.

Quality:
The results are generally technically sound. I've briefly gone through the proofs and they look sound.

Clarity:
The paper is clearly written and well organized, with examples illustrating implications of main results.

Significance:
This work is an important step forward in understanding and practicing pessimism for offline RL under general function approximation. Though the data generating process (i.i.d. transitions) is a bit simplified, it will open the possibility for future research.

**Time Spent Reviewing:**

6

---

> ### Author Response · Authors · 2021-08-10
> **Authors' response**
>
> We thank the reviewer for their valuable comment. We agree that the case of non-i.i.d. data would be an interesting future direction. This can often be handled by assuming mixing properties of data (e.g., $\beta$-mixing) and leveraging martingale concentration inequalities, as done by Antos et al’08.

---

### Official Review · Reviewer_W5J5 · 2021-07-16

**Rating:** 8
**Confidence:** 4

**Summary:**

The paper studies the idea of pessimism in offline RL not through the addition of (negative) bonuses, but by incorporating pessimism implicitly in the loss function. This brings several advantages, see detailed comments below.

**Main Review:**

There are several things to like about the paper.

1) handling of general function classes
2) approximation error evaluated over distributions rather than point wise
3) related to 2), the transfer error notion is the weakest I’ve seen in this space
4) while the algorithm is theoretical, a computationally tractable implementation is given.

The implementable version has a bad convergence rate but the authors give a sound explanation for why this is the case (increased complexity of the policy class). I agree with them that this can likely be lowered substantially on specific function classes.

While I have a very high opinion of the work, I think in few places the exposition may need to be improved. Already in the abstract:

1) “Our theoretical guarantees only require Bellman closedness as standard in the exploratory setting, in which case bonus-based pessimism fails to provide guarantees”
Asm 2 requires `completeness’ with regards to any policy in the prescribed class, not just the greedy one as commonly done in online RL (e.g., Zanette et al ’20, Jin et al ’21, Du et all ‘21, Jiang et al. '17). Thus, your assumption is much stronger as every policy in your policy class needs to play nice with the prescribed function class.

2) ``Our result improves upon a recent bonus-based approach by O(d) in its sample complexity when the action space is finite.’’
I would put more emphasis that the action space needs to be small (non-exponential in d). There is a well known sqrt(d) gap between the small & large action space already in linear bandits.

Minor: the completeness assumption could be further motivated by recent lower bounds already in the linear setting (which you cover with your more general theory); for policy learning, the work Weisz et al ’21 (Exponential Lower Bounds for Planning in MDPs With Linearly-Realizable Optimal Action-Value Functions) implies an offline lower bound of the same order and likewise Zanette ’21 (Exponential Lower Bounds for Batch Reinforcement Learning: Batch RL can be Exponentially Harder than Online RL) has explicit lower bounds for policy learning in the offline setting.


**Time Spent Reviewing:**

2

---

> ### Author Response · Authors · 2021-08-10
> **Authors' response**
>
> We thank the reviewer for their valuable comments and respond to the questions below.
>
> *****
>
> > Assumption 2 requires 'completeness' with regards to any policy in the prescribed class, not just the greedy one as commonly done in online RL (e.g., Zanette et al '20, Jin et al '21, Du et al '21, Jiang et al. '17)
>
> 1. You are correct that we need “completeness” under $\mathcal T^\pi$ for any $\pi$ in the class, instead of closure only under the Bellman optimality operator $\mathcal T$. However, the latter only makes sense when we can hope to compete with the optimal policy $\pi^\star$, as $\mathcal T$ is closely tied to $Q^\star$ and thus $\pi^\star$. In offline RL without sufficient coverage, if the data does not cover $\pi^\star$, there is no hope to compete with $\pi^\star$, and the policy we can compete with will depend on the properties of the data as we have shown in the paper. In this case, the completeness under the optimality operator $\mathcal T$ is not that useful.
>
> 2. The same assumption is also made in many existing works such as [Antos et al ’08]. In fact, completeness under $\mathcal T$ is mostly used for value iteration algorithms (for approximating $Q^\star$ or $V^\star$). Algorithms that optimize policy based on policy evaluation (such as actor-critic), which form a large family of practical algorithms, need our type of completeness in the analyses.
>
>
> 3. You mentioned that [Jiang et al '17, Du et al '21] need completeness under $\mathcal T$; this is incorrect. They only need realizability of $Q^\star$ (and even that can be relaxed; e.g., in [Jiang et al '17]'s $(\Pi, \mathcal G)$ formulation, they compete with any $\pi$ whose $V^\pi$ is realizable.) In fact, in the regime of general function approximation, online RL seems to require much less demanding function-approximation assumptions than offline RL, and there is reason to suspect that this is a fundamental limitation of offline RL. Also as you mentioned, this limitation can be motivated by recent lower bound results. We will add a discussion about them in our final version.
>
> *****
>
> > I would put more emphasis that the action space needs to be small (non-exponential in $d$)
>
> We completely agree and will revise accordingly.

---

### Official Review · Reviewer_mdnB · 2021-07-19

**Rating:** 8
**Confidence:** 5

**Summary:**

The paper proposes the use of Bellman-consistent pessimism for offline RL with general function approximation (under realizability and completeness assumptions), and manages to obtain non-trivial "dataset-dependent" sample complexity guarantees that generally hold under all possible data coverage scenarios of the dataset (i.e., the sample complexity bound gracefully degrades from guaranteeing a near-optimal policy under standard coverage assumptions to offering no improvement over the data collection policy in the most degenerate case). Such adaptivity makes the proposed approach attractive as it does not rely on any specific data coverage assumption. The paper provides two algorithms: the first attains sharp theoretical guarantees in a computationally inefficient manner, while the second attains worse guarantees in a more practical way.

**Limitations And Societal Impact:**

While Bellman-consistent pessimism brings some theoretical advantages over prior pessimistic approaches, it also introduces additional issues which may make it perform worse than prior approaches in practice. The original form of Bellman-consistent pessimism (described in Section 3) is definitely very difficult to implement, and the paper provides a more practical variant PSPI in Section 4. However, PSPI only has a much slower $O(n^{-1/5})$ rate in general, which makes it not necessarily more appealing than prior algorithms (e.g., the ones in Liu et al. 2020 [15]) when one wants strong statistical performance. Moreover, since the algorithms in Liu et al. (2020) may converge much faster than PSPI (the former should converge in $O(\frac{\log n}{1-\gamma})$ optimization rounds while the latter may require $O(n^{2/5})$ optimization rounds), the computational complexity of PSPI can be much higher (as in each optimization round PSPI needs to solve a regularized loss minimization problem). Finally, while the authors claim that an important advantage of Bellman-consistent pessimism is that it does not require a density estimation step (and I fully agree with this point), the current paper assumes a known deterministic initial state $s_0$; when $s_0$ is generated by an initial state distribution $d_0$, I would expect Bellman-consistent pessimism to require the knowledge of $d_0$ --- note that estimating $d_0$ from the batch data may also involve density estimation, and the final performance bound may degrade with the estimation error of $d_0$.

Of course, every approach has limitations. The paper's theoretical contributions are significant.



**Main Review:**

Originality: While the use of pessimism in offline RL to avoid strong data coverage assumptions has already been studied in many recent papers, this paper has a clear discussion of prior works and clearly demonstrates its new contributions beyond prior works. I particularly enjoy the decomposition of the error bound (into on-support and off-support components) and the related statistical analysis, which bring new technical insights to this field.

Quality: The paper is technically sound and the claimed results are correct.

Clarity: The paper is well-written and I enjoy reading this paper.

Significance: Compared with prior works which use other notions of pessimism to obtain other forms of "dataset-dependent" performance guarantees (e.g., Liu et al. 2020 [15], Jin et al. 2020 [16]), the approach proposed in this paper enjoys several advantages such as more general guarantees under weaker coverage/function class assumptions. The ability to automatically adapt to the best bias-variance tradeoff in terms of $C_2$ is particularly important, as the hyperparameters tuning tasks commonly required in prior works are in fact very challenging in offline RL.
On the downside, I would like to say that most of the results in this paper are still far from being practical (see the limitation section for more details). Still, they are definitely important theoretical contributions.

Minor Comments:

1. In the abstract, the notation $d$ is used without any explanation. It would be better to say something like "$d$-dimensional linear MDP" earlier.

2. Line 69: The use of "where" before "most prior approaches" seems strange. Did you mean "whereas"?

3. Line 154: It should be $\Delta f_{\pi}(s_0,\pi)$ rather than $\Delta f(s_0,\pi)$.

4. Line 177: "Rather than committing to"

5. Algorithm 1: $\eta$ should also be input.

**Time Spent Reviewing:**

8

---

> ### Author Response · Authors · 2021-08-10
> **Authors' response**
>
> We thank the reviewer for their valuable comments and respond to the questions below.
>
> *****
>
> > Most of the results in this paper are still far from being practical... The computational complexity of PSPI can be much higher than [Liu et al 2020].
>
> We agree that the practical use of our algorithm is important for future work. That said, we would like to point out that approaches in [Liu et al 2020] were based on constructing a single pointwise pessimistic MDP that is pessimistic for all policies; while using a single pessimistic MDP makes optimization easy, it can be overly pessimistic. Our paper addressed this issue by optimizing policies against a set of MDPs that dynamically change during optimization (each policy has its own pessimistic MDP). The benefit of our approach is a tighter performance lower bound in optimization, but the cost is the need of solving a more challenging optimization problem.
>
> *****
>
> > Density estimation for $d_0$
>
> We do not need density estimation for $d_0$ when $d_0$ is unknown. Note that our algorithm only depends on $s_0$ (and hence $d_0$) through the $f(s_0, \pi)$ term, which becomes $\mathbb{E}\_{s_0 \sim d_0}[f(s_0, \pi)]$. We can simply approximate this term by its finite-sample approximation, that is, if we have $\mathcal D_0$, a bag of states $s$ sampled from $d_0$, our approximation would be $\frac{1}{|\mathcal D_0|} \sum_{s \in \mathcal D_0} f(s, \pi)$. For finite $\mathcal F$ and $\Pi$ classes, the estimation error for this term (that is, its difference from $\mathbb{E}_{s_0 \sim d_0}[f(s_0, \pi)])$ can be bounded by the standard Hoeffding + union bounding over $\mathcal F$ and $\Pi$. For infinite classes (such as linear), standard uniform convergence argument can be used. The estimation error is small and will be dominated by the existing error terms, so it will not change the results in any significant manner. Liu et al needed density estimation because they need to identify low-density regions and threshold them to perform pessimism, which is not needed in our Bellman-consistent pessimism.
>
> An alternative way of thinking about this issue is to add an “artificial” start state $s_0$ on top of the existing analysis, whose only action $a_0$ transitions to the unknown $d_0$, so that now $d_0$ becomes part of the transition dynamics. At a higher level, the major difficulty in RL is reasoning about how different policies produce different distributions, but $d_0$ is the distribution shared by all policies, which is relatively easy to deal with. It is precisely because of this reason that we chose to perform our analysis assuming $s_0$ is deterministic.

---

> > ### Comment · Reviewer_mdnB · 2021-08-24
> > **Reply to the response**
> >
> > I have read your response and I agree with your points. Thanks for the explanations. I keep my positive evaluation of the paper.

---

> > ### Comment · Reviewer_mdnB · 2021-08-27
> > **An additional question**
> >
> > Just to clarify, when $d_0$ is unknown and you only have the offline dataset $\mathcal{D}$ (i.e., if you do not have the additional dataset $\mathcal{D}_0$ described in your response), can you still run your approach? It seems that identifying $d_0$ from the offline dataset is very hard in this case. I understand that access to $\mathcal{D}_0$ is a very reasonable assumption (which can be easily satisfied if there is an additional label indicating "initial state or not" in each tuple); I just feel that this is still an additional (though not big) requirement on the offline data.

---

> > > ### Author Response · Authors · 2021-08-27
> > > **About d_0**
> > >
> > > Thanks for agreeing with our explanations to your questions.
> > >
> > > Regarding the additional question about $d_0$: you are right, in that we require access to the additional dataset $\mathcal{D}_0$ to run our approach in case $d_0$ is unknown. We would like to emphasize two additional points regarding this:
> > >
> > > 1. As you mentioned, "access to $\mathcal{D}_0$ is ... very reasonable ... (which can be easily satisfied if there is an additional label indicating "initial state or not" in each tuple)". More concretely, many real-world applications for offline RL (in e.g., medical applications, customer relation management, educational games) are episodic tasks, where data are given in the form of trajectories, beginning with starting states and ending in absorbing (terminal) states. In this case, $\mathcal{D}_0$ is simply the collection of the first state in each trajectory.
> > >
> > > 2. In the general function-approximation setting, one often cannot hope to compete with the globally optimal policy $\pi^\star$, because the policy class $\Pi$ cannot represent it and/or the dataset does not cover $d^{\pi^\star}$. In such a scenario, an initial state distribution is necessary to determine the relative performance of different suboptimal policies. That is, $d_0$ is a crucial part of the _problem definition_ for the learning problem to be meaningful, so naturally such an information needs to be made available to the learner in one way or another. It is only in specialized settings where it may be possible to relax such a requirement, but the cost is to introduce strong structural assumptions on the MDP (see e.g., prior works on tabular and linear MDPs), which is arguably much stronger than requiring $\mathcal{D}_0$.

---

### Decision · Program_Chairs · 2021-09-27

**Decision:**

Accept (Oral)

**Comment:**

Reviewers agreed that this paper constitutes a significant step forward for offline reinforcement learning with general function approximation, as it facilitates weaker notions of both transfer error and approximation error than prior work. While the algorithm in the paper is primarily theoretical, the authors also give a more practical/implementable version, which is valuable as well. In addition, the paper is clear and well-written.